# A Unified Approach to Fair Online Learning via Blackwell Approachability

**Evgenii Chzhen**      **Christophe Giraud**      **Gilles Stoltz**

Université Paris-Saclay, CNRS, Laboratoire de mathématiques d'Orsay, 91405, Orsay, France
`{evgenii.chzhen, christophe.giraud, gilles.stoltz} @universite-paris-saclay.fr`

## Abstract

We provide a setting and a general approach to fair online learning with stochastic sensitive and non-sensitive contexts. The setting is a repeated game between the Player and Nature, where at each stage both pick actions based on the contexts. Inspired by the notion of unawareness, we assume that the Player can only access the non-sensitive context before making a decision, while we discuss both cases of Nature accessing the sensitive contexts and Nature unaware of the sensitive contexts. Adapting Blackwell's approachability theory to handle the case of an unknown contexts' distribution, we provide a general necessary and sufficient condition for learning objectives to be compatible with some fairness constraints. This condition is instantiated on (group-wise) no-regret and (group-wise) calibration objectives, and on demographic parity as an additional constraint. When the objective is not compatible with the constraint, the provided framework permits to characterise the optimal trade-off between the two.

## 1   Introduction

Classically, the goal of the decision maker in sequential environment is purely performance driven — she wants to obtain as high reward as if she has had a complete information about the environment. In contrast, algorithmic fairness shifts the attention from the performance-driven behavior by taking into account additional ethical considerations. The latter is often formalized via the notion of fairness constraint [10, 26, 6] on the decision maker's strategies. The goal of this work is to bring to light Blackwell's approachability theory as a suitable theoretical formalism for fair online learning under *group fairness* constraints. The appealing feature of this theory is two-fold: first, it gives explicit criteria when learning is possible; second, if this criteria is met, it comes with an explicit strategy.

It is well known that Blackwell's approachability theory may be used to characterize online learning problems that are tractable and to design strategies to solve them—for instance, for no-regret learning or calibration. Extensive references to such uses may be found in Cesa-Bianchi and Lugosi [5], Perchet [23] and Abernethy et al. [1]. Actually, as noted by the latter two references, no-regret learning, calibration, and approachability imply each other in some sense. The main first achievement of this article is to extend this use to online learning under fairness constraints. This idea, though natural and intuitive, requires some extensions to Blackwell's approachability theory, like ignoring the target set and having to estimate it.

*Related works in fair online learning.* Several frameworks have been proposed to tackle various problems of fairness arising in online learning. Blum et al. [4] consider the problem of online prediction with experts and define fairness via (approximate) equality of average payoffs. Hébert-Johnson et al. [16], Gupta et al. [14] consider the problem of group-wise calibration. (In passing, we may note that Gupta et al. [14] consider some techniques with a flavor of approachability.) Bechavod et al. [2] consider the problem of online binary classification with partial feedback and

equal opportunity constraint [15]. We treat the above works as sources of inspiration; they all differ in the specific (sensitive and non-sensitive) information that the Player may or may not access before or after taking an action. We apply the general formalism of approachability theory to give new insights into online learning under fairness constraints, and approach this goal in a unified (and geometric) way. In particular, the generality of this formalism allows to derive (im)possibility results nearly effortlessly. But we also go beyond such a mere compatibility/incompatibility check between the learning objectives and fairness constraints, and note that approachability theory also gives a clear strategy for the study of trade-offs between incompatible learning objectives and fairness constraints, which often arise in batch setup [6].

*Outline.* We describe our approachability setting in Section 2 and provide some learning objectives (no-regret and calibration) and fairness constraints (group-wise controls, demographic parity, equalized average payoffs) that fit our framework. A slight extension of the classical result of Blackwell [3] is required and discussed in Section 3. We then support the generality of our framework by deriving (im)possibility results for some objective–constraint pairs in Section 4. We also illustrate in Section 5 how this formalism can be used to derive optimal trade-offs (Pareto frontiers) between performance and fairness for incompatible objective–constraint pairs; as an example, we deal with group-wise calibration (studied by [16, 14]) under demographic parity constraint. For the sake of exposition, we deal in Sections 2–5 with stochastic sensitive contexts whose distribution is known; Section 6 explains how to overcome this and develops a theory of approachability relying on ignoring but estimating the target set.

*Notation.* The Euclidean norm is denoted by $\|\cdot\|$, while the $\ell_1$ norm is denoted by $\|\cdot\|_1$. Given a convex closed set $\mathcal{C} \subset \mathbb{R}^d$, we denote by $\mathrm{Proj}_{\mathcal{C}}(\cdot)$ the projection operator onto $\mathcal{C}$ in Euclidean norm.

## 2 Fair online learning cast as an approachability problem

In this section, we propose a setting for fair online learning based on approachability—a theory introduced by Blackwell [3] (see also the more modern expositions by Perchet [25] or Mertens et al. [22]). More precisely, we consider the following repeated game between a Player and Nature, with stochastic contexts. The existence of these contexts is a (minor) variation on the classical statement of the approachability problem.

The Player and Nature have respective finite action sets $\mathcal{A}$ and $\mathcal{B}$. The sets of sensitive and non-sensitive contexts are respectively denoted by $\mathcal{S}$ and $\mathcal{X}$. The set $\mathcal{X}$ is a general Borel set, while $\mathcal{S}$ is a finite set with cardinality denoted by $|\mathcal{S}|$. Typical choices are $\mathcal{S} = \{0, 1\}$ and $\mathcal{X} = \mathbb{R}^m$ for some $m \in \mathbb{N}$. A joint distribution $\mathbf{Q}$ on $\mathcal{X} \times \mathcal{S}$ is fixed and is unknown to the Player. Finally, a (bounded) Borel-measurable vector-valued payoff function $\boldsymbol{m} : \mathcal{A} \times \mathcal{B} \times \mathcal{X} \times \mathcal{S} \to \mathbb{R}^d$, as well as a closed target set $\mathcal{C} \subseteq \mathbb{R}^d$, are given and known by the Player.

At each round $t \geqslant 1$ the pair of non-sensitive and sensitive contexts $(x_t, s_t) \sim \mathbf{Q}$ is generated independently from the past. The Player observes only the non-sensitive context $x_t$; while Nature also observes $x_t$, it may or may not observe the sensitive context $s_t$. Then, Nature and the Player simultaneously pick (possibly in a randomized fashion) $b_t \in \mathcal{B}$ and $a_t \in \mathcal{A}$, respectively. The Player finally accesses the obtained reward $\boldsymbol{m}(a_t, b_t, x_t, s_t)$ and the sensitive context $s_t$, while Nature has a more complete monitoring and may observe $a_t$ and $s_t$. We introduce an observation operation $G$ to indicate whether Nature observes $x_t$ only—i.e., $G(x_t, s_t) = x_t$, the case of Nature's unawareness—or whether Nature observes both contexts—i.e., $G(x_t, s_t) = (x_t, s_t)$, the case of Nature's awareness.

We consider the short-hand notation $\boldsymbol{m}_t := \boldsymbol{m}(a_t, b_t, x_t, s_t)$,

$$\overline{\boldsymbol{m}}_T := \frac{1}{T} \sum_{t=1}^{T} \boldsymbol{m}(a_t, b_t, x_t, s_t), \qquad \text{and} \qquad \overline{\boldsymbol{c}}_T = \mathrm{Proj}_{\mathcal{C}}\big(\overline{\boldsymbol{m}}_T\big) = \arg\min_{\boldsymbol{v} \in \mathcal{C}} \|\overline{\boldsymbol{m}}_T - \boldsymbol{v}\|$$

for the instantaneous and average payoffs of the player, as well as the Euclidean projection of the latter onto the closed set $\mathcal{C}$, respectively. The distance of $\overline{\boldsymbol{m}}_T$ to $\mathcal{C}$ thus equals $d_T := \|\overline{\boldsymbol{m}}_T - \overline{\boldsymbol{c}}_T\|$. The game protocol is summarized on the next page.

We recall that the Player does not know the context distribution $\mathbf{Q}$.

**Definition 1.** *A target set $\mathcal{C}$ is called $\boldsymbol{m}$–approachable by the Player under the distribution $\mathbf{Q}$ if there exists a strategy of the Player such that, for all strategies of the Nature, $\overline{\boldsymbol{m}}_T \to \mathcal{C}$ a.s.*

**Parameters:** Observation operator $G$ for Nature; distribution $\mathbf{Q}$ on $\mathcal{X} \times \mathcal{S}$

**For** $t = 1, 2, \ldots$
  1. Contexts $(x_t, s_t)$ are sampled according to $\mathbf{Q}$, independently from the past;
  2. Simultaneously,
     - Nature observes $G(x_t, s_t)$ and picks $b_t \in \mathcal{B}$;
     - the Player observes $x_t$ and picks an action $a_t \in \mathcal{A}$;
  3. The Player observes the reward $\boldsymbol{m}(a_t, b_t, x_t, s_t)$ and the sensitive context $s_t$, while Nature observes $(a_t, b_t, x_t, s_t)$.

**Aim:** The Player wants to ensure that $\overline{\boldsymbol{m}}_T \to \mathcal{C}$ a.s., i.e., $d_T = \|\overline{\boldsymbol{m}}_T - \overline{\boldsymbol{c}}_T\| \to 0$ a.s.

**Remark 1** (Awareness for the Player). *We are mostly interested in a Player unaware of the sensitive contexts $s_t$ (Gajane and Pechenizkiy [13]). However, the setting above also covers the case of a Player aware of these contexts: simply consider the lifted non-sensitive contexts $x'_t = (x_t, s_t)$.*

We now describe payoff functions and target sets corresponding to online learning objectives or online fairness constraints. They may be combined together. For instance, vanilla calibration corresponds below to the $\boldsymbol{m}_{\mathrm{cal}}$–approachability of a set $\mathcal{C}_{\mathrm{cal}}$, demographic parity, to the $\boldsymbol{m}_{\mathrm{DP}}$–approachability of a set $\mathcal{C}_{\mathrm{DP}}$, so that vanilla calibration under a demographic parity constraint translates into the $(\boldsymbol{m}_{\mathrm{cal}}, \boldsymbol{m}_{\mathrm{DP}})$–approachability of the product set $\mathcal{C}_{\mathrm{cal}} \times \mathcal{C}_{\mathrm{DP}}$. We therefore consider each objective and each constraint as some elementary brick, to be combined with one or several other bricks. We recall that $\mathcal{S}$ is a finite set and will indicate the cases where we only consider $\mathcal{S} = \{0, 1\}$.

We discuss two objectives: no-regret and approximate calibration, as well as three fairness constraints: group-wise (per-group) control, demographic parity, and equal average payoffs.

## 2.1 Statement of the objectives

For the sake of a more compact exposition, we define the objectives in two forms: global objectives (the vanilla form of objectives) and group-wise objectives. We denote $\gamma_s = \mathbb{P}(s_t = s)$, so that $(\gamma_s)_{s \in \mathcal{S}}$ corresponds to the marginal of $\mathbf{Q}$ on $\mathcal{S}$.

**Objective 1: (Vanilla and group-wise) no-regret.** The definition is based on some payoff function $r$, possibly taking contexts into account: at each round $t$, the Player obtains the payoff $r(a_t, b_t, x_t, s_t)$. The aim is to get, on average, almost as much payoff as the best constant action, all things equal. The vanilla (average) regret equals

$$R_T = \min_{a \in \mathcal{A}} \frac{1}{T} \sum_{t=1}^{T} \left( r(a_t, b_t, x_t, s_t) - r(a, b_t, x_t, s_t) \right),$$

while the group-wise (average) regret equals

$$R_{\mathrm{gr},T} = \min_{s \in \mathcal{S}} \min_{a'_s \in \mathcal{A}} \frac{1}{T} \sum_{t=1}^{T} \left( r(a_t, b_t, x_t, s_t) - r(a'_s, b_t, x_t, s_t) \right) \mathbb{I}\{s_t = s\}.$$

The aim is that $\liminf R_T \geqslant 0$ a.s. (no-regret) and $\liminf R_{\mathrm{gr},T} \geqslant 0$ a.s. (group-wise no-regret), respectively. We could replace the $1/T$ factor by a $1/(\gamma_s T)$ factor in the definition of $R_{\mathrm{gr},T}$, as we will do for the $C_T$ calibration criterion, but given the wish of a non-negative limit, this is irrelevant.

Denote by $N = |\mathcal{A}|$ the cardinality of $\mathcal{A}$. No-regret corresponds to the $\boldsymbol{m}_{\mathrm{reg}}$–approachability of $([0, +\infty))^N$, with the global payoff function $\boldsymbol{m}_{\mathrm{reg}}(a, b, x, s) = \left( r(a, b, x, s) - r(a', b, x, s) \right)_{a' \in \mathcal{A}}$. We also duplicate $\boldsymbol{m}_{\mathrm{reg}}$ into the group-wise payoff function

$$\boldsymbol{m}_{\mathrm{gr\text{-}reg}}(a, b, x, s) = \left( \boldsymbol{m}_{\mathrm{reg}}(a, b, x, s) \mathbb{I}\{s' = s\} \right)_{s' \in \mathcal{S}}.$$

Group-wise no-regret then corresponds to the $\boldsymbol{m}_{\mathrm{gr\text{-}reg}}$–approachability of $\mathcal{C}_{\mathrm{gr\text{-}reg}} = ([0, +\infty))^{N|\mathcal{S}|}$.

**Objective 2: Approximate (vanilla or group-wise) calibration.** Online calibration was first solved by Foster and Vohra [12] and Foster [11]; see the monograph by Cesa-Bianchi and Lugosi [5, Section 4.8] for references to other solutions and extensions. For simplicity, we focus on binary outcomes $b_t \in \{0, 1\}$ and ask the Player to provide at each round forecasts $a_t$ in $[0, 1]$, and even in a discretization of $[0, 1]$ based on a fixed number $N \geqslant 2$ of points:

$$\mathcal{A} = \left\{ a^{(k)} := (k - 1/2)/N, \ \ k \in \{1, \ldots, N\} \right\}.$$

Each $x \in [0, 1]$ can be approximated by some $a^{(k)} \in \mathcal{A}$ with $|x - a^{(k)}| \leqslant 1/(2N)$. At each round, the Player picks $k_t \in \{1, \ldots, N\}$ and forecasts $a_t = a^{(k_t)}$. The action set $\mathcal{A}$ can thus be identified with $\{1, \ldots, N\}$.

This problem is actually called $1/N$–calibration or approximate calibration. The global (vanilla) form of the criterion reads

$$C_T = \sum_{k=1}^{N} \left| \frac{1}{T} \sum_{t=1}^{T} \left( a^{(k)} - b_t \right) \mathbb{I}\{k_t = k\} \right|,$$

while the approximate group-wise calibration criterion is defined as

$$C_{\mathrm{gr},T} = \sum_{s \in \mathcal{S}} \sum_{k=1}^{N} \left| \frac{1}{\gamma_s T} \sum_{t=1}^{T} \left( a^{(k)} - b_t \right) \mathbb{I}\{k_t = k\} \mathbb{I}\{s_t = s\} \right|.$$

The aim is that $\limsup C_T \leqslant 1/N$ a.s. or $\limsup C_{\mathrm{gr},T} \leqslant 1/N$, respectively. Note that unlike vanilla calibration, its group-wise version requires to be calibrated on each sensitive attribute $s \in \mathcal{S}$. In particular, the classical $1/T$ factor is replaced by $1/(\gamma_s T)$, the expected number of appearances of $s_t = s$ for $t = 1, \ldots, T$.

Mannor and Stoltz [18] and Abernethy et al. [1] rewrote the problem of approximate calibration as an approachability problem as follows: introduce the global payoff function

$$\boldsymbol{m}_{\mathrm{cal}}(k, b) = \left( (a^{(1)} - b) \, \mathbb{I}\{k = 1\}, \ \ldots, \ (a^{(N)} - b) \, \mathbb{I}\{k = N\} \right),$$

and duplicate it into the group-wise payoff function as follows:

$$\boldsymbol{m}_{\mathrm{gr\text{-}cal}}(k, b, s) = \left( \boldsymbol{m}_{\mathrm{cal}}(k, b) \, \mathbb{I}\{s = s'\} / \gamma_{s'} \right)_{s' \in \mathcal{S}}.$$

The calibration criteria $C_T$ and $C_{\mathrm{gr},T}$ can now be rewritten as the $\ell^1$–norms of the average payoff vectors $\overline{m}_{\mathrm{cal},T}$ and $\overline{m}_{\mathrm{gr\text{-}cal},T}$. Approximate vanilla calibration thus corresponds to the $\boldsymbol{m}_{\mathrm{cal}}$–approachability of $\mathcal{C}_{\mathrm{cal}} = \left\{ \boldsymbol{v} \in \mathbb{R}^N : \|\boldsymbol{v}\|_1 \leqslant 1/N \right\}$, while approximate group-wise calibration corresponds to the $\boldsymbol{m}_{\mathrm{gr\text{-}cal}}$–approachability of $\mathcal{C}_{\mathrm{gr\text{-}cal}} = \left\{ \boldsymbol{v} \in \mathbb{R}^{N|\mathcal{S}|} : \|\boldsymbol{v}\|_1 \leqslant 1/N \right\}$.

Note that non-sensitive contexts play no role in the calibration objectives, but the Player can (and *must*) leverage these non-sensitive contexts to possibly infer sensitive contexts when handling group-wise calibration.

## 2.2 Statement of the fairness constraints

**Fairness constraint 1: Group-wise objectives.** We already considered possibly group-wise objectives above and Section 4 will show that handling them is already a challenge in our setting where the Player is unaware of the sensitive contexts.

**Fairness constraint 2: Demographic parity.** We will consider it only in the setting of approximate calibration and further restrict our attention to the case of two groups: $\mathcal{S} = \{0, 1\}$. The demographic parity criterion measures the difference between the average forecasts issued for the two groups:

$$D_T = \left| \frac{1}{\gamma_0 T} \sum_{t=1}^{T} a_t \, \mathbb{I}\{s_t = 0\} - \frac{1}{\gamma_1 T} \sum_{t=1}^{T} a_t \, \mathbb{I}\{s_t = 1\} \right|.$$

Given the discretization used, the wish is that $\limsup D_T \leqslant 1/N$. Abiding by a demographic parity constraint is equivalent to $\boldsymbol{m}_{\mathrm{DP}}$–approaching $\mathcal{C}_{\mathrm{DP}} = \left\{ (u, v) \in \mathbb{R}^2 : |u - v| \leqslant 1/N \right\}$, where

$$\boldsymbol{m}_{\mathrm{DP}}(k, s) = \left( a^{(k)} \, \mathbb{I}\{s = 0\} / \gamma_0, \ a^{(k)} \, \mathbb{I}\{s = 1\} / \gamma_1 \right).$$

**Fairness constraint 3: Equalized average payoffs.** This criterion is to be combined with a no-regret criterion; in particular, a base payoff function $r$ is considered. We restrict our attention to the case of two groups, $\mathcal{S} = \{0, 1\}$, and measure the difference of average payoffs:

$$P_T = \left| \frac{1}{\gamma_0 T} \sum_{t=1}^{T} r(a_t, b_t, x_t, s_t) \, \mathbb{I}\{s_t = 0\} - \frac{1}{\gamma_1 T} \sum_{t=1}^{T} r(a_t, b_t, x_t, s_t) \, \mathbb{I}\{s_t = 1\} \right| .$$

Ensuring $\limsup P_T \leqslant \varepsilon$ corresponds to $\boldsymbol{m}_{\text{eq-pay}}$–approaching $\mathcal{C}_{\text{eq-pay}} = \{(u, v) \in \mathbb{R}^2 : |u - v| \leqslant \varepsilon\}$, where

$$\boldsymbol{m}_{\text{eq-pay}}(a, b, x, s) = \big( r(a, b, x, 0) \, \mathbb{I}\{s = 0\}/\gamma_0, \ r(a, b, x, 1) \, \mathbb{I}\{s = 1\}/\gamma_1 \big) .$$

**Remark 2.** *Note that in this general form, the equality of average payoffs encompasses the demographic parity constraint. Indeed, the latter is obtained by setting $r(a, b, x, s) = a$ and $\varepsilon = 1/N$.*

### 2.3 Summary table

The table below gives a summary of different criteria and associated pairs of payoff function and target set. We remark that some of the payoff functions depend on the marginals $(\gamma_s)_{s \in \mathcal{S}}$. Meanwhile, Protocol 2.1 assumes the perfect knowledge of the former. In Section 6 we will show how to bypass this issue, transferring all the unknown quantities into the target set and estimating it.

| Criterion | Vector payoff function | Closed convex target set |
|---|---|---|
| Calibration | $\boldsymbol{m}_{\text{cal}}(k, b) = \big( (a^{(k')} - b) \, \mathbb{I}\{k = k'\} \big)_{k' \in \mathcal{A}}$ | $\mathcal{C}_{\text{cal}} = \{ \boldsymbol{v} \in \mathbb{R}^N : \|\boldsymbol{v}\|_1 \leqslant 1/N \}$ |
| Group-calibration | $\boldsymbol{m}_{\text{gr-cal}}(k, b, s) = \big( \boldsymbol{m}_{\text{cal}}(k, b) \, \mathbb{I}\{s = s'\}/\gamma_{s'} \big)_{s' \in \mathcal{S}}$ | $\mathcal{C}_{\text{gr-cal}} = \{ \boldsymbol{v} \in \mathbb{R}^{N|\mathcal{S}|} : \|\boldsymbol{v}\|_1 \leqslant 1/N \}$ |
| No-regret | $\boldsymbol{m}_{\text{reg}}(a, b, x, s) = \big( r(a, b, x, s) - r(a', b, x, s) \big)_{a' \in \mathcal{A}}$ | $\mathcal{C}_{\text{reg}} = \big( [0, +\infty) \big)^N$ |
| Group-no-regret | $\boldsymbol{m}_{\text{gr-reg}}(a, b, x, s) = \big( \boldsymbol{m}_{\text{reg}}(a, b, x, s) \, \mathbb{I}\{s' = s\} \big)_{s' \in \mathcal{S}}$ | $\mathcal{C}_{\text{gr-reg}} = \big( [0, +\infty) \big)^{N|\mathcal{S}|}$ |
| Demographic parity | $\boldsymbol{m}_{\text{DP}}(k, s) = \big( a^{(k)} \, \mathbb{I}\{s = 0\}/\gamma_0, \ a^{(k)} \, \mathbb{I}\{s = 1\}/\gamma_1 \big)$ | $\mathcal{C}_{\text{DP}} = \{ (u, v) \in \mathbb{R}^2 : |u - v| \leqslant 1/N \}$ |
| Equalized payoffs | $\boldsymbol{m}_{\text{eq-pay}}(a, b, x, s) = \big( r(a, b, x, s') \, \mathbb{I}\{s = s'\}/\gamma_{s'} \big)_{s' \in \{0, 1\}}$ | $\mathcal{C}_{\text{eq-pay}} = \{ (u, v) \in \mathbb{R}^2 : |u - v| \leqslant \varepsilon \}$ |

## 3 Approachability theory adapted

We provide a rather straightforward extension of the approachability theory to deal with Protocol 2.1, namely, with the existence of stochastic contexts, drawn according to an unknown distribution $\mathbf{Q}$. We want to characterize closed convex sets that are approachable.

**Pure vs. mixed actions.** To conclude the description of the setting, we provide more details on the randomized draws of the (pure) actions $a_{t+1}$ and $b_{t+1}$ of the Player and Nature at round $t + 1$. We denote by $h_t$ the information available to Player at the end of round $t$, and by $H_t$ the full history of the first $t$ rounds: $h_t = (\boldsymbol{m}_{t'}, x_{t'}, s_{t'})_{t' \leqslant t}$ and $H_t = (a_{t'}, b_{t'}, x_{t'}, s_{t'})_{t' \leqslant t}$. At the beginning of round $t + 1$, the Player thus picks in a $h_t$–measurable way a measurable family $\big( \boldsymbol{p}_{t+1}^x \big)_{x \in \mathcal{X}}$ of probability distributions over $\mathcal{A}$ (i.e., a collection of distributions such that $x \in \mathcal{X} \mapsto \boldsymbol{p}_{t+1}^x$ is Borel-measurable), and then draws $a_{t+1}$ independently at random according to the mixed action $\boldsymbol{p}_{t+1}^{x_{t+1}}$. Similarly, Nature picks in a $H_t$–measurable way a measurable family $\big( \boldsymbol{q}_{t+1}^{G(x, s)} \big)_{(x, s) \in \mathcal{X} \times \mathcal{S}}$ of probability distributions over $\mathcal{B}$, and uses $\boldsymbol{q}_{t+1}^{G(x_{t+1}, s_{t+1})}$ to draw $b_{t+1}$.

**Approachability strategy.** We adapt the original strategy by Blackwell [3] by asuming the existence of and substituting a sequence of estimates $\hat{\mathbf{Q}}_t$ that are $h_t$–adapted in place of the unknown distribution $\mathbf{Q}$. We will assume that this sequence is convergent in the total variation distance in the sense of Assumption 1 below. To state the strategy, we extend linearly $\boldsymbol{m}$: for all probability distributions $\boldsymbol{p}$ over $\mathcal{A}$ and $\boldsymbol{q}$ over $\mathcal{B}$, for all $(x, s) \in \mathcal{X} \times \mathcal{S}$,

$$\boldsymbol{m}(\boldsymbol{p}, \boldsymbol{q}, x, s) = \sum_{a \in \mathcal{A}} \sum_{b \in \mathcal{B}} \boldsymbol{p}(a) \, \boldsymbol{q}(b) \, \boldsymbol{m}(a, b, x, s) .$$

Now, the Player uses an arbitrary measurable family of distributions $(\boldsymbol{p}_1^x)_{x \in \mathcal{X}}$ for the first round, gets the estimate $\hat{\mathbf{Q}}_1$, and then uses, for rounds $t + 1$, where $t \geqslant 1$:

$$(\boldsymbol{p}_{t+1}^x)_{x \in \mathcal{X}} \in \underset{(\boldsymbol{p}^x)_{x \in \mathcal{X}}}{\arg\min} \ \max_{(\boldsymbol{q}^{G(x,s)})_{(x,s) \in \mathcal{X} \times \mathcal{S}}} \left\langle \overline{\boldsymbol{m}}_t - \overline{\boldsymbol{c}}_t, \int_{\mathcal{X} \times \mathcal{S}} \boldsymbol{m}\big( \boldsymbol{p}^x, \boldsymbol{q}^{G(x,s)}, x, s \big) \, \mathrm{d}\hat{\mathbf{Q}}_t(x, s) \right\rangle , \quad (1)$$

where the minimum and maximum are over all measurable families of probability distributions over $\mathcal{A}$ and $\mathcal{B}$, respectively. The Player then gets access to $h_{t+1}$ and may compute the estimate $\hat{\mathbf{Q}}_{t+1}$ to be used at the next round.

**Necessary and sufficient condition for approachability.** We were able to work out such a condition under the assumption that $\mathbf{Q}$ can be estimated well enough, e.g., faster than at a $1/\ln^3(T)$ rate in total variation distance. We recall that the total variation distance between two probability distributions $\mathbf{Q}_1$ and $\mathbf{Q}_2$ on $\mathcal{X} \times \mathcal{S}$ equals (see, e.g., Devroye [8]):

$$\mathrm{TV}(\mathbf{Q}_1, \mathbf{Q}_2) = \sup_{E \subseteq \mathcal{X} \times \mathcal{S}} |\mathbf{Q}_1(E) - \mathbf{Q}_2(E)| = \frac{1}{2} \int_{\mathcal{X} \times \mathcal{S}} |g_1(x,s) - g_2(x,s)| \, \mathrm{d}\mu(x,s) \,,$$

where the supremum is over all Borel sets $E$ of $\mathcal{X} \times \mathcal{S}$, and where $g_1$ and $g_2$ denote densities of $\mathbf{Q}_1$ and $\mathbf{Q}_2$ with respect to a common dominating probability distribution $\mu$.

**Assumption 1** (fast enough sequential estimation of $\mathbf{Q}$). *The sequence of $(h_t)$–adapted estimators $(\hat{\mathbf{Q}}_t)$ used is such that $\sum_{t=1}^{+\infty} \frac{1}{t} \sqrt{\mathbb{E}[\mathrm{TV}^2(\hat{\mathbf{Q}}_t, \mathbf{Q})]} < +\infty$.*

The above assumption implies both $\frac{1}{T} \sum_{t=1}^{T-1} \sqrt{\mathbb{E}[\mathrm{TV}^2(\hat{\mathbf{Q}}_t, \mathbf{Q})]}$ and $\sum_{t \geqslant T+1} \frac{1}{t} \sqrt{\mathbb{E}[\mathrm{TV}^2(\hat{\mathbf{Q}}_t, \mathbf{Q})]}$ converge to zero (with $T$; see Appendix A for details). Assumption 1 is trivially satisfied in the case when $\mathbf{Q}$ is known, as it is sufficient to take $\hat{\mathbf{Q}}_t = \mathbf{Q}$. When both $\mathcal{X}$ and $\mathcal{S}$ are finite sets, we may use the empirical frequencies as estimators $\hat{\mathbf{Q}}_t$; they satisfy $\mathbb{E}[\mathrm{TV}^2(\hat{\mathbf{Q}}_t, \mathbf{Q})] = O(1/t)$; see, e.g., [7, Lemma 3]. The general case of an uncountable $\mathcal{X}$, e.g., $\mathcal{X} = \mathbb{R}^m$ requires results for density estimation in the $L^1$ or $L^2$ norms; such results rely typically on moving averages or kernel estimates and may be found, for instance, in the monographs by Devroye and Györfi [9] and Devroye [8] (see also Tsybakov [28]). Under mild conditions, the estimation takes place at a polynomial rate in total variation distance (e.g., a $T^{-1/5}$ rate in dimension $m = 1$). Note that the needed rate of decrease for $\mathbb{E}[\mathrm{TV}^2(\hat{\mathbf{Q}}_t, \mathbf{Q})]$ in Assumption 1 is extremely slow: a $1/\ln^3(T)$ rate would suffice.

**Assumption 2** (boundedness). *We assume that $\|\boldsymbol{m}\|_{\infty,2} := \max_{a,b \in \mathcal{A} \times \mathcal{B}} \sup_{(x,s) \in \mathcal{X} \times \mathcal{S}} \|\boldsymbol{m}(a,b,x,s)\| < +\infty.$*

We may now state our main result; the distance of $\overline{\boldsymbol{m}}_T$ to $\mathcal{C}$ was denoted by $d_T$ in Protocol 2.1.

**Theorem 1.** *Assume that $\mathcal{C}$ is a closed convex set and that Assumptions 1 (fast enough sequential estimation of $\mathbf{Q}$) and 2 (bounded reward function) are satisfied, then $\mathcal{C}$ is approachable if and only if*

$$\forall (\boldsymbol{q}^{G(x,s)})_{(x,s) \in \mathcal{X} \times \{0,1\}} \quad \exists (\boldsymbol{p}^x)_{x \in \mathcal{X}} \quad s.t. \quad \int_{\mathcal{X} \times \mathcal{S}} \boldsymbol{m}(\boldsymbol{p}^x, \boldsymbol{q}^{G(x,s)}, x, s) \, \mathrm{d}\mathbf{Q}(x,s) \in \mathcal{C} \,. \quad (2)$$

*In this case, the strategy of Eq. (1) achieves the following rates for $L^2$ and almost-sure convergences:*

$$\mathbb{E}[d_T^2] \leqslant \sqrt{\frac{K}{T}} + 4\|\boldsymbol{m}\|_{\infty,2} \overbrace{\frac{1}{T} \sum_{t=1}^{T-1} \sqrt{\mathbb{E}[\mathrm{TV}^2(\hat{\mathbf{Q}}_t, \mathbf{Q})]}}^{:=\overline{\Delta}_T} \qquad and$$

$$\mathbb{P}\left(\sup_{t \geqslant T} d_t \geqslant \varepsilon\right) \leqslant \frac{3K}{T\varepsilon^2} + \frac{16\|\boldsymbol{m}\|_{\infty,2}}{\varepsilon^2}\left(\sqrt{\frac{K}{T-1}} + 2\left(\sup_{t \geqslant T} \overline{\Delta}_t\right)\left(\overline{\Delta}_T + \sum_{t \geqslant T} \frac{1}{t}\sqrt{\mathbb{E}[\mathrm{TV}^2(\hat{\mathbf{Q}}_t, \mathbf{Q})]}\right)\right)$$

*where $K < +\infty$ denotes the maximal distance to $\mathcal{C}$ of an element of the compact set $\boldsymbol{m}(\mathcal{A}, \mathcal{B}, \mathcal{X}, \mathcal{S})$.*

The proof lies in Appendix A. The necessity part of the theorem actually relies on no assumption other than $\mathcal{C}$ being closed; it consists of showing that Nature has a stationary strategy such that there exists $\alpha > 0$ with $d_T \geqslant \alpha$ in the limit, i.e., the average payoff vectors $\overline{\boldsymbol{m}}_T$ remain $\alpha$–away from $\mathcal{C}$ in the limit. This exactly indicates that the underlying fair online learning problem is not tractable: the underlying objectives and underlying fairness constraints cannot be simultaneously satisfied.

## 4 Working out some objective–constraint pairs: (im)possibility results

In this section we apply Theorem 1 to deal with some examples of objective–constraint pairs described in Sections 2.1 and 2.2. Some of them have been considered before in the literature (sometimes in the batch setup) using various tools [4, 16, 21, 14], as discussed in Section 1.

We keep the original criteria and obtain possiblity or impossibility results. This is a first step, meanwhile, Section 5 will explain how to go further and obtain a trade-off, if needed, between the objective and the fairness constraint.

*Additional notation.* We recall that $\gamma_s = \mathbb{P}(s_t = s)$ and denote by $\mathbf{Q}^s$ the conditional distribution of $x_t$ given $s_t = s$, so that $\mathrm{d}\mathbf{Q}(x, s) = \gamma_s \, \mathrm{d}\mathbf{Q}^s(x)$. We denote by $\mathrm{supp}(\mathbf{Q}^s) \subseteq \mathcal{X}$ the support of $\mathbf{Q}^s$.

**Example 1: Vanilla calibration under a demographic parity constraint—achievable.** Consider the following payoff function and target set, obtained by simultaneously considering the objective of vanilla calibration and the constraint of demographic parity: $\boldsymbol{m} = (\boldsymbol{m}_{\mathrm{cal}}, \boldsymbol{m}_{\mathrm{DP}})$ and $\mathcal{C} = \mathcal{C}_{\mathrm{cal}} \times \mathcal{C}_{\mathrm{DP}}$.

Defining $\psi(u_1, u_2) := |u_1 - u_2|$, the approachability condition (2) then reads as follows (where we introduce short-hand notation $\mathtt{C}$ and $\mathtt{DP}$):

$$\forall (\boldsymbol{q}^{G(x,s)})_{(x,s) \in \mathcal{X} \times \{0,1\}} \; \exists (\boldsymbol{p}^x)_{x \in \mathcal{X}} \quad \text{s.t.} \quad \begin{cases} \mathtt{C} := \left\| \displaystyle\int_{\mathcal{X} \times \{0,1\}} \boldsymbol{m}_{\mathrm{cal}}(\boldsymbol{p}^x, \boldsymbol{q}^{G(x,s)}) \, \mathrm{d}\mathbf{Q}(x, s) \right\|_1 \leqslant \dfrac{1}{N} \, ; \\[2em] \mathtt{DP} := \psi\left( \displaystyle\int_{\mathcal{X} \times \{0,1\}} \boldsymbol{m}_{\mathrm{DP}}(\boldsymbol{p}^x, s) \, \mathrm{d}\mathbf{Q}(x, s) \right) \leqslant \dfrac{1}{N} \, . \end{cases}$$

$$(3)$$

Recalling the notation $\mathbf{Q}^0$ and $\mathbf{Q}^1$ for the conditional distributions, we observe that

$$\mathtt{DP} = \left| \int_{\mathcal{X}} \sum_{k=1}^{N} \boldsymbol{p}^x(k) \, a^{(k)} \, \mathrm{d}\mathbf{Q}^0(x) - \int_{\mathcal{X}} \sum_{k=1}^{N} \boldsymbol{p}^x(k) \, a^{(k)} \, \mathrm{d}\mathbf{Q}^1(x) \right| .$$

We now show that the condition in Eq. (3) is satisfied. For any $(\boldsymbol{q}^{G(x,s)})$, we define the family $(\boldsymbol{p}^x)$ as the constant family $(\mathrm{dirac}(Q_{\mathcal{A}}))$, where $\mathrm{dirac}(Q_{\mathcal{A}})$ denotes the Dirac mass supported on $Q_{\mathcal{A}}$, the closest point of $\mathcal{A}$ to $Q := \int_{\mathcal{X} \times \{0,1\}} \boldsymbol{q}^{G(x,s)}(1) \, \mathrm{d}\mathbf{Q}(x, s)$. We have $\mathtt{DP} = 0$ as $\boldsymbol{p}^x$ does not depend on $x$. Substituting the expression for $\boldsymbol{m}_{\mathrm{cal}}$ into the definition of $\mathtt{C}$, we observe that for such a choice of $(\boldsymbol{p}^x)_{x \in \mathcal{X}}$, we have

$$\mathtt{C} = \left| \int_{\mathcal{X} \times \{0,1\}} \left( Q_{\mathcal{A}} - \boldsymbol{q}^{G(x,s)}(1) \right) \mathrm{d}\mathbf{Q}(x, s) \right| \leqslant \frac{1}{2N} + \underbrace{\left| \int_{\mathcal{X} \times \{0,1\}} \left( Q - \boldsymbol{q}^{G(x,s)}(1) \right) \mathrm{d}\mathbf{Q}(x, s) \right|}_{=0},$$

where the inequality holds by taking the effect of discretization in $\mathcal{A}$ into account and by the very definition of $Q$. The condition of Eq. (3) is thus satisfied. Therefore, under Assumption 1 (the existence of fast enough sequential estimators of $\mathbf{Q}$) and thanks to Theorem 1, the vanilla calibration and the demographic parity can be achieved simultaneously no matter the monitoring of the Nature.

**Example 2: Group-wise no-regret—mixed picture.** Let the target set be $\mathcal{C}_{\mathrm{gr\text{-}reg}} = \left( [0, +\infty) \right)^{N|\mathcal{S}|}$ and the payoff function be $\boldsymbol{m}_{\mathrm{gr\text{-}reg}}$, i.e., we consider the case of group-wise no-regret under no additional constraint. The approachability condition in Eq. (2) demands that

$$\forall (\boldsymbol{q}^{G(x,s)}) \; \exists (\boldsymbol{p}^x) \quad \text{s.t.} \quad \int_{\mathcal{X} \times \mathcal{S}} \boldsymbol{m}_{\mathrm{gr\text{-}reg}}(\boldsymbol{p}^x, \boldsymbol{q}^{G(x,s)}) \, \mathrm{d}\mathbf{Q}(x, s) \in \left( [0, +\infty) \right)^{N|\mathcal{S}|}, \qquad \text{i.e.,} \quad (4)$$

$$\forall (a', s), \quad \int_{\mathrm{supp}(\mathbf{Q}^s)} \sum_{a \in \mathcal{A}} \boldsymbol{p}^x(a) \left( \sum_{b \in \mathcal{B}} \boldsymbol{q}^{G(x,s)}(b) \left( r(a, b, x, s) - r(a', b, x, s) \right) \right) \mathrm{d}\mathbf{Q}^s(x) \geqslant 0 \, .$$

No-regret seems a harmless challenge, and it is so when the sensitive context is directly observed by the Player, which we do not assume. (In this case, the Player may simply run several no-regret algorithms in parallel, one per sensitive group $s$.) In our context, the direct observation is emulated in some sense when the non-sensitive context $x$ reveals the sensitive context $s$; this is the case, for instance, when the supports of the distributions $\mathbf{Q}^s$ are pairwise disjoint. Note, however, that these distributions $\mathbf{Q}^s$ are unknown to the Player and need to be learned. The second part of Proposition 1

shows that in this case, the group-wise no-regret may be controlled. We get a similar control in the case of irrelevant sensitive contexts, i.e., not affecting the payoffs and not used by Nature; see the first part of Proposition 1, which corresponds to the case of vanilla no-regret minimization. In both cases, the group-wise no-regret can be controlled under Assumption 1, thanks to Theorem 1. However, as we show by means of counter-examples, these are the only cases that may be favorably dealt with.

**Proposition 1.** *The condition of Eq.* (4) *holds when*

- *the sensitive context is irrelevant, i.e., the payoff function is such that* $r(a, b, x, s) = r(a, b, x)$ *and Nature's monitoring is* $G(x, s) = x$;
- *for all* $s \neq s'$, *it holds that* $\mathrm{supp}(\mathbf{Q}^s) \cap \mathrm{supp}(\mathbf{Q}^{s'}) = \emptyset$, *no matter Nature's monitoring* $G$.

*Otherwise, the condition of Eq.* (4) *may not hold.*

*Proof.* We mimic the classical proof of no-regret by approachability for the positive results. For the *first* positive result: for any $(\boldsymbol{q}^x)$, we define $a^x \in \arg\max_{a \in \mathcal{A}} \sum_{b \in \mathcal{B}} \boldsymbol{q}^x(b)\, r(a, b, x)$ and let $(\boldsymbol{p}^x) = \big(\mathrm{dirac}(a^x)\big)$. For the *second* positive result: fix any $(\boldsymbol{q}^{G(x,s)})$; we define $(\boldsymbol{p}^x)_{x \in \mathcal{X}}$ point-wise as follows. For all $s \in \mathcal{S}$, all $x \in \mathrm{supp}(\mathbf{Q}^s)$, we set $\boldsymbol{p}^x = \mathrm{dirac}(a^x)$, where we validly define $a^x \in \arg\max_{a \in \mathcal{A}} \sum_{b \in \mathcal{B}} \boldsymbol{q}^{G(x,s)}(b)\, r(a, b, x, s)$ on the union of the supports of $(\mathbf{Q}^s)_{s \in \mathcal{S}}$, since they are pair-wise disjoint; we define the $a^x$ arbitrarily elsewhere.

Two counter-examples detailed in Appendix B back up the final part of the proposition: we show that Eq. (4) does not hold. In the first counter-example, the monitoring is $G(x, s) = x$, the payoff function depends on $s$, and the supports of $(\mathbf{Q}^s)_{s \in \mathcal{S}}$ have non negligible intersection. In the second example, the monitoring is $G(x, s) = (x, s)$, the payoff function does not depend on $s$, and the supports of $(\mathbf{Q}^s)_{s \in \mathcal{S}}$ have non negligible intersection. $\qquad\square$

**Example 3: (Vanilla) no-regret under the equalized average payoffs constraint.** For the sake of space we deal with this example in Appendix B, obtaining similar conclusions as that of Blum et al. [4].

## 5 Group-wise calibration under a demographic parity constraint: trade-off

In this section, we consider the problem of group-wise calibration under the demographic parity constraint; in particular, $\mathcal{S} = \{0, 1\}$. As we will see, except for special cases, the corresponding two error criteria cannot be simultaneously smaller than the desired $1/N$ in the limit. However, a (possibly optimal) trade-off may be set between the calibration error $\varepsilon$ and the violation level $\delta$ of demographic parity. To that end, we introduce neighborhoods of the original target sets $\mathcal{C}_{\mathrm{gr\text{-}cal}}$ and $\mathcal{C}_{\mathrm{DP}}$:

$$\mathcal{C}^{\varepsilon}_{\mathrm{gr\text{-}cal}} = \big\{ \boldsymbol{v} \in \mathbb{R}^{2N} : \|\boldsymbol{v}\|_1 \leqslant \varepsilon \big\} \qquad \text{and} \qquad \mathcal{C}^{\delta}_{\mathrm{DP}} = \big\{ (u, v) \in \mathbb{R}^2 : |u - v| \leqslant \delta \big\}.$$

A pair $(\varepsilon, \delta) \in \mathbb{R}_+ \times \mathbb{R}_+$ is said *achievable* when $\mathcal{C}^{\varepsilon}_{\mathrm{gr\text{-}cal}} \times \mathcal{C}^{\delta}_{\mathrm{DP}}$ is approachable with $\boldsymbol{m} = (\boldsymbol{m}_{\mathrm{gr\text{-}cal}}, \boldsymbol{m}_{\mathrm{DP}})$. Theorem 1 provides a characterization of this approachability as well as an associated strategy; in particular, when $(\varepsilon, \delta)$ is achievable, this strategy ensures that the calibration error $C_T$ and the violation $D_T$ of demographic parity satisfy: $\limsup C_T \leqslant \varepsilon$ a.s. and $\limsup D_T \leqslant \delta$ a.s.

The goal of this section is to identify all achievable pairs $(\varepsilon, \delta)$. We will do so by determining, for $\delta \geqslant 0$ of interest, the *smallest* $\varepsilon \geqslant 0$ such that $(\varepsilon, \delta)$ is achievable[1]; we denote it by $\varepsilon^{\star}(\delta)$. The line $\big(\delta, \varepsilon^{\star}(\delta)\big)$ is a Pareto frontier.

**Re-parametrization of the problem.** Under Assumption 1 (the existence of fast enough sequential estimators of $\mathbf{Q}$) and thanks to Theorem 1, the $(\boldsymbol{m}_{\mathrm{gr\text{-}cal}}, \boldsymbol{m}_{\mathrm{DP}})$–approachability of $\mathcal{C}^{\varepsilon}_{\mathrm{gr\text{-}cal}} \times \mathcal{C}^{\delta}_{\mathrm{DP}}$ holds if and only if the condition of Eq. (2) is satisfied. The latter can be stated as follows:

$$\forall (\boldsymbol{q}^{G(x,s)})_{(x,s) \in \mathcal{X} \times \{0,1\}} \ \exists (\boldsymbol{p}^x)_{x \in \mathcal{X}} \quad \text{s.t.} \quad \begin{cases} \left\| \displaystyle\int_{\mathcal{X} \times \{0,1\}} \boldsymbol{m}_{\mathrm{gr\text{-}cal}}\big(\boldsymbol{p}^x, \boldsymbol{q}^{G(x,s)}\big)\, \mathrm{d}\mathbf{Q}(x, s) \right\|_1 \leqslant \varepsilon \,; \\[2ex] \psi\left( \displaystyle\int_{\mathcal{X} \times \{0,1\}} \boldsymbol{m}_{\mathrm{DP}}(\boldsymbol{p}^x, s)\, \mathrm{d}\mathbf{Q}(x, s) \right) \leqslant \delta \,, \end{cases}$$

$$\tag{5}$$

---

[1]Note that if $(\varepsilon, \delta)$ is achievable, then $(\varepsilon', \delta')$ with $\varepsilon' \geqslant \varepsilon$ and $\delta' \geqslant \delta$ is also achievable.

where we recall that $\psi(u_1, u_2) = |u_1 - u_2|$. Now, one can show (see comments after Lemma 3 of Appendix C) that the $\psi(\dots)$ term above is always smaller than $\mathrm{TV}(\mathbf{Q}^0, \mathbf{Q}^1)$. Thus, we can re-parameterize the problem and focus only on $\delta_\tau = \tau \cdot \mathrm{TV}(\mathbf{Q}^0, \mathbf{Q}^1)$, where $\tau \in [0, 1]$.

**Computation of the Pareto frontier.** The condition of Eq. (5) indicates that

$$\varepsilon^\star(\delta_\tau) = \max_{(\boldsymbol{q}^{G(x,s)})} \min_{(\boldsymbol{p}^x)} \left\| \int_{\mathcal{X} \times \{0,1\}} \boldsymbol{m}_{\text{gr-cal}}(\boldsymbol{p}^x, \boldsymbol{q}^{G(x,s)}, s) \, \mathrm{d}\mathbf{Q}(x, s) \right\|_1$$

$$\text{s.t.} \quad \psi\left( \int_{\mathcal{X} \times \{0,1\}} \boldsymbol{m}_{\text{DP}}(\boldsymbol{p}^x, s) \, \mathrm{d}\mathbf{Q}(x, s) \right) \leqslant \tau \cdot \mathrm{TV}(\mathbf{Q}^0, \mathbf{Q}^1) . \tag{6}$$

Propositions 2 and 3 below compute the values (up to the $1/N$ discretization error) of $\varepsilon^\star(\delta_\tau)$ in two scenarios, depending on whether Nature observes the sensitive contexts $s_t$.

**Proposition 2** (Nature awareness: $G(x, s) = (x, s)$)**.** *Under Assumption 1 and with the monitoring $G(x, s) = (x, s)$ for Nature, the Pareto frontier $\left(\varepsilon^\star(\delta_\tau), \delta_\tau\right)_{\tau \in [0,1]}$ of achievable pairs satisfies*

$$\delta_\tau = \tau \cdot \mathrm{TV}(\mathbf{Q}^0, \mathbf{Q}^1) \quad \text{and} \quad 1 - \tau \cdot \mathrm{TV}(\mathbf{Q}^0, \mathbf{Q}^1) \leqslant \varepsilon^\star(\delta_\tau) \leqslant 1 - \tau \cdot \mathrm{TV}(\mathbf{Q}^0, \mathbf{Q}^1) + \frac{1}{N} .$$

**Proposition 3** (Nature unawareness: $G(x, s) = x$)**.** *Under Assumption 1 and with the monitoring $G(x, s) = x$ for Nature, the Pareto frontier $\left(\varepsilon^\star(\delta_\tau), \delta_\tau\right)_{\tau \in [0,1]}$ of achievable pairs satisfies:*

$$\delta_\tau = \tau \cdot \mathrm{TV}(\mathbf{Q}^0, \mathbf{Q}^1) \quad \text{and} \quad (1 - \tau) \cdot \mathrm{TV}(\mathbf{Q}^0, \mathbf{Q}^1) \leqslant \varepsilon^\star(\delta_\tau) \leqslant (1 - \tau) \cdot \mathrm{TV}(\mathbf{Q}^0, \mathbf{Q}^1) + \frac{1}{N} .$$

The parameter $\tau \in [0, 1]$ is set by the user.

We observe that in the case when the true label $b_t$ provided by the Nature can be directly influenced by the sensitive attribute $s_t$, Proposition 2 shows that approximate group-wise calibration with $\varepsilon = 1/N$ is never possible, unless $\mathrm{TV}(\mathbf{Q}^0, \mathbf{Q}^1) = 1$ (and $\tau = 1$ is picked). The latter case corresponds to the situation when the supports of $\mathbf{Q}^0$ and $\mathbf{Q}^1$ are disjoint, hence allowing the Player to infer the sensitive context $s$ from the non-sensitive one $x$, essentially reducing (up to unknown $\mathbf{Q}$) the problem to the previously studied setup of Player's awareness [16].

When the true label $b_t$ provided by the Nature is not *directly* influenced by the sensitive attribute $s_t$ (it is influenced by $s_t$ only via $x_t$), Proposition 3 indicates that calibration is always possible by setting $\tau = 1$, no matter the value of $\mathrm{TV}(\mathbf{Q}^0, \mathbf{Q}^1)$. Interestingly, this proposition also shows that if $\mathrm{TV}(\mathbf{Q}^0, \mathbf{Q}^1) = 0$, i.e., the $x_t$ and the $s_t$ are independent, then the Player is able to achieve calibration and satisfy the demographic parity constraint simultaneously.

# 6 Approachability of an unknown target set

A limitation of the calibration problems under demographic parity constraint discussed in Section 4 (Example 1) and Section 5 is that the unknown probabilities $\gamma_0$ and $\gamma_1$ enter the payoff functions $\boldsymbol{m}_{\text{gr-cal}}$ and $\boldsymbol{m}_{\text{DP}}$. We already pointed out this issue in Section 2.3. Even worse, the trade-off claimed in Propositions 2 and 3 relies on the knowledge of the unknown $\mathrm{TV}(\mathbf{Q}^0, \mathbf{Q}^1)$, to set the values of the achievable pair $(\delta, \varepsilon)$ targeted; that is, the target set is unknown. To bypass the first limitation we transfer the unknown $(\gamma_0, \gamma_1)$ to the target set, which makes the payoff function fully known to the Player. We will then be left with the problem of approaching an unknown target set only. For instance, in the context of Section 5, we can define

$$\widetilde{\boldsymbol{m}}_{\text{gr-cal}}(k, y, s) = \left(\boldsymbol{m}_{\text{cal}}(k, y) \, \mathbb{I}\{s = s'\}\right)_{s'=0,1} \quad \text{and} \quad \widetilde{\boldsymbol{m}}_{\text{DP}}(k, s) = \left(a^{(k)} \, \mathbb{I}\{s = 0\}, \; a^{(k)} \, \mathbb{I}\{s = 1\}\right),$$

and set $\widetilde{\boldsymbol{m}} := (\widetilde{\boldsymbol{m}}_{\text{gr-cal}}, \widetilde{\boldsymbol{m}}_{\text{DP}})$. Taking into account the definition of $\boldsymbol{m}_{\text{cal}}$, we note that $\widetilde{\boldsymbol{m}}$ does not depend on $(\gamma_0, \gamma_1)$. Furthermore, by considering the closed convex target sets

$$\widetilde{\mathcal{C}}^\varepsilon_{\text{gr-cal}} = \left\{ (\boldsymbol{v}_0, \boldsymbol{v}_1) \in \mathbb{R}^{2N} \; : \; \frac{\|\boldsymbol{v}_0\|_1}{\gamma_0} + \frac{\|\boldsymbol{v}_1\|_1}{\gamma_1} \leqslant \varepsilon \right\}, \quad \widetilde{\mathcal{C}}^\delta_{\text{DP}} = \left\{ (u, v) \in \mathbb{R}^2 \; : \; \left| \frac{u}{\gamma_0} - \frac{v}{\gamma_1} \right| \leqslant \delta \right\},$$

we remark that the $(\widetilde{\boldsymbol{m}}_{\text{gr-cal}}, \widetilde{\boldsymbol{m}}_{\text{DP}})$–approachability of $\widetilde{\mathcal{C}}^\varepsilon_{\text{gr-cal}} \times \widetilde{\mathcal{C}}^\delta_{\text{DP}}$ is equivalent to the $(\boldsymbol{m}_{\text{gr-cal}}, \boldsymbol{m}_{\text{DP}})$–approachability of $\mathcal{C}^\varepsilon_{\text{gr-cal}} \times \mathcal{C}^\delta_{\text{DP}}$. The unknown quantities appear only in the target set $\widetilde{\mathcal{C}}^\varepsilon_{\text{gr-cal}} \times \widetilde{\mathcal{C}}^\delta_{\text{DP}}$ (and $\delta$ and $\varepsilon$ count as unknown quantities given the trade-off exhibited), while the payoff $\widetilde{\boldsymbol{m}}$ is known beforehand. Thus, it is sufficient to consider the setup of Protocol 2.1 with an *unknown* target set $\mathcal{C}$.

**Approachability strategy for an unknown target set $\mathcal{C}$.** We still assume that the Player is able to build an $h_t$–adapted sequence of estimates $\hat{\mathbf{Q}}_t$. Additionally, we assume that for $T_r := 2^r$, with $r \geqslant 0$, the Player can construct an $h_{T_r}$–adapted estimate $\hat{\mathcal{C}}_r$ of $\mathcal{C}$. We discuss this assumption at the end of this section. We define $d(\hat{\mathcal{C}}_r, \mathcal{C}) = \sup_{x \in \hat{\mathcal{C}}_r} d(x, \mathcal{C})$.

**Assumption 3.** *There exist $B < +\infty$ and a summable non-increasing sequence $(\beta_r)_{r \geqslant 0}$ such that for all $r \geqslant 0$, the sets $\hat{\mathcal{C}}_r$ are convex closed, with $\|\mathbf{v} - \mathrm{Proj}_{\hat{\mathcal{C}}_r}(\mathbf{v})\| \leqslant B$ for all $\mathbf{v} \in \mathbf{m}(\mathcal{A}, \mathcal{B}, \mathcal{X}, \{0, 1\})$,*

$$\mathbb{P}\big(\mathcal{C} \subset \hat{\mathcal{C}}_r\big) \geqslant 1 - 1/(2T_r), \quad and \quad \max\Big\{\mathbb{E}\big[d(\hat{\mathcal{C}}_r, \mathcal{C})^2\big], \ \mathbb{E}\big[d(\mathcal{C}, \hat{\mathcal{C}}_r)^2\big]\Big\} \leqslant \beta_r^2 \ .$$

For all $r \geqslant 0$ and all $t \in \{T_r, \ldots, T_{r+1} - 1\}$, define $\hat{\mathbf{c}}_t := \mathrm{Proj}_{\hat{\mathcal{C}}_r}(\overline{\mathbf{m}}_t)$. The idea of the approachability strategy is to use $\hat{\mathbf{c}}_t$ in place of $\overline{\mathbf{c}}_t$ in Eq. (1) and update the estimate $\hat{\mathcal{C}}_r$ of the target $\mathcal{C}$ only at the end of rounds $t = T_r$. More precisely, the strategy of the Player is:

$$(\mathbf{p}_{t+1}^x)_{x \in \mathcal{X}} \in \underset{(\mathbf{p}^x)}{\arg\min} \ \underset{(\mathbf{q}^{G(x,s)})}{\max} \ \left\langle \overline{\mathbf{m}}_t - \hat{\mathbf{c}}_t, \int \mathbf{m}\big(\mathbf{p}^x, \mathbf{q}^{G(x,s)}, x, s\big) \, \mathrm{d}\hat{\mathbf{Q}}_t(x, s) \right\rangle . \tag{7}$$

**Theorem 2.** *Under Assumption 3 and the assumptions of Theorem 1, a convex closed set $\mathcal{C}$, unknown to the Player, is $\mathbf{m}$–approachable if and only if Blackwell's condition in Eq. (2) is satisfied. In this case, the strategy of Eq. (7) is an approachability strategy.*

Appendix D.2 provides a proof of Theorem 2. But before we do so, we discuss in Appendix D.1 why and how the target set $\mathcal{C}$ may be estimated by sets $\hat{\mathcal{C}}_r$ satisfying Assumption 3. The construction is idiosyncratic and strongly depends on the problem and exact setting considered (in particular, whether the set of non-sensitive contexts $\mathcal{X}$ is finite or not). We provide an illustration for the target set $\mathcal{C} = \widetilde{\mathcal{C}}_{\mathrm{gr\text{-}cal}}^\varepsilon \times \widetilde{\mathcal{C}}_{\mathrm{DP}}^\delta$ of Section 5, in the case of a finite set $\mathcal{X}$.

# 7   Limitations of the current work and topics for future work

The anonymous reviewers of this article pointed out some limitations to or possible extensions of the current work, which we list now.

We only considered, for the sake of readability, the case of demographic parity with two groups. While the criterion of demographic parity easily extends to more groups, the extension of the trade-offs stated in Section 5 is less clear. It would probably involve the total variation distances $\mathrm{TV}(\mathbf{Q}^s, \mathbf{Q}^{s'})$ between each pair $\mathbf{Q}^s$ and $\mathbf{Q}^{s'}$ of marginal distributions, where $s, s' \in \mathcal{S}$, or the distances $\mathrm{TV}(\mathbf{Q}^s, \mathbf{Q})$.

The complexity of the generic approachability strategy of Section 3 is at least linear in the number of groups (if an approximate solution is used, and even polynomial in this number for an exact solution), see Mannor and Stoltz [18, Sections 3.3 and 3.4]. The convergence rates achieved in Theorem 1 involve total variation distances that would also probably depend in at least a linear fashion on the number of groups, unless some special structure is assumed. Both facts may be an issue for large numbers of groups. More generally, we only provide in this article a generic strategy, that is, a first approach to tackle a given fair online learning problem, but specific strategies may be more efficient and get better regret bounds, in particular for large numbers of groups. (For instance, for group-wise calibration, Gupta et al. [14] base a specific and computationally more efficient strategy on an exponential surrogate loss: this strategy enjoys a sample complexity only logarithmic in the number of groups.) The design of such specific strategies remains largely open.

As mentioned in the introduction, Bechavod et al. [2] consider the objective of online binary classification and an equal-opportunity fairness constraint, under some partial monitoring known as "apple tasting". In this article we considered a bandit monitoring for the Player: she observes the reward obtained at each round. Partial monitoring, which was introduced by Rustichini [27], consists of only receiving feedback which is a (possibly) random function of the actions played by the Player and Nature. A theory of approachability under partial monitoring was initiated by Perchet [24], who stated a necessary and sufficient condition (see also Mannor et al. [19]); Mannor et al. [20] then exhibited a computationally more efficient strategy, with improved convergence rates, and Kwon and Perchet [17] finally obtained the optimal convergence rates. A question to investigate is therefore the extension of the results of this article from a bandit monitoring to a partial monitoring.

## Acknowledgments and Disclosure of Funding

Evgenii Chzhen was fully funded by grant ANR-11-LABX-0056-LMH (Labex LMH, part of Programme d'investissements d'avenir). Christophe Giraud received partial support by grant ANR-19-CHIA-0021-01 ("BiSCottE", Agence Nationale de la Recherche). Gilles Stoltz has no direct funding to acknowledge.

Additional revenues for authors are: Evgenii Chzhen—none; Christophe Giraud—none; Gilles Stoltz—part time employment as an affiliate professor with HEC Paris.

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
