# Supplementary Material for
# "A Unified Approach to Fair Online Learning via Blackwell Approachability"

# by Evgenii Chzhen, Christophe Giraud, Gilles Stoltz

This supplementary material contains all the proofs omitted from the main body. Each section provides the proofs of claims, theorems, or propositions of a section of the main body; more precisely, Appendix A provides proofs for Section 3, Appendix B does so for Section 4, Appendix C, for Section 5, and finally, Appendix D deals with Section 6.

# A Proofs for Section 3

We start by proving a claim stated right after Assumption 1: that

$$\sum_{t=1}^{+\infty} \frac{1}{t}\sqrt{\mathbb{E}\big[\mathrm{TV}^2(\hat{\mathbf{Q}}_t, \mathbf{Q})\big]} := C < +\infty \qquad \text{entails} \qquad \overline{\Delta}_T := \frac{1}{T}\sum_{t=1}^{T-1}\sqrt{\mathbb{E}\big[\mathrm{TV}^2(\hat{\mathbf{Q}}_t, \mathbf{Q})\big]} \longrightarrow 0\,.$$

Indeed,

$$\frac{1}{T}\sum_{t=1}^{T-1}\sqrt{\mathbb{E}\big[\mathrm{TV}^2(\hat{\mathbf{Q}}_t, \mathbf{Q})\big]} = \frac{1}{T}\sum_{t=1}^{\lfloor\sqrt{T}\rfloor}\sqrt{\mathbb{E}\big[\mathrm{TV}^2(\hat{\mathbf{Q}}_t, \mathbf{Q})\big]} + \frac{1}{T}\sum_{t=\lfloor\sqrt{T}\rfloor+1}^{T-1}\sqrt{\mathbb{E}\big[\mathrm{TV}^2(\hat{\mathbf{Q}}_t, \mathbf{Q})\big]}$$

$$\leqslant \frac{1}{\sqrt{T}}\underbrace{\sum_{t=1}^{\lfloor\sqrt{T}\rfloor}\frac{1}{t}\sqrt{\mathbb{E}\big[\mathrm{TV}^2(\hat{\mathbf{Q}}_t, \mathbf{Q})\big]}}_{\leqslant C} + \sum_{t=\lfloor\sqrt{T}\rfloor+1}^{T}\frac{1}{t}\sqrt{\mathbb{E}\big[\mathrm{TV}^2(\hat{\mathbf{Q}}_t, \mathbf{Q})\big]}\,,$$

which converges to 0, as it is the sum of $C/\sqrt{T}$ with a quantity smaller than the remainder of a convergent series.

We recall that for all Borel-measurable functions $f : \mathcal{X} \times \mathcal{S} \to \mathbb{R}^d$ with $\displaystyle\sup_{(x,s)\in\mathcal{X}\times\mathcal{S}}\|f(x,s)\| \leqslant M$,

$$\left\|\int_{\mathcal{X}\times\mathcal{S}} f(x,s)\,\mathrm{d}\mathbf{Q}_1(x,s) - \int_{\mathcal{X}\times\mathcal{S}} f(x,s)\,\mathrm{d}\mathbf{Q}_2(x,s)\right\|$$

$$\leqslant \int_{\mathcal{X}\times\mathcal{S}}\|f(x,s)\|\,\big|g_1(x,s)-g_2(x,s)\big|\,\mathrm{d}\mu(x,s) \leqslant 2M\cdot\mathrm{TV}(\mathbf{Q}_1,\mathbf{Q}_2)\,, \quad (8)$$

where $g_1$ and $g_2$ denote densities of the distributions $\mathbf{Q}_1$ and $\mathbf{Q}_2$ with respect to a common dominating measure $\mu$.

We now move to the proof of Theorem 1, which we restate below. It relies on two lemmas stated below in Section A.1. Unless stated otherwise (namely, for matters related to the estimation of $\mathbf{Q}$), all material is standard and was introduced by Blackwell [3] (see also the more modern expositions by Perchet [25] or Mertens et al. [22]).

**Theorem 1.** *Assume that $\mathcal{C}$ is a closed convex set and that Assumptions 1 (fast enough sequential estimation of $\mathbf{Q}$) and 2 (bounded reward function) are satisfied, then $\mathcal{C}$ is approachable if and only if*

$$\forall(\boldsymbol{q}^{G(x,s)})_{(x,s)\in\mathcal{X}\times\{0,1\}}\ \exists(\boldsymbol{p}^x)_{x\in\mathcal{X}}\quad \text{s.t.}\quad \int_{\mathcal{X}\times\mathcal{S}}\boldsymbol{m}\big(\boldsymbol{p}^x,\boldsymbol{q}^{G(x,s)},x,s\big)\,\mathrm{d}\mathbf{Q}(x,s) \in \mathcal{C}\,. \quad (2)$$

*In this case, the strategy of Eq. (1) achieves the following rates for $L^2$ and almost-sure convergences:*

$$\mathbb{E}\big[d_T^2\big] \leqslant \sqrt{\frac{K}{T}} + 4\|\boldsymbol{m}\|_{\infty,2}\,\overbrace{\frac{1}{T}\sum_{t=1}^{T-1}\sqrt{\mathbb{E}\big[\mathrm{TV}^2(\hat{\mathbf{Q}}_t,\mathbf{Q})\big]}}^{:=\overline{\Delta}_T} \qquad \text{and}$$

$$\mathbb{P}\left(\sup_{t\geqslant T} d_t \geqslant \varepsilon\right) \leqslant \frac{3K}{T\varepsilon^2} + \frac{16\|\boldsymbol{m}\|_{\infty,2}}{\varepsilon^2}\left(\sqrt{\frac{K}{T-1}} + 2\Big(\sup_{t\geqslant T}\overline{\Delta}_t\Big)\Big(\overline{\Delta}_T + \sum_{t\geqslant T}\frac{1}{t}\sqrt{\mathbb{E}\big[\mathrm{TV}^2(\hat{\mathbf{Q}}_t,\mathbf{Q})\big]}\Big)\right)$$

*where $K < +\infty$ denotes the maximal distance to $\mathcal{C}$ of an element of the compact set $\boldsymbol{m}(\mathcal{A},\mathcal{B},\mathcal{X},\mathcal{S})$.*

*Proof of Theorem 1.* **Part I: Necessity.** Assume that the condition in Eq. (2) is not satisfied, then

$$\exists(\boldsymbol{q}_0^{G(x,s)})_{(x,s)\in\mathcal{X}\times\mathcal{S}}\ \forall(\boldsymbol{p}^x)_{x\in\mathcal{X}}\quad \text{s.t.}\quad \int_{\mathcal{X}\times\mathcal{S}}\boldsymbol{m}\big(\boldsymbol{p}^x,\boldsymbol{q}_0^{G(x,s)},x,s\big)\,\mathrm{d}\mathbf{Q}(x,s) \notin \mathcal{C}\,.$$

Since $\mathcal{C}$ is closed and by continuity of the norm, there exists $\alpha > 0$ such that

$$\forall(\boldsymbol{p}^x)_{x\in\mathcal{X}}\qquad \min_{\boldsymbol{v}\in\mathcal{C}}\left\|\boldsymbol{v} - \int_{\mathcal{X}\times\mathcal{S}}\boldsymbol{m}\big(\boldsymbol{p}^x,\boldsymbol{q}_0^{G(x,s)},x,s\big)\,\mathrm{d}\mathbf{Q}(x,s)\right\| \geqslant \alpha\,. \quad (9)$$

Let Nature play using this distribution $(q_0^{G(x,s)})_{(x,s)}$ at each stage $t \geqslant 1$ to draw $b_t$. Given that the sensitive attributes and contexts $(x_t, s_t)$ are drawn i.i.d., the conditional expectation of the reward of the player at round $t \geqslant 1$ based on the history $H_{t-1} = (a_{t'}, b_{t'}, x_{t'}, s_{t'})_{t' \leqslant t-1}$ equals

$$\mathbb{E}\big[\boldsymbol{m}(a_t, b_t, x_t, s_t) \,\big|\, H_{t-1}\big] = \int_{\mathcal{X} \times \mathcal{S}} \boldsymbol{m}\big(\boldsymbol{p}_t^x, \boldsymbol{q}_0^{G(x,s)}, x, s\big)\, \mathrm{d}\mathbf{Q}(x, s)\,.$$

Then, for any strategy of the player, it holds by martingale convergence (e.g., by the Hoeffding-Azuma inequality and the Borel–Cantelli lemma, used for each component of $\boldsymbol{m}$) that

$$\left\| \frac{1}{T} \sum_{t=1}^{T} \boldsymbol{m}(a_t, b_t, x_t, s_t) - \frac{1}{T} \sum_{t=1}^{T} \int_{\mathcal{X} \times \mathcal{S}} \boldsymbol{m}\big(\boldsymbol{p}_t^x, \boldsymbol{q}_0^{G(x,s)}, x, s\big)\, \mathrm{d}\mathbf{Q}(x, s) \right\| \longrightarrow 0 \quad \text{a.s.}$$

Set $\overline{\boldsymbol{p}}_T^x := \dfrac{1}{T} \sum_{t=1}^{T} \boldsymbol{p}_t^x$, then the above implies that

$$\left\| \overline{\boldsymbol{m}}_T - \int_{\mathcal{X} \times \mathcal{S}} \boldsymbol{m}(\overline{\boldsymbol{p}}_T^x, \boldsymbol{q}_0^{G(x,s)}, x, s)\, \mathrm{d}\mathbf{Q}(x, s) \right\| \longrightarrow 0 \quad \text{a.s.} \tag{10}$$

By the triangle inequality for the Euclidean norm, Eqs. (9) and (10) entail that

$$\liminf_{T \to +\infty} d(\overline{\boldsymbol{m}}_T, \mathcal{C}) = \liminf_{T \to +\infty} \big\| \overline{\boldsymbol{m}}_T - \overline{\boldsymbol{c}}_T \big\| = \liminf_{T \to +\infty} \min_{\boldsymbol{v} \in \mathcal{C}} \big\| \boldsymbol{v} - \overline{\boldsymbol{m}}_T \big\| \geqslant \alpha \quad \text{a.s.}$$

That is, Nature prevents the player from approaching $\mathcal{C}$ (and even: Nature approaches the complement of the $\alpha$-neighborhood of $\mathcal{C}$).

Note that in this part we did not use that the target set $\mathcal{C}$ was convex, only that it was a closed set.

**Part II: Sufficiency.** Recall that we denoted by $d_t := \|\overline{\boldsymbol{m}}_t - \overline{\boldsymbol{c}}_t\|_2$ the Euclidean distance of $\overline{\boldsymbol{m}}_t$ to $\mathcal{C}$. Observe that by definition of the projections $\overline{\boldsymbol{c}}_{t+1}$ and $\overline{\boldsymbol{c}}_t$ and by expanding the square norm,

$$\begin{aligned} d_{t+1}^2 &\leqslant \|\overline{\boldsymbol{m}}_{t+1} - \overline{\boldsymbol{c}}_t\|^2 = \left\| \frac{t}{t+1}(\overline{\boldsymbol{m}}_t - \overline{\boldsymbol{c}}_t) + \frac{1}{t+1}(\boldsymbol{m}_{t+1} - \overline{\boldsymbol{c}}_t) \right\|^2 \\ &= \left( \frac{t}{t+1} \right)^2 d_t^2 + \frac{\|\boldsymbol{m}_{t+1} - \overline{\boldsymbol{c}}_t\|^2}{(t+1)^2} + \frac{2t}{(t+1)^2} \langle \overline{\boldsymbol{m}}_t - \overline{\boldsymbol{c}}_t, \boldsymbol{m}_{t+1} - \overline{\boldsymbol{c}}_t \rangle\,. \end{aligned} \tag{11}$$

Moreover, we have, by definition of $(\boldsymbol{p}_{t+1}^x)_{x \in \mathcal{X}}$ in Eq. (1) as the argmin of a maximum,

$$\begin{aligned} &\langle \overline{\boldsymbol{m}}_t - \overline{\boldsymbol{c}}_t, \boldsymbol{m}_{t+1} - \overline{\boldsymbol{c}}_t \rangle \\ &= \left\langle \overline{\boldsymbol{m}}_t - \overline{\boldsymbol{c}}_t, \boldsymbol{m}_{t+1} - \int_{\mathcal{X} \times \mathcal{S}} \boldsymbol{m}\big(\boldsymbol{p}_{t+1}^x, \boldsymbol{q}_{t+1}^{G(x,s)}, x, s\big)\, \mathrm{d}\hat{\mathbf{Q}}_t(x, s) \right\rangle \\ &\quad + \left\langle \overline{\boldsymbol{m}}_t - \overline{\boldsymbol{c}}_t, \int_{\mathcal{X} \times \mathcal{S}} \boldsymbol{m}\big(\boldsymbol{p}_{t+1}^x, \boldsymbol{q}_{t+1}^{G(x,s)}, x, s\big)\, \mathrm{d}\hat{\mathbf{Q}}_t(x, s) - \overline{\boldsymbol{c}}_t \right\rangle \\ &\leqslant \left\langle \overline{\boldsymbol{m}}_t - \overline{\boldsymbol{c}}_t, \boldsymbol{m}_{t+1} - \int_{\mathcal{X} \times \mathcal{S}} \boldsymbol{m}\big(\boldsymbol{p}_{t+1}^x, \boldsymbol{q}_{t+1}^{G(x,s)}, x, s\big)\, \mathrm{d}\hat{\mathbf{Q}}_t(x, s) \right\rangle \\ &\quad + \min_{(\boldsymbol{p}^x)_x} \max_{(\boldsymbol{q}^{G(x,s)})_{G(x,s)}} \left\langle \overline{\boldsymbol{m}}_t - \overline{\boldsymbol{c}}_t, \int_{\mathcal{X} \times \mathcal{S}} \boldsymbol{m}\big(\boldsymbol{p}^x, \boldsymbol{q}^{G(x,s)}, x, s\big)\, \mathrm{d}\hat{\mathbf{Q}}_t(x, s) - \overline{\boldsymbol{c}}_t \right\rangle\,. \end{aligned} \tag{12}$$

Furthermore, the Cauchy-Schwarz inequality, followed by an application of the bound of Eq. (8), indicates that for all $(\boldsymbol{p}^x)_x$ and all $(\boldsymbol{q}^{G(x,s)})_{(x,s)}$,

$$\begin{aligned} &\left| \left\langle \overline{\boldsymbol{m}}_t - \overline{\boldsymbol{c}}_t, \int_{\mathcal{X} \times \mathcal{S}} \boldsymbol{m}\big(\boldsymbol{p}^x, \boldsymbol{q}^{G(x,s)}, x, s\big)\, \mathrm{d}\hat{\mathbf{Q}}_t(x, s) - \int_{\mathcal{X} \times \mathcal{S}} \boldsymbol{m}\big(\boldsymbol{p}^x, \boldsymbol{q}^{G(x,s)}, x, s\big)\, \mathrm{d}\mathbf{Q}(x, s) \right\rangle \right| \\ &\leqslant d_t \cdot \left\| \int_{\mathcal{X} \times \mathcal{S}} \boldsymbol{m}\big(\boldsymbol{p}^x, \boldsymbol{q}^{G(x,s)}, x, s\big)\, \mathrm{d}\hat{\mathbf{Q}}_t(x, s) - \int_{\mathcal{X} \times \mathcal{S}} \boldsymbol{m}\big(\boldsymbol{p}^x, \boldsymbol{q}^{G(x,s)}, x, s\big)\, \mathrm{d}\mathbf{Q}(x, s) \right\| \\ &\leqslant 2 d_t \cdot \mathrm{TV}(\hat{\mathbf{Q}}_t, \mathbf{Q}) \cdot \|\boldsymbol{m}\|_{\infty, 2}\,. \end{aligned}$$

Hence, using twice this bound in Eq. (12) and introducing

$$Z_{t+1} := \left\langle \overline{\boldsymbol{m}}_t - \overline{\boldsymbol{c}}_t, \boldsymbol{m}_{t+1} - \int_{\mathcal{X} \times \mathcal{S}} \boldsymbol{m}\big(\boldsymbol{p}_{t+1}^x, \boldsymbol{q}_{t+1}^{G(x,s)}, x, s\big) \, \mathrm{d}\mathbf{Q}(x,s) \right\rangle, \tag{13}$$

we obtain

$$\langle \overline{\boldsymbol{m}}_t - \overline{\boldsymbol{c}}_t, \boldsymbol{m}_{t+1} - \overline{\boldsymbol{c}}_t \rangle \leqslant Z_{t+1} + 4 d_t \cdot \mathrm{TV}(\hat{\mathbf{Q}}_t, \mathbf{Q}) \cdot \|\boldsymbol{m}\|_{\infty,2}$$
$$+ \min_{(\boldsymbol{p}^x)_x} \max_{(\boldsymbol{q}^{G(x,s)})_{G(x,s)}} \left\langle \overline{\boldsymbol{m}}_t - \overline{\boldsymbol{c}}_t, \int_{\mathcal{X} \times \mathcal{S}} \boldsymbol{m}\big(\boldsymbol{p}^x, \boldsymbol{q}^{G(x,s)}, x, s\big) \, \mathrm{d}\mathbf{Q}(x,s) - \overline{\boldsymbol{c}}_t \right\rangle. \tag{14}$$

We recall that the Euclidean projection $\boldsymbol{c}$ of a vector $\boldsymbol{n}$ onto a closed convex set $\mathcal{C} \subset \mathbb{R}^d$ satisfies:

$$\forall \boldsymbol{c}' \in \mathcal{C}, \qquad \langle \boldsymbol{n} - \boldsymbol{c}, \boldsymbol{c}' - \boldsymbol{c} \rangle \leqslant 0.$$

Thus, thanks to von Neumann's minmax theorem (for the equality) and the Blackwell's condition in Eq. (2) together with the above-recalled property of the projection,

$$\min_{(\boldsymbol{p}^x)_x} \max_{(\boldsymbol{q}^{G(x,s)})_{G(x,s)}} \left\langle \overline{\boldsymbol{m}}_t - \overline{\boldsymbol{c}}_t, \int_{\mathcal{X} \times \mathcal{S}} \boldsymbol{m}\big(\boldsymbol{p}^x, \boldsymbol{q}^{G(x,s)}, x, s\big) \, \mathrm{d}\mathbf{Q}(x,s) - \overline{\boldsymbol{c}}_t \right\rangle$$
$$= \max_{(\boldsymbol{q}^{G(x,s)})_{G(x,s)}} \min_{(\boldsymbol{p}^x)_x} \left\langle \overline{\boldsymbol{m}}_t - \overline{\boldsymbol{c}}_t, \int_{\mathcal{X} \times \mathcal{S}} \boldsymbol{m}\big(\boldsymbol{p}^x, \boldsymbol{q}^{G(x,s)}, x, s\big) \, \mathrm{d}\mathbf{Q}(x,s) - \overline{\boldsymbol{c}}_t \right\rangle \leqslant 0. \tag{15}$$

Hence, combining Eqs. (14) and (15) with Eq. (11), and bounding $\|\boldsymbol{m}_{t+1} - \overline{\boldsymbol{c}}_t\|^2$ by $K$ (by definition of $K$), we have obtained so far

$$d_{t+1}^2 \leqslant \left(\frac{t}{t+1}\right)^2 d_t^2 + \frac{K}{(t+1)^2} + \frac{2t}{(t+1)^2} \left(Z_{t+1} + 4 d_t \cdot \mathrm{TV}(\hat{\mathbf{Q}}_t, \mathbf{Q}) \cdot \|\boldsymbol{m}\|_{\infty,2}\right). \tag{16}$$

The $4 d_t \cdot \mathrm{TV}(\hat{\mathbf{Q}}_t, \mathbf{Q}) \cdot \|\boldsymbol{m}\|_{\infty,2}$ is the sole difference to the standard proof of approachability. We deal with it by adapting the conclusions of the original proof.

Before we do so, we note that the $Z_{t+1}$ introduced in Eq. (13) form a martingale difference sequence with respect to the history $H_t$: indeed, $\overline{\boldsymbol{m}}_t$ and $\overline{\boldsymbol{c}}_t$ are $H_t$–measurable and so are the $(\boldsymbol{p}_{t+1}^x)_x$ and the $(\boldsymbol{q}_{t+1}^{G(x,s)})_{x,s}$; since in addition $(x_{t+1}, s_{t+1})$ is drawn independently from everything according to $\mathbf{Q}$ and $a_{t+1}$ and $b_{t+1}$ are drawn independently at random according to $\boldsymbol{p}_{t+1}^{x_t}$ and $\boldsymbol{q}^{G(x_t, s_t)}$, we have

$$\mathbb{E}[\boldsymbol{m}_{t+1} \,|\, H_t] = \mathbb{E}\big[\boldsymbol{m}(a_{t+1}, b_{t+1}, x_{t+1}, s_{t+1}) \,|\, H_t\big] = \int_{\mathcal{X} \times \mathcal{S}} \boldsymbol{m}\big(\boldsymbol{p}_{t+1}^x, \boldsymbol{q}_{t+1}^{G(x,s)}, x, s\big) \, \mathrm{d}\mathbf{Q}(x,s),$$

so that $\mathbb{E}[Z_{t+1} \,|\, H_t] = 0$.

**Part II: Sufficiency—convergence in $L^2$.** In particular, taking expectations in Eq. (16) and applying the tower rule (for the first inequality) and applying the Cauchy-Schwarz inequality (for the second inequality), we have

$$\mathbb{E}\big[d_{t+1}^2\big] \leqslant \left(\frac{t}{t+1}\right)^2 \mathbb{E}\big[d_t^2\big] + \frac{K}{(t+1)^2} + \frac{8t\|\boldsymbol{m}\|_{\infty,2}}{(t+1)^2} \mathbb{E}\big[d_t \cdot \mathrm{TV}(\hat{\mathbf{Q}}_t, \mathbf{Q})\big]$$
$$\leqslant \left(\frac{t}{t+1}\right)^2 \mathbb{E}\big[d_t^2\big] + \frac{K}{(t+1)^2} + \frac{8t\|\boldsymbol{m}\|_{\infty,2}}{(t+1)^2} \sqrt{\mathbb{E}\big[d_t^2\big]} \sqrt{\mathbb{E}\big[\mathrm{TV}^2(\hat{\mathbf{Q}}_t, \mathbf{Q})\big]}. \tag{17}$$

Applying Lemma 1 below, we get

$$\sqrt{\mathbb{E}\big[d_T^2\big]} \leqslant B_T := \sqrt{\frac{K}{T}} + 4\|\boldsymbol{m}\|_{\infty,2} \underbrace{\frac{1}{T} \sum_{t=1}^{T-1} \sqrt{\mathbb{E}\big[\mathrm{TV}^2(\hat{\mathbf{Q}}_t, \mathbf{Q})\big]}}_{:=\overline{\Delta}_T}. \tag{18}$$

By (a consequence of) Assumption 1, the second term in the right-hand side converges to zero, and we obtain convergence in $L^2$.

**Part II: Sufficiency—almost-sure convergence.** We define

$$S_T := d_T^2 + \mathbb{E}\left[\sum_{t \geqslant T} \left(\frac{K}{(t+1)^2} + \frac{8t\|\boldsymbol{m}\|_{\infty,2}}{(t+1)^2}\, d_t \cdot \mathrm{TV}(\hat{\mathbf{Q}}_t, \mathbf{Q})\right) \,\Big|\, H_T\right],$$

and note that $(S_T)_{T\geqslant 1}$ is a non-negative super-martingale with respect to the filtration induced by $(H_T)_{T\geqslant 1}$; indeed, the recursion of Eq. (16) entails, together with $\left(t/(t+1)\right)^2 \leqslant 1$ and $\mathbb{E}[Z_{T+1}\,|\,H_T] = 0$:

$$\mathbb{E}[S_{T+1} \mid H_T] = \mathbb{E}\big[d_{T+1}^2 \mid H_T\big] + \mathbb{E}\left[\sum_{t \geqslant T+1} \left(\frac{K}{(t+1)^2} + \frac{8t\|\boldsymbol{m}\|_{\infty,2}}{(t+1)^2}\, d_t \cdot \mathrm{TV}(\hat{\mathbf{Q}}_t, \mathbf{Q})\right) \,\Big|\, H_T\right]$$

$$\leqslant d_T^2 + \mathbb{E}\left[\sum_{t \geqslant T} \left(\frac{K}{(t+1)^2} + \frac{8t\|\boldsymbol{m}\|_{\infty,2}}{(t+1)^2}\, d_t \cdot \mathrm{TV}(\hat{\mathbf{Q}}_t, \mathbf{Q})\right) \,\Big|\, H_T\right] = S_T\,.$$

We may thus use $d_T^2 \leqslant S_T$ and apply Doob's maximal inequality for non-negative super-martingales (Lemma 2):

$$\mathbb{P}\left(\sup_{T' \geqslant T} d_{T'} \geqslant \varepsilon\right) = \mathbb{P}\left(\sup_{T' \geqslant T} d_{T'}^2 \geqslant \varepsilon^2\right) \leqslant \mathbb{P}\left(\sup_{T' \geqslant T} S_{T'} \geqslant \varepsilon^2\right) \leqslant \frac{\mathbb{E}[S_T]}{\varepsilon^2}\,.$$

The proof is concluded by upper bounding $\mathbb{E}[S_T]$. The tower rule, the Cauchy-Schwarz inequality, and the bound $t/(t+1)^2 \leqslant 1/(t+1) \leqslant 1/t$ yield

$$\mathbb{E}[S_T] \leqslant \mathbb{E}\big[d_T^2\big] + \sum_{t \geqslant T} \frac{K}{(t+1)^2} + 8\|\boldsymbol{m}\|_{\infty,2} \sum_{t \geqslant T} \sqrt{\mathbb{E}\big[d_t^2\big]}\, \frac{\sqrt{\mathbb{E}\big[\mathrm{TV}^2(\hat{\mathbf{Q}}_t, \mathbf{Q})\big]}}{t}\,.$$

We substitute the bound from Eq. (18), keeping in mind that the total variation distance is always smaller than 1:

$$\mathbb{E}[S_T] \leqslant \mathbb{E}\big[d_T^2\big] + \frac{K}{T} + 8\|\boldsymbol{m}\|_{\infty,2} \sum_{t \geqslant T} \frac{1}{t}\sqrt{\frac{K}{t}} + 32\|\boldsymbol{m}\|_{\infty,2}^2 \sum_{t \geqslant T} \overline{\Delta}_t\, \frac{\sqrt{\mathbb{E}\big[\mathrm{TV}^2(\hat{\mathbf{Q}}_t, \mathbf{Q})\big]}}{t}\,.$$

Eq. (18) also implies, together with $(a+b)^2 \leqslant 2a^2 + 2b^2$, that $\mathbb{E}\big[d_T^2\big] \leqslant 2K/T + 32\|\boldsymbol{m}\|_{\infty,2}^2(\overline{\Delta}_T)^2$. All in all, we get the final bound

$$\mathbb{E}[S_T] \leqslant \frac{3K}{T} + \frac{16\|\boldsymbol{m}\|_{\infty,2}\sqrt{K}}{\sqrt{T-1}} + 32\|\boldsymbol{m}\|_{\infty,2}^2 \left(\sup_{t \geqslant T+1} \overline{\Delta}_t\right)\left(\overline{\Delta}_T + \sum_{t \geqslant T} \frac{1}{t}\sqrt{\mathbb{E}\big[\mathrm{TV}^2(\hat{\mathbf{Q}}_t, \mathbf{Q})\big]}\right).$$

$\square$

## A.1 Two lemmas used in the proof of Theorem 1

The following lemma is an ad-hoc and new, but elementary, tool to deal with the additional term appearing in Eq. (17) compared to the original proof of approachability.

**Lemma 1.** *Let $t^* \geqslant 0$, and consider two non-negative sequences $(d_t)_{t\geqslant t^*}$ and $(\delta_t)_{t\geqslant t^*}$ fulfilling, for $t \geqslant t^*$, the recursive inequality*

$$d_{t+1}^2 \leqslant \left(\frac{t}{t+1}\right)^2 d_t^2 + \frac{K}{(t+1)^2} + \frac{2t}{(t+1)^2}\,\delta_t\, d_t\,. \tag{19}$$

*Then, for all $t \geqslant t^* + 1$,*

$$d_t \leqslant \frac{\sqrt{K(t-t^*)}}{t} + \frac{1}{t}\sum_{t'=t^*}^{t-1} \delta_{t'} + \frac{t^* d_{t^*}}{t}\,.$$

*In particular, if $(d_t)_{t\geqslant 1}$ and $(\delta_t)_{t\geqslant 1}$ are two non-negative sequences fulfilling the recursive inequality (19) for $t \geqslant 1$, and if $d_1 \leqslant \sqrt{K}$, then, for all $t \geqslant 1$,*

$$d_t \leqslant \sqrt{\frac{K}{t}} + \frac{1}{t}\sum_{t'=1}^{t-1} \delta_{t'}\,.$$

*Proof.* Second part of the lemma. Let us first check that the second part of the lemma follows from the first part. Setting $d_0 = \delta_0 = 0$, the sequences $(d_t)_{t \geqslant 0}$ and $(\delta_t)_{t \geqslant 0}$ fulfill Eq. (19) for $t \geqslant t^* = 0$, hence

$$d_t \leqslant \sqrt{\frac{K}{t}} + \frac{1}{t} \sum_{t'=0}^{t-1} \delta_{t'} = \sqrt{\frac{K}{t}} + \frac{1}{t} \sum_{t'=1}^{t-1} \delta_{t'} \,,$$

where the equality in the right-hand side comes from $\delta_0 = 0$.

First part of the lemma. Set $U_t = t \, d_t$ and $\Delta_t^* = \delta_{t^*} + \ldots + \delta_t$ with the convention that $\Delta_t^* = 0$ for all $t < t^*$. It is equivalent to prove that for all $t \geqslant t^* + 1$, we have

$$U_t \leqslant \sqrt{K(t - t^*)} + \Delta_{t-1}^* + U_{t^*} \,. \tag{20}$$

We observe that Eq. (20) trivially holds for $t = t^*$. Assume that Eq. (20) holds for $t \geqslant t^*$. By assumption, we have $U_{t+1}^2 \leqslant U_t^2 + K + 2\delta_t U_t \leqslant (U_t + \delta_t)^2 + K$. Substituting Eq. (20) together with the fact that $U_t \geqslant 0$ and $\delta_t \geqslant 0$, we get

$$\begin{aligned}
U_{t+1}^2 &\leqslant (U_t + \delta_t)^2 + K \leqslant \left( \sqrt{K(t - t^*)} + \Delta_t^* + U_{t^*} \right)^2 + K \\
&= K(t + 1 - t^*) + (\Delta_t^* + U_{t^*})^2 + 2\sqrt{K(t - t^*)} \left( \Delta_t + U_{t^*} \right) \\
&\leqslant \left( \sqrt{K(t + 1 - t^*)} + \Delta_t^* + U_{t^*} \right)^2.
\end{aligned}$$

We have proved that $U_{t+1} \leqslant \sqrt{K(t + 1 - t^*)} + \Delta_t^* + U_{t^*}$, and we conclude by induction. $\qquad\square$

Two maximal inequalities for martingales are called Doob's inequality. We use the less famous one, for non-negative super-martingales.

**Lemma 2** (One of Doob's maximal inequalities)**.** *Let $(S_n)_{n \geqslant 1}$ be a non-negative super-martingale, then*

$$\mathbb{P}\left( \sup_{m \geqslant n} S_m \geqslant \eta \right) \leqslant \frac{\mathbb{E}[S_n]}{\eta} \,.$$

# B   Proofs for Section 4

We first detail the two counter-examples alluded at in the proof of Proposition 1, relative to Example 2 on group-wise no-regret. We then discuss Example 3 on vanilla no-regret under the equalized average payoffs constraint.

## B.1   Counter-examples for group-wise no-regret

*First counter-example.*   We take $\mathcal{S} = \mathcal{A} = \mathcal{B} = \{0, 1\}$ and let $\mathcal{X}$ be an arbitrary finite set. The monitoring is assumed to be $G(x, s) = x$. Finally, we consider the specific payoff function

$$\forall (a, b, x) \in \mathcal{A} \times \mathcal{B} \times \mathcal{X}, \qquad r(a, b, x, 0) = a^2 \quad \text{and} \quad r(a, b, x, 1) = (a-1)^2 .$$

The integral conditions in Eq. (4) read: for all $a' \in \{0, 1\}$,

$$\int_{\mathcal{X}} \sum_{a \in \{0,1\}} \boldsymbol{p}^x(a) \, a^2 \, \mathrm{d}\mathbf{Q}^0(x) \geqslant (a')^2 \quad \text{and} \quad \int_{\mathcal{X}} \sum_{a \in \{0,1\}} \boldsymbol{p}^x(a) \, (a-1)^2 \, \mathrm{d}\mathbf{Q}^1(x) \geqslant (a'-1)^2 ,$$

or equivalently, simply

$$\int_{\mathrm{supp}(\mathbf{Q}^0)} \boldsymbol{p}^x(1) \, \mathrm{d}\mathbf{Q}^0(x) \geqslant 1 \quad \text{and} \quad \int_{\mathrm{supp}(\mathbf{Q}^1)} \boldsymbol{p}^x(0) \, \mathrm{d}\mathbf{Q}^1(x) \geqslant 1 .$$

As $\boldsymbol{p}^x(1) \in [0, 1]$, the fact that the first integral above is larger than 1 entails that $\boldsymbol{p}^x(1) = 1$ on $\mathrm{supp}(\mathbf{Q}^0)$. Similarly, $\boldsymbol{p}^x(0) = 1$ on $\mathrm{supp}(\mathbf{Q}^1)$. As we also have $\boldsymbol{p}^x(0) + \boldsymbol{p}^x(1) = 1$ for all $x \in \mathcal{X}$, we see that the condition in Eq. (4) cannot hold as soon as $\mathrm{supp}(\mathbf{Q}^0) \cap \mathrm{supp}(\mathbf{Q}^1) \neq \emptyset$.

*Second counter-example.*   Again, we take $\mathcal{S} = \mathcal{A} = \mathcal{B} = \{0, 1\}$ and let $\mathcal{X}$ be an arbitrary finite set but assume this time that Nature's monitoring is $G(x, s) = (x, s)$. Another difference is that we consider a payoff function not depending on $s$:

$$\forall (a, b, x, s) \in \mathcal{A} \times \mathcal{B} \times \mathcal{X} \times \mathcal{S}, \qquad r(a, b, x, s) = \mathbb{I}\{a = b\} = 1 - (a-b)^2 .$$

Nature picks the following difficult family of distributions: $\boldsymbol{q}^{(x,0)} = (1, 0)^{\top}$ and $\boldsymbol{q}^{(x,1)} = (0, 1)^{\top}$ for all $x \in \mathcal{X}$, so that $\boldsymbol{q}^{(x,s)}(b) = 1$ if and only if $b = s$. The integral conditions in Eq. (4) therefore read: for all $s \in \{0, 1\}$,

$$\min_{a' \in \{0,1\}} \int_{\mathcal{X}} \sum_{a \in \{0,1\}} \boldsymbol{p}^x(a) \big( r(a, s, x, s) - r(a', s, x, s) \big) \, \mathrm{d}\mathbf{Q}^s(x) = \int_{\mathcal{X}} \boldsymbol{p}^x(s) \, \mathrm{d}\mathbf{Q}^s(x) - 1 \geqslant 0 .$$

From here we conclude similarly to the previous counter-example.

## B.2   Vanilla no-regret under the equalized average payoffs constraint

Blum et al. [4, Section 4] study online regret minimization under a constraint of equal average payoffs, that is, they discuss the $(\boldsymbol{m}_{\mathrm{reg}}, \boldsymbol{m}_{\mathrm{eq\text{-}pay}})$–approachability of $\mathcal{C}_{\mathrm{reg}} \times \mathcal{C}_{\mathrm{eq\text{-}pay}}$, with the notation of Section 2.

Their setting is different from the setting considered in this article, as the latter relies on the no-regret based on a fixed base payoff function $r$, while the former considers prediction with expert advice, that may be assimilated to an adversarially chosen sequence $(r_t)$ of payoff functions.

Yet, we mimic the spirit of their results, which is two-fold.

First, we show an impossibility result for the simultaneous satisfaction of the vanilla no-regret objective and the constraint of equal average payoffs, i.e., for the $(\boldsymbol{m}_{\mathrm{reg}}, \boldsymbol{m}_{\mathrm{eq\text{-}pay}})$–approachability of $\mathcal{C}_{\mathrm{reg}} \times \mathcal{C}_{\mathrm{eq\text{-}pay}}$. We do so for an example of binary online classification. This corresponds to Theorem 4 of Blum et al. [4, Section 4].

Second, we provide a positive result for the mentioned approachability problem, in the case of a Player aware of the sensitive contexts, i.e., following Remark 1, when the Player accesses the contexts $x'_t = (x_t, s_t)$. This corresponds to Theorem 3 of Blum et al. [4, Section 4].

Before we do so, we first instantiate the approachability condition of Eq. (2) with the vector payoff function $\boldsymbol{m} = (\boldsymbol{m}_{\text{reg}}, \boldsymbol{m}_{\text{eq-pay}})$ and the target set $\mathcal{C} = \mathcal{C}_{\text{reg}} \times \mathcal{C}_{\text{eq-pay}}$; it reads: $\forall (\boldsymbol{q}^{G(x,s)})_{(x,s)} \; \exists (\boldsymbol{p}^x)_x$ such that

$$\int_{\mathcal{X} \times \{0,1\}} r\big(\boldsymbol{p}^x, \boldsymbol{q}^{G(x,s)}, x, s\big) \, \mathrm{d}\mathbf{Q}(x,s) \geqslant \max_{a' \in \mathcal{A}} \int_{\mathcal{X} \times \{0,1\}} r\big(a', \boldsymbol{q}^{G(x,s)}, x, s\big) \, \mathrm{d}\mathbf{Q}(x,s) \qquad (21)$$

$$\text{and} \quad \left| \int_{\mathcal{X}} r\big(\boldsymbol{p}^x, \boldsymbol{q}^{G(x,0)}, x, 0\big) \, \mathrm{d}\mathbf{Q}^0(x) - \int_{\mathcal{X}} r\big((\boldsymbol{p}^x, \boldsymbol{q}^{G(x,1)}, x, 1\big) \, \mathrm{d}\mathbf{Q}^1(x) \right| \leqslant \varepsilon. \qquad (22)$$

Second, we also introduce some additional notation.

**Additional notation and reminder on total variation distance.** Recall that we denoted by $\mathbf{Q}^0$ and $\mathbf{Q}^1$ the two marginals of $\mathbf{Q}$ on $\mathcal{X}$. We fix some measure $\mu$ which dominates both $\mathbf{Q}^0$ and $\mathbf{Q}^1$, e.g., $\mu = \mathbf{Q}^0 + \mathbf{Q}^1$, and denote by $g_0$ and $g_1$ densities of $\mathbf{Q}^0$ and $\mathbf{Q}^1$ with respect to $\mu$. We introduce the following three sets (defined up to $\mu$–neglectable events):

$$\mathcal{X}_0 = \big\{ x \in \mathcal{X} \; : \; g_0(x) > g_1(x) \big\},$$
$$\mathcal{X}_1 = \big\{ x \in \mathcal{X} \; : \; g_1(x) > g_0(x) \big\},$$
$$\mathcal{X}_= = \big\{ x \in \mathcal{X} \; : \; g_1(x) = g_0(x) \big\}.$$

Using the above defined sets and densities, we remind that the total variation distance between $\mathbf{Q}^0$ and $\mathbf{Q}^1$ can be expressed in the following equivalent ways (see, e.g., Devroye [8] or Tsybakov [28, Lemma 2.1]):

$$\text{TV}(\mathbf{Q}^0, \mathbf{Q}^1) = \frac{1}{2} \int_{\mathcal{X}} \big| g_0(x) - g_1(x) \big| \, \mathrm{d}\mu(x)$$
$$= \int_{\mathcal{X}_1} \big( g_1(x) - g_0(x) \big) \, \mathrm{d}\mu(x) = \int_{\mathcal{X}_0} \big( g_0(x) - g_1(x) \big) \, \mathrm{d}\mu(x)$$
$$= 1 - \int_{\mathcal{X}} \min\big\{ g_0(x), \, g_1(x) \big\} \, \mathrm{d}\mu(x).$$

We may now describe the impossibility example.

**Impossibility example for online classification.** Binary classification corresponds to the sets of actions $\mathcal{A} = \mathcal{B} = \{0,1\}$ and to the payoff function $r(a, b, x, s) = \mathbb{I}\{a = b\}$. In particular, for all distributions $\boldsymbol{q}$ and $\boldsymbol{q}$, for all contexts $(x,s)$,

$$r(\boldsymbol{p}, \boldsymbol{q}, x, s) = \boldsymbol{p}(0) \, \boldsymbol{q}(0) + \boldsymbol{p}(1) \, \boldsymbol{q}(1) := \langle \boldsymbol{p}, \, \boldsymbol{q} \rangle.$$

We focus our attention on the monitoring $G(x,s) = x$, which gives less freedom to Nature. Our impossibility result holds in particular in the case of the more complete monitoring $G(x,s) = (x,s)$.

We will have Nature pick distributions $(\boldsymbol{q}^x)_{x \in \mathcal{X}}$ such that $\boldsymbol{q}^x(0) > 1/2$ for all $x \in \mathcal{X}$; the maximum in the right-hand side of Eq. (21) is then achieved for $a' = 0$. Because of this and with the notion introduced, the regret criterion of Eq. (21) may be rewritten as

$$\int_{\mathcal{X} \times \{0,1\}} \underbrace{\big( \langle \boldsymbol{p}^x, \, \boldsymbol{q}^x \rangle - \boldsymbol{q}^x(0) \big)}_{\leqslant 0} \, \mathrm{d}\mathbf{Q}(x,s) \geqslant 0.$$

The inequality $\langle \boldsymbol{p}^x, \, \boldsymbol{q}^x \rangle - \boldsymbol{q}^x(0) \leqslant 0$ holds because $\boldsymbol{q}^x(0) > \boldsymbol{q}^x(1)$ by the constraint $\boldsymbol{q}^x(0) > 1/2$; this inequality is strict unless $\boldsymbol{p}^x(1) = 0$. Therefore, the regret constraint imposes $\langle \boldsymbol{p}^x, \, \boldsymbol{q}^x \rangle = \boldsymbol{q}^x(0)$ and $\boldsymbol{p}^x(1) = 0$ on the support of $\mathbf{Q}$ (which is the union of the supports of $\mathbf{Q}^0$ and $\mathbf{Q}^1$).

The constraint of equal average payoffs relies on the following difference, which we rewrite based on the equality just proved:

$$\int_{\mathcal{X}} \langle \boldsymbol{p}^x, \, \boldsymbol{q}^x \rangle \, \mathrm{d}\mathbf{Q}^0(x) - \int_{\mathcal{X}} \langle \boldsymbol{p}^x, \, \boldsymbol{q}^x \rangle \, \mathrm{d}\mathbf{Q}^1(x) = \int_{\mathcal{X}} \boldsymbol{q}^x(0) \, \mathrm{d}\mathbf{Q}^0(x) - \int_{\mathcal{X}} \boldsymbol{q}^x(0) \, \mathrm{d}\mathbf{Q}^1(x).$$

We let Nature pick the distributions $(\boldsymbol{q}^x)$ defined by

$$\boldsymbol{q}^x(0) = \begin{cases} 1 & \text{for } x \in \text{supp}(\mathbf{Q}^0), \\ 1/2 + \varepsilon & \text{for } x \in \text{supp}(\mathbf{Q}^1). \end{cases}$$

We also replace $\mathrm{d}\mathbf{Q}^0$ and $\mathrm{d}\mathbf{Q}^1$ by $g_0\,\mathrm{d}\mu$ and $g_1\,\mathrm{d}\mu$, respectively. The difference in average payoffs thus rewrites, given the various expressions of the total variation distance recalled above:

$$
\int_{\mathcal{X}} \boldsymbol{q}^x(0)\,\mathrm{d}\mathbf{Q}^0(x) - \int_{\mathcal{X}} \boldsymbol{q}^x(0)\,\mathrm{d}\mathbf{Q}^1(x) = \int_{\mathcal{X}} \boldsymbol{q}^x(0)\,\big(g_0(x) - g_1(x)\big)\,\mathrm{d}\mu(x)
$$

$$
= \underbrace{\int_{\mathcal{X}_0} \big(g_0(x) - g_1(x)\big)\,\mathrm{d}\mu(x)}_{=\,\mathrm{TV}(\mathbf{Q}^0,\mathbf{Q}^1)} + \left(\frac{1}{2} + \varepsilon\right) \underbrace{\int_{\mathcal{X}_1 \cup \mathcal{X}_=} \big(g_0(x) - g_1(x)\big)\,\mathrm{d}\mu(x)}_{=\,-\,\mathrm{TV}(\mathbf{Q}^0,\mathbf{Q}^1)}
$$

$$
= \left(\frac{1}{2} - \varepsilon\right) \mathrm{TV}(\mathbf{Q}^0,\mathbf{Q}^1)\,.
$$

All in all, the equal average payoffs constraint of Eq. (22), and thus, the approachability condition of Eq. (2), hold if and only if

$$
\mathrm{TV}(\mathbf{Q}^0,\mathbf{Q}^1) \leqslant \frac{\varepsilon}{1/2 - \varepsilon}\,,
$$

i.e., if the distributions $\mathbf{Q}^0$ and $\mathbf{Q}^1$ are close enough.

This is typically not the case, and having such a small distance between $\mathbf{Q}^0$ and $\mathbf{Q}^1$ should be considered a degenerate case. The limit case $\mathrm{TV}(\mathbf{Q}^0,\mathbf{Q}^1) = 0$ indeed corresponds to the case when the sensitive attributes $s_t$ are independent of the non-sensitive contexts $x_t$.

**Positive result for a Player aware of the $s_t$ and a fair-in-isolation payoff function $r$.** The positive result will be exhibited in the same spirit as the one of Theorem 3 of Blum et al. [4, Section 4]. This spirit is interesting but somewhat limited, as it relies on a (heavy) fair-in-isolation assumption. The latter indeed indicates that for all sequences of contexts and observations, the average loss achieved by a given expert is the same among sensitive groups. This "for all sequences" requirement is particularly demanding. (A question not answered in Blum et al. [4] is the existence of experts that are fair in isolation, for general reward functions $r_t$ or general loss functions $\ell_t$, and metrics $\mathcal{M}$, using their notation.) See comments after Eq. (23) below for the adaptation of this assumption in our context.

A second ingredient for the positive result is that the Player accesses the sensitive contexts $s_t$. Following Remark 1, this translates into our setting by considering that the Player accesses the contexts $x_t' = (x_t, s_t)$; hence, the distributions picked by the Player will be indexed by $(x, s)$ in this example. Nature's monitoring is $G(x, s) = (x, s)$ as well.

Given $(\boldsymbol{q}^{(x,s)})$, to fulfill the no-regret condition

$$
\int_{\mathcal{X} \times \{0,1\}} r\big(\boldsymbol{p}^{(x,s)}, \boldsymbol{q}^{(x,s)}, x, s\big)\,\mathrm{d}\mathbf{Q}(x, s) \geqslant \max_{a' \in \mathcal{A}} \int_{\mathcal{X} \times \{0,1\}} r\big(a', \boldsymbol{q}^{(x,s)}, x, s\big)\,\mathrm{d}\mathbf{Q}(x, s)\,,
$$

the Player may pick $\boldsymbol{p}^{(x,s)}$ based only on $s$:

$$
\boldsymbol{p}^{(x,s)} = \mathrm{dirac}(a^s)\,, \qquad \text{where} \qquad a^s \in \arg\max_{a \in \mathcal{A}} \int_{\mathcal{X}} r\big(a, \boldsymbol{q}^{(x,s)}, x, s\big)\,\mathrm{d}\mathbf{Q}^s(x)\,.
$$

This corresponds to using separate no-regret algorithms in the construction of Theorem 3 of Blum et al. [4], one algorithm per sensitive context. The no-regret algorithms based on approachability used here actually have a regret converging to zero in the limit (they do not just approach the set of non-negative numbers) and thus share the same "not worse but not better" property with respect to the best action $a$ as the one used in Theorem 3 of Blum et al. [4].

The constraint of equal average payoffs requires that the following difference is smaller than $\varepsilon$ in absolute values:

$$
\int_{\mathcal{X}} r\big(\boldsymbol{p}^{(x,0)}, \boldsymbol{q}^{(x,0)}, x, 0\big)\,\mathrm{d}\mathbf{Q}^0(x) - \int_{\mathcal{X}} r\big(\boldsymbol{p}^{(x,1)}, \boldsymbol{q}^{(x,1)}, x, 1\big)\,\mathrm{d}\mathbf{Q}^1(x)
$$

$$
= \max_{a \in \mathcal{A}} \int_{\mathcal{X}} r\big(a, \boldsymbol{q}^{(x,0)}, x, 0\big)\,\mathrm{d}\mathbf{Q}^0(x) - \max_{a \in \mathcal{A}} \int_{\mathcal{X}} r\big(a, \boldsymbol{q}^{(x,1)}, x, 1\big)\,\mathrm{d}\mathbf{Q}^1(x)\,. \quad (23)
$$

This constraint is automatically taken care of by the fair-in-isolation assumption: its analogue in our context (keeping in mind that the actions $a \in \mathcal{A}$ play here the role of the experts in Blum et al. [4]) is to require that for all distributions $(\boldsymbol{q}^{(x,s)})$ picked by the opponent,

$$\forall a \in \mathcal{A}, \qquad \left| \int_{\mathcal{X}} r\big(a, \boldsymbol{q}^{(x,0)}, x, 0\big) \, \mathrm{d}\mathbf{Q}^0(x) - \int_{\mathcal{X}} r\big(a, \boldsymbol{q}^{(x,1)}, x, 1\big) \, \mathrm{d}\mathbf{Q}^1(x) \right| \leqslant \varepsilon .$$

This basically corresponds to an assumption on the effective range of the payoff function $r$ and is therefore a heavy assumption, of limited interest.

# C Proofs for Section 5

We recalled various expressions of the total variation distance in Section B.2. We keep the notation defined therein. The following inequality will be used repeatedly in our proofs.

**Lemma 3.** *For all Borel-measurable functions* $f : \mathcal{X} \to [0, 1]$,

$$\left| \int_{\mathcal{X}} f \, \mathrm{d}\mathbf{Q}^0 - \int_{\mathcal{X}} f \, \mathrm{d}\mathbf{Q}^1 \right| = \left| \int_{\mathcal{X}} f \, (g_0 - g_1) \, \mathrm{d}\mu \right| \leqslant \mathrm{TV}(\mathbf{Q}^0, \mathbf{Q}^1) \,.$$

*Proof.* The proof heavily relies on the fact that $f$ takes values in $[0, 1]$. Since, by definitions of $\mathcal{X}_0$, $\mathcal{X}_1$, and $\mathcal{X}_=$,

$$\int_{\mathcal{X}} f \, (g_0 - g_1) \, \mathrm{d}\mu = \int_{\mathcal{X}_0} f \, \underbrace{(g_0 - g_1)}_{>0} \, \mathrm{d}\mu + \int_{\mathcal{X}_1} f \, \underbrace{(g_0 - g_1)}_{<0} \, \mathrm{d}\mu \,,$$

we have

$$- \mathrm{TV}(\mathbf{Q}^0, \mathbf{Q}^1) = - \int_{\mathcal{X}_1} (g_1 - g_0) \, \mathrm{d}\mu \leqslant \int_{\mathcal{X}} f \, (g_0 - g_1) \, \mathrm{d}\mu \leqslant \int_{\mathcal{X}_0} (g_0 - g_1) \, \mathrm{d}\mu = \mathrm{TV}(\mathbf{Q}^0, \mathbf{Q}^1) \,,$$

which concludes the proof. $\square$

An application of Lemma 3 is the bound $\mathrm{TV}(\mathbf{Q}^0, \mathbf{Q}^1)$ on the $\psi(\dots)$ quantity of Eq. (5). Indeed,

$$\begin{aligned}
\mathtt{DP} &:= \psi \left( \int_{\mathcal{X} \times \{0,1\}} \boldsymbol{m}_{\mathrm{DP}}(\boldsymbol{p}^x, s) \, \mathrm{d}\mathbf{Q}(x, s) \right) \\
&= \left| \int_{\mathcal{X}} \sum_{k=1}^{N} \boldsymbol{p}^x(k) \, a^{(k)} \, \mathrm{d}\mathbf{Q}^0(x) - \int_{\mathcal{X}} \sum_{k=1}^{N} \boldsymbol{p}^x(k) \, a^{(k)} \, \mathrm{d}\mathbf{Q}^1(x) \right| \\
&= \left| \int_{\mathcal{X}} A(\boldsymbol{p}^x) \, \mathrm{d}\mathbf{Q}^0(x) - \int_{\mathcal{X}} A(\boldsymbol{p}^x) \, \mathrm{d}\mathbf{Q}^1(x) \right| \leqslant \mathrm{TV}(\mathbf{Q}^0, \mathbf{Q}^1) \,,
\end{aligned}$$

where we introduced

$$A(\boldsymbol{p}^x) := \sum_{k=1}^{N} \boldsymbol{p}^x(k) \, a^{(k)} \in [0, 1] \,.$$

Before providing the proofs of Propositions 2 and 3 we introduce some additional short-hand notation. We set

$$t^* := \mathrm{TV}(\mathbf{Q}^0, \mathbf{Q}^1) \,.$$

The objective function of the maxmin problem in (6), relative to group-wise calibration, is denoted by and equals

$$\begin{aligned}
\mathtt{GC} &:= \left\| \int_{\mathcal{X} \times \mathcal{S}} \boldsymbol{m}_{\mathrm{gr\text{-}cal}}(\boldsymbol{p}^x, \boldsymbol{q}^{G(x,s)}) \, \mathrm{d}\mathbf{Q}(x, s) \right\|_1 \\
&= \sum_{k=1}^{N} \left| \int_{\mathcal{X}} \boldsymbol{p}^x(k) \left( a^{(k)} - \boldsymbol{q}^{G(x,0)}(1) \right) g_0(x) \, \mathrm{d}\mu(x) \right| \\
&\quad + \sum_{k=1}^{N} \left| \int_{\mathcal{X}} \boldsymbol{p}^x(k) \left( a^{(k)} - \boldsymbol{q}^{G(x,1)}(1) \right) g_1(x) \, \mathrm{d}\mu(x) \right| \,.
\end{aligned}$$

The problem of Eq. (6) can now be written as

$$\varepsilon^\star(\delta_\tau) = \max_{(\boldsymbol{q}^{G(x,s)})} \, \min_{(\boldsymbol{p}^x)} \, \{ \mathtt{GC} \, : \, \mathtt{DP} \leqslant \tau t^* \} \,. \tag{24}$$

The proof technique for each of Propositions 2 and 3 consists of two steps. First, by setting some convenient family $(\boldsymbol{q}^{G(x,s)})_{(x,s)}$, we obtain a lower bound on $\varepsilon^\star(\delta_\tau)$. Second, by exhibiting some

convenient family $(\boldsymbol{p}^x)_x$, possibly based on the knowledge of $(\boldsymbol{q}^{G(x,s)})_{(x,s)}$, an upper bound on $\varepsilon^\star(\delta_\tau)$ is derived.

The definitions of these families will be based on a rounding operator $p \in [0,1] \mapsto \Pi_{\mathcal{A}} \in \mathcal{A}$, that maps a number $p \in [0,1]$ to the closest element in the grid $\mathcal{A}$. Note that by definition of $\mathcal{A}$ and $\Pi_{\mathcal{A}}$, it holds that $|p - \Pi_{\mathcal{A}}(p)| \leqslant 1/(2N)$ for all $p \in [0,1]$. We are finally in the position of proving Propositions 2 and 3, and start with the former.

*Proof of Proposition 2.* Fix some $\tau \in [0,1]$. Recall that $G(x,s) = (x,s)$, so that families of distributions picked by Nature are "truly" indexed by $(x,s)$.

For the lower bound on $\varepsilon^\star(\delta_\tau)$, we consider, for all $x \in \mathcal{X}$,

$$\boldsymbol{q}^{(x,0)} = \mathrm{dirac}(1) \qquad \text{and} \qquad \boldsymbol{q}^{(x,0)} = \mathrm{dirac}(0)\,.$$

Then, for all choices $(\boldsymbol{p}^x)_x$, using that $\displaystyle\sum_{k=1}^N \boldsymbol{p}^x(k) = 1$ for each $x \in \mathcal{X}$:

$$\begin{aligned}
\mathtt{GC} &= \sum_{k=1}^N \left| \int_{\mathcal{X}} \boldsymbol{p}^x(k)\,\big(a^{(k)} - 1\big)\, g_0(x)\,\mathrm{d}\mu(x) \right| + \sum_{k=1}^N \left| \int_{\mathcal{X}} \boldsymbol{p}^x(k)\, a^{(k)}\, g_1(x)\,\mathrm{d}\mu(x) \right| \\
&= \sum_{k=1}^N \int_{\mathcal{X}} \boldsymbol{p}^x(k)\,\big(1 - a^{(k)}\big)\, g_0(x)\,\mathrm{d}\mu(x) + \sum_{k=1}^N \int_{\mathcal{X}} \boldsymbol{p}^x(k)\, a^{(k)}\, g_1(x)\,\mathrm{d}\mu(x) \\
&= \underbrace{\int_{\mathcal{X}} g_0(x)\,\mathrm{d}\mu(x)}_{=1} + \underbrace{\int_{\mathcal{X}} A(\boldsymbol{p}^x)\,\big(g_1(x) - g_0(x)\big)\,\mathrm{d}\mu(x)}_{\text{absolute value equals } \mathtt{DP}} \geqslant 1 - \mathtt{DP}\,,
\end{aligned}$$

so that the rewriting of Eq. (24) entails $\varepsilon^\star(\delta_\tau) \geqslant 1 - \tau t^*$, as claimed.

To derive an upper bound on $\varepsilon^\star(\delta_\tau)$, we consider, for each $(\boldsymbol{q}^{(x,s)})_{(x,s) \in \mathcal{X} \times \mathcal{S}}$ and each $x \in \mathcal{X}$,

$$\boldsymbol{p}^{\tau,x} = (1 - \tau) \cdot \mathrm{dirac}\big(\Pi_{\mathcal{A}}(1/2)\big) + \tau \cdot \mathrm{dirac}\big(f(x)\big)\,,$$

$$\text{where} \qquad f(x) = \begin{cases} \Pi_{\mathcal{A}}\big(\boldsymbol{q}^{(x,1)}\big)(1) & \text{if } x \in \mathcal{X}_1 \cup \mathcal{X}_=\,; \\ \Pi_{\mathcal{A}}\big(\boldsymbol{q}^{(x,0)}\big)(1) & \text{if } x \in \mathcal{X}_0\,. \end{cases} \qquad (25)$$

Note that for this strategy of the Player, $\mathtt{DP} \leqslant \tau t^*$; indeed, $A(\boldsymbol{p}^{\tau,x}) = (1 - \tau) \cdot \Pi_{\mathcal{A}}(1/2) + \tau \cdot f(x)$, so that

$$\begin{aligned}
\mathtt{DP} = \left| \int_{\mathcal{X}} A(\boldsymbol{p}^{\tau,x})\,\mathrm{d}\mathbf{Q}^0(x) - \int_{\mathcal{X}} A(\boldsymbol{p}^{\tau,x})\,\mathrm{d}\mathbf{Q}^1(x) \right| &= \tau \cdot \left| \int_{\mathcal{X}} f(x)\,\mathrm{d}\mathbf{Q}^0(x) - \int_{\mathcal{X}} f(x)\,\mathrm{d}\mathbf{Q}^1(x) \right| \\
&\leqslant \tau \cdot \mathrm{TV}(\mathbf{Q}^0, \mathbf{Q}^1)\,,
\end{aligned}$$

where we applied Lemma 3 for the final inequality. Moreover, the choice of Eq. (25) ensures that $\mathtt{GC} \leqslant 1 - \tau t^* + 1/N$, as we will prove below. This will lead to $\varepsilon^\star(\delta_\tau) \leqslant 1 - \tau t^* + 1/N$ and will conclude the proof. Indeed,

$$\begin{aligned}
\mathtt{GC} &= \sum_{k=1}^N \left| \int_{\mathcal{X}} \boldsymbol{p}^{\tau,x}(k)\,\big(a^{(k)} - \boldsymbol{q}^{(x,0)}(1)\big)\, g_0(x)\,\mathrm{d}\mu(x) \right| \\
&\quad + \sum_{k=1}^N \left| \int_{\mathcal{X}} \boldsymbol{p}^{\tau,x}(k)\,\big(a^{(k)} - \boldsymbol{q}^{(x,1)}(1)\big)\, g_1(x)\,\mathrm{d}\mu(x) \right| \\
&= (1 - \tau) \left| \int_{\mathcal{X}} \big(\Pi_{\mathcal{A}}(1/2) - \boldsymbol{q}^{(x,0)}(1)\big)\, g_0(x)\,\mathrm{d}\mu(x) \right| \\
&\quad + (1 - \tau) \left| \int_{\mathcal{X}} \big(\Pi_{\mathcal{A}}(1/2) - \boldsymbol{q}^{(x,1)}(1)\big)\, g_1(x)\,\mathrm{d}\mu(x) \right| \\
&\quad + \tau \left| \int_{\mathcal{X}} \big(f(x) - \boldsymbol{q}^{(x,0)}(1)\big)\, g_0(x)\,\mathrm{d}\mu(x) \right| + \tau \left| \int_{\mathcal{X}} \big(f(x) - \boldsymbol{q}^{(x,1)}(1)\big)\, g_1(x)\,\mathrm{d}\mu(x) \right|\,.
\end{aligned}$$

We replace $f(x)$ by its specific values and take care of all rounding operators $\Pi_A$ by adding a $2 \times 1/(2N) = 1/N$ term after application of triangle inequalities:

$$\mathtt{GC} \leqslant \frac{1}{N} + (1 - \tau)\left|\int_{\mathcal{X}} \overbrace{\left(1/2 - \boldsymbol{q}^{(x,0)}(1)\right)}^{\in[-1/2,\,1/2]} g_0(x)\,\mathrm{d}\mu(x)\right| + (1 - \tau)\left|\int_{\mathcal{X}} \overbrace{\left(1/2 - \boldsymbol{q}^{(x,1)}(1)\right)}^{\in[-1/2,\,1/2]} g_1(x)\,\mathrm{d}\mu(x)\right|$$

$$+ \tau\left|\int_{\mathcal{X}_1 \cup \mathcal{X}_=} \overbrace{\left(\boldsymbol{q}^{(x,1)}(1) - \boldsymbol{q}^{(x,0)}(1)\right)}^{\in[-1,1]} g_0(x)\,\mathrm{d}\mu(x)\right| + \tau\left|\int_{\mathcal{X}_0} \overbrace{\left(\boldsymbol{q}^{(x,0)}(1) - \boldsymbol{q}^{(x,1)}(1)\right)}^{\in[-1,1]} g_1(x)\,\mathrm{d}\mu(x)\right|$$

$$\leqslant \frac{1}{N} + \frac{1-\tau}{2} + \frac{1-\tau}{2} + \tau\int_{\mathcal{X}_1 \cup \mathcal{X}_=} g_0(x)\,\mathrm{d}\mu(x) + \tau\int_{\mathcal{X}_0} g_1(x)\,\mathrm{d}\mu(x)$$

$$= 1 - \tau + \tau\underbrace{\int_{\mathcal{X}} \min\{g_0(x), g_1(x)\}\,\mathrm{d}\mu(x)}_{=1-t^*} + \frac{1}{N} = 1 - \tau t^* + \frac{1}{N}\,,$$

where we used one of the expressions of $t^* = \mathrm{TV}(\mathbf{Q}^0, \mathbf{Q}^1)$ in the last equality. $\qquad\square$

*Proof of Proposition 3.* Fix some $\tau \in [0, 1]$. Recall that $G(x, s) = x$, so that families of distributions picked by Nature are only indexed by $x$ and may not depend on $s$.

For the lower bound on $\varepsilon^\star(\delta_\tau)$, we consider $(\boldsymbol{q}^x)_{x \in \mathcal{X}}$ defined as

$$\boldsymbol{q}^x = \begin{cases} \mathrm{dirac}(1) & \text{if } x \in \mathcal{X}_1 \cup \mathcal{X}_= \,; \\ \mathrm{dirac}(0) & \text{if } x \in \mathcal{X}_0\,. \end{cases}$$

Then, for all choices $(\boldsymbol{p}^x)_x$, using the notation $A(\boldsymbol{p}^x)$ and the fact that $\displaystyle\sum_{k=1}^N \boldsymbol{p}^x(k) = 1$ for each $x \in \mathcal{X}$:

$$\mathtt{GC} = \sum_{k=1}^N\left|\int_{\mathcal{X}} \boldsymbol{p}^x(k)\left(a^{(k)} - \boldsymbol{q}^x(1)\right) g_0(x)\,\mathrm{d}\mu(x)\right| + \sum_{k=1}^N\left|\int_{\mathcal{X}} \boldsymbol{p}^x(k)\left(a^{(k)} - \boldsymbol{q}^x(1)\right) g_1(x)\,\mathrm{d}\mu(x)\right|$$

$$= \sum_{k=1}^N\left|\int_{\mathcal{X}_1 \cup \mathcal{X}_=} \boldsymbol{p}^x(k)\left(a^{(k)} - 1\right) g_0(x)\,\mathrm{d}\mu(x) + \int_{\mathcal{X}_0} \boldsymbol{p}^x(k)\, a^{(k)}\, g_0(x)\,\mathrm{d}\mu(x)\right|$$

$$+ \sum_{k=1}^N\left|\int_{\mathcal{X}_1 \cup \mathcal{X}_=} \boldsymbol{p}^x(k)\left(a^{(k)} - 1\right) g_1(x)\,\mathrm{d}\mu(x) + \int_{\mathcal{X}_0} \boldsymbol{p}^x(k)\, a^{(k)}\, g_1(x)\,\mathrm{d}\mu(x)\right|$$

$$\geqslant \left|\int_{\mathcal{X}_1 \cup \mathcal{X}_=} \left(A(\boldsymbol{p}^x) - 1\right) g_0(x)\,\mathrm{d}\mu(x) + \int_{\mathcal{X}_0} A(\boldsymbol{p}^x)\, g_0(x)\,\mathrm{d}\mu(x)\right|$$

$$+ \left|\int_{\mathcal{X}_1 \cup \mathcal{X}_=} \left(A(\boldsymbol{p}^x) - 1\right) g_1(x)\,\mathrm{d}\mu(x) + \int_{\mathcal{X}_0} A(\boldsymbol{p}^x)\, g_1(x)\,\mathrm{d}\mu(x)\right|$$

$$= \left|\int_{\mathcal{X}} A(\boldsymbol{p}^x)\, g_0(x)\,\mathrm{d}\mu(x) - \int_{\mathcal{X}_1 \cup \mathcal{X}_=} g_0(x)\,\mathrm{d}\mu(x)\right|$$

$$+ \left|\int_{\mathcal{X}} A(\boldsymbol{p}^x)\, g_1(x)\,\mathrm{d}\mu(x) - \int_{\mathcal{X}_1 \cup \mathcal{X}_=} g_1(x)\,\mathrm{d}\mu(x)\right|\,,$$

$$\geqslant \left|\underbrace{\int_{\mathcal{X}_1 \cup \mathcal{X}_=} \left(g_1(x) - g_0(x)\right)\mathrm{d}\mu(x)}_{=t^*} - \underbrace{\int_{\mathcal{X}_1 \cup \mathcal{X}_=} A(\boldsymbol{p}^x)\left(g_1(x) - g_0(x)\right)\mathrm{d}\mu(x)}_{\text{absolute value equals } \mathtt{DP}}\right| \geqslant t^* - \mathtt{DP}\,.$$

where all the inequalities follows from the triangle inequality. Eq. (24) entails $\varepsilon^\star(\delta_\tau) \geqslant (1 - \tau)t^*$, as claimed.

To derive an upper bound on $\varepsilon^\star(\delta_\tau)$, we consider, for each $(\boldsymbol{q}^x)_{x \in \mathcal{X}}$ and each $x \in \mathcal{X}$,

$$\boldsymbol{p}^{\tau,x} = (1 - \tau) \cdot \operatorname{dirac}\big(\Pi_{\mathcal{A}}(Q)\big) + \tau \cdot \operatorname{dirac}\Big(\Pi_{\mathcal{A}}\big(\boldsymbol{q}^x(1)\big)\Big)$$

$$\text{where} \qquad Q = \int_{\mathcal{X}} \boldsymbol{q}^u(1) \, g_0(u) \, \mathrm{d}\mu(u) \,. \quad (26)$$

Note that for this strategy of the Player, $\mathrm{DP} \leqslant \tau t^*$; indeed, $A(\boldsymbol{p}^{\tau,x}) = (1-\tau) \cdot \Pi_{\mathcal{A}}(Q) + \tau \cdot \Pi_{\mathcal{A}}\big(\boldsymbol{q}^x(1)\big)$, so that

$$\mathrm{DP} = \left| \int_{\mathcal{X}} A(\boldsymbol{p}^{\tau,x}) \, \mathrm{d}\mathbf{Q}^0(x) - \int_{\mathcal{X}} A(\boldsymbol{p}^{\tau,x}) \, \mathrm{d}\mathbf{Q}^1(x) \right|$$

$$= \tau \cdot \left| \int_{\mathcal{X}} \Pi_{\mathcal{A}}\big(\boldsymbol{q}^x(1)\big) \, \mathrm{d}\mathbf{Q}^0(x) - \int_{\mathcal{X}} \Pi_{\mathcal{A}}\big(\boldsymbol{q}^x(1)\big) \, \mathrm{d}\mathbf{Q}^1(x) \right| \leqslant \tau \cdot \mathrm{TV}(\mathbf{Q}^0, \mathbf{Q}^1) \,,$$

where we applied Lemma 3 for the final inequality. Moreover, the choice of Eq. (26) ensures that $\mathrm{GC} \leqslant (1 - \tau)t^* + 1/N$, as we will prove below. This will lead to $\varepsilon^\star(\delta_\tau) \leqslant (1 - \tau)t^* + 1/N$ and will conclude the proof. Indeed,

$$\mathrm{GC} = \sum_{k=1}^N \left| \int_{\mathcal{X}} \boldsymbol{p}^{\tau,x}(k) \big(a^{(k)} - \boldsymbol{q}^x(1)\big) g_0(x) \, \mathrm{d}\mu(x) \right| + \sum_{k=1}^N \left| \int_{\mathcal{X}} \boldsymbol{p}^{\tau,x}(k) \big(a^{(k)} - \boldsymbol{q}^x(1)\big) g_1(x) \, \mathrm{d}\mu(x) \right|$$

$$= (1 - \tau) \left| \int_{\mathcal{X}} \big(\Pi_{\mathcal{A}}(Q) - \boldsymbol{q}^x(1)\big) g_0(x) \, \mathrm{d}\mu(x) \right| + (1 - \tau) \left| \int_{\mathcal{X}} \big(\Pi_{\mathcal{A}}(Q) - \boldsymbol{q}^x(1)\big) g_1(x) \, \mathrm{d}\mu(x) \right|$$

$$+ \tau \left| \int_{\mathcal{X}} \underbrace{\big(\Pi_{\mathcal{A}}\big(\boldsymbol{q}^x(1)\big) - \boldsymbol{q}^x(1)\big)}_{\leqslant 1/(2N)} g_0(x) \, \mathrm{d}\mu(x) \right| + \tau \left| \int_{\mathcal{X}} \underbrace{\big(\Pi_{\mathcal{A}}\big(\boldsymbol{q}^x(1)\big) - \boldsymbol{q}^x(1)\big)}_{\leqslant 1/(2N)} g_1(x) \, \mathrm{d}\mu(x) \right| \,.$$

Taking into account the rounding errors, i.e., replacing the two occurrences of $\Pi_{\mathcal{A}}(Q)$ by $Q$ by adding twice a $(1 - \tau)/(2N)$ term, we get

$$\mathrm{GC} \leqslant (1 - \tau) \left| \int_{\mathcal{X}} \big(Q - \boldsymbol{q}^x(1)\big) g_0(x) \, \mathrm{d}\mu(x) \right| + (1 - \tau) \left| \int_{\mathcal{X}} \big(Q - \boldsymbol{q}^x(1)\big) g_1(x) \, \mathrm{d}\mu(x) \right| + \frac{1}{N}$$

$$= (1 - \tau) \left| \underbrace{Q - \int_{\mathcal{X}} \boldsymbol{q}^x(1) \, g_0(x) \, \mathrm{d}\mu(x)}_{=0} \right| + (1 - \tau) \left| Q - \int_{\mathcal{X}} \boldsymbol{q}^x(1) \, g_1(x) \, \mathrm{d}\mu(x) \right| + \frac{1}{N}$$

$$= (1 - \tau) \left| \int_{\mathcal{X}} \boldsymbol{q}^x(1) \, g_0(x) \, \mathrm{d}\mu(x) - \int_{\mathcal{X}} \boldsymbol{q}^x(1) \, g_1(x) \, \mathrm{d}\mu(x) \right| + \frac{1}{N} \leqslant (1 - \tau)t^* + 1/N \,,$$

where the last equality holds by definition of $Q$ as an integral and we applied Lemma 3 for the final inequality. $\qquad \square$

# D    Proofs for Section 6

In this section, we go over the results alluded at in Section 6. We first illustrate that Assumption 3 (which indicates that the target set $\mathcal{C}$ should be estimated in some way) is realistic. We do so in Section D.1 by dealing with the most involved situation discussed in this article, namely, the example discussed at the beginning of Section 6. We then provide in Section D.2 a more complete statement of Theorem 2, with convergence rates, and prove it.

## D.1    Assumption 3 is realistic

The beginning of Section 6 explained why and how performing an optimal trade-off between accuracy in group-calibration and unfairness in terms of demographic parity amounts to studying the $(\widetilde{\boldsymbol{m}}_{\text{gr-cal}}, \widetilde{\boldsymbol{m}}_{\text{DP}})$–approachability of $\mathcal{C} = \widetilde{\mathcal{C}}^{\varepsilon}_{\text{gr-cal}} \times \widetilde{\mathcal{C}}^{\delta}_{\text{DP}}$, where $(\widetilde{\boldsymbol{m}}_{\text{gr-cal}}, \widetilde{\boldsymbol{m}}_{\text{DP}})$ is a known vector payoff function and where $\mathcal{C} := \widetilde{\mathcal{C}}^{\varepsilon}_{\text{gr-cal}} \times \widetilde{\mathcal{C}}^{\delta}_{\text{DP}}$ is unknown:

$$\widetilde{\mathcal{C}}^{\varepsilon}_{\text{gr-cal}} = \left\{ (\boldsymbol{v}_0, \boldsymbol{v}_1) \in \mathbb{R}^{2N} \ : \ \frac{\|\boldsymbol{v}_0\|_1}{\gamma_0} + \frac{\|\boldsymbol{v}_1\|_1}{\gamma_1} \leqslant \varepsilon \right\}, \qquad \widetilde{\mathcal{C}}^{\delta}_{\text{DP}} = \left\{ (u, v) \in \mathbb{R}^2 \ : \ \left| \frac{u}{\gamma_0} - \frac{v}{\gamma_1} \right| \leqslant \delta \right\},$$

with $\varepsilon = (1 - \tau) \cdot \text{TV}(\mathbf{Q}^0, \mathbf{Q}^1)$ and $\delta = \tau \cdot \text{TV}(\mathbf{Q}^0, \mathbf{Q}^1)$ for some *known* $\tau \in [0, 1]$ but an unknown $\text{TV}(\mathbf{Q}^0, \mathbf{Q}^1)$, and with unknown probabilities $\gamma_0, \gamma_1$. The parameter $\tau$ controls the desired trade-off between the calibration error and the discrepancy in demographic parity and thus is left as a parameter of user's choice.

We recall that the strategy of the Player proceeds in phases: at each time $T_r := 2^r$ for $r \geqslant 1$, the Player updates the estimate $\hat{\mathcal{C}}_r$ of $\mathcal{C}$. The focus of this section is to provide a sequence of estimates $\hat{\mathcal{C}}_r$ of $\mathcal{C}$ fulfilling Assumption 3. The latter is a key requirement for the existence of an approachability strategy stated in Theorem 2. We must therefore prove that it is a realistic assumption.

**The four requirements of Assumption 3.**    For the convenience of the reader, we restate the various requirements of Assumption 3, giving them nicknames, to be able to refer to them easily in the sequel: for all $r \geqslant 0$, the sets $\hat{\mathcal{C}}_r$

| | |
|---|---|
| (CC) | are convex closed; |
| (Proj-dist) | satisfy $\|\boldsymbol{v} - \text{Proj}_{\hat{\mathcal{C}}_r}(\boldsymbol{v})\| \leqslant B$, for all $\boldsymbol{v} \in \boldsymbol{m}(\mathcal{A}, \mathcal{B}, \mathcal{X}, \{0, 1\})$; |
| (Super-set) | satisfy $\mathbb{P}\big(\mathcal{C} \subset \hat{\mathcal{C}}_r\big) \geqslant 1 - 1/(2T_r)$; |
| (L2-Hausdorff) | satisfy $\max\left\{ \mathbb{E}\big[d(\hat{\mathcal{C}}_r, \mathcal{C})^2\big], \ \mathbb{E}\big[d(\mathcal{C}, \hat{\mathcal{C}}_r)^2\big] \right\} \leqslant \beta_r^2.$ |

The constant $B < +\infty$ is independent of $r$ and the sequence $(\beta_r)_{r \geqslant 0}$ is summable and non-increasing. The vector payoff function $\boldsymbol{m}$ above refers to $(\widetilde{\boldsymbol{m}}_{\text{gr-cal}}, \widetilde{\boldsymbol{m}}_{\text{DP}})$.

(Proj-dist) requires that the distance of a possible vector payoff to sets $\hat{\mathcal{C}}_r$ are uniformly controlled. (Super-set) requires that the $\hat{\mathcal{C}}_r$ are, with high probability, super-sets of $\mathcal{C}$. Finally, (L2-Hausdorff) requires that some $L^2$ criterion of Hausdorff distance between sets is controlled. We will go over each of these requirements but first deepen our reduction scheme.

In the sequel, and as in the main body of the paper, we focus on the case where $\gamma_0 > 0$ and $\gamma_1 > 0$, i.e., there are two effective values for the sensitive contexts.

**But first, a further reduction.**    Recall that the average vector payoff, described in Section 6, is equal to

$$\frac{1}{T} \sum_{t=1}^{T} \big( \widetilde{\boldsymbol{m}}_{\text{gr-cal}}(a_t, b_t, x_t, s_t), \widetilde{\boldsymbol{m}}_{\text{DP}}(a_t, b_t, x_t, s_t) \big),$$

where the first $2N$ components always lie in the interval $[-1, 1]$, while the last two ones lie in the interval $[0, 1]$. Therefore, in the definition of $\widetilde{\mathcal{C}}^{\delta}_{\text{DP}}$, we may restrict our attention to $(u, v) \in [0, 1]^2$ and use rather the alternative definition

$$\widetilde{\mathcal{C}}^{\delta}_{\text{DP}} := \left\{ (u, v) \in [0, 1]^2 \ : \ \left| \frac{u}{\gamma_0} - \frac{v}{\gamma_1} \right| \leqslant \delta \right\} = \left\{ (u, v) \in [0, 1]^2 \ : \ |\gamma_1 u - \gamma_0 v| \leqslant \gamma_0 \gamma_1 \delta \right\}.$$

As for $\widetilde{\mathcal{C}}^{\varepsilon}_{\text{gr-cal}}$, given we are studying a calibration problem, we note that the $\varepsilon$ of interest lie in $[0, 1]$ (with 0 included). Vectors $(\boldsymbol{v}_0, \boldsymbol{v}_1)$ of $\widetilde{\mathcal{C}}^{\varepsilon}_{\text{gr-cal}}$ satisfy in particular that $\|\boldsymbol{v}_0\|_1 + \|\boldsymbol{v}_1\|_1 \leqslant \varepsilon$, which shows that $\widetilde{\mathcal{C}}^{\varepsilon}_{\text{gr-cal}} \subseteq B^{\ell_1}_{\mathbb{R}^{2N}}$, where $B^{\ell_1}_{\mathbb{R}^{2N}} = \left\{\boldsymbol{v} \in \mathbb{R}^{2N} : \|\boldsymbol{v}\|_1 \leqslant 1\right\}$ is the unit $\ell_1$ ball in $\mathbb{R}^{2N}$. Therefore,

$$\widetilde{\mathcal{C}}^{\varepsilon}_{\text{gr-cal}} = \left\{(\boldsymbol{v}_0, \boldsymbol{v}_1) \in B^{\ell_1}_{\mathbb{R}^{2N}} : \frac{\|\boldsymbol{v}_0\|_1}{\gamma_0} + \frac{\|\boldsymbol{v}_1\|_1}{\gamma_1} \leqslant \varepsilon\right\}$$

$$= \left\{(\boldsymbol{v}_0, \boldsymbol{v}_1) \in B^{\ell_1}_{\mathbb{R}^{2N}} : \gamma_1\|\boldsymbol{v}_0\|_1 + \gamma_0\|\boldsymbol{v}_1\|_1 \leqslant \gamma_0\gamma_1\varepsilon\right\}.$$

**Plug-in estimation of $\mathcal{C}$.** We consider estimators $\hat{\gamma}_{0,t}, \hat{\gamma}_{1,t} \in [0, 1]$ of $\gamma_0, \gamma_1$ and an estimator $\widehat{M}_t \in [0, 1]$ of $\text{TV}(\mathbf{Q}^0, \mathbf{Q}^1)$, based on the first $t$ i.i.d. samples from $\mathbf{Q}$, see the end of the section for examples. We substitute them in the definitions of the target sets. We actually perform a careful such substitution by considering possibly data-dependent parameters $\alpha_1(t) \in (0, 1]$ and $\alpha_2(t) \in (0, 1]$, to be specified by the analysis, that will provide the needed upper confidence bounds (i.e., super-set condition). More precisely, we define estimators of $\widetilde{\mathcal{C}}^{\varepsilon}_{\text{gr-cal}}$ and $\widetilde{\mathcal{C}}^{\delta}_{\text{DP}}$ by

$$\widehat{\mathcal{C}}^{\varepsilon}_{\text{gr-cal}}(t) = \left\{(\boldsymbol{v}_0, \boldsymbol{v}_1) \in B^{\ell_1}_{\mathbb{R}^{2N}} : \hat{\gamma}_{1,t}\|\boldsymbol{v}_0\|_1 + \hat{\gamma}_{0,t}\|\boldsymbol{v}_1\|_1 \leqslant \hat{\gamma}_{0,t}\hat{\gamma}_{1,t}\hat{\varepsilon}_t + \alpha_1(t) + 4\alpha_2(t)\right\},$$

$$\widehat{\mathcal{C}}^{\delta}_{\text{DP}}(t) = \left\{(u, v) \in [0, 1] : \left|\hat{\gamma}_{1,t}u - \hat{\gamma}_{0,t}v\right| \leqslant \hat{\gamma}_{0,t}\hat{\gamma}_{1,t}\hat{\delta}_t + \alpha_1(t) + 4\alpha_2(t)\right\},$$

where $\hat{\varepsilon}_t = (1 - \tau)\widehat{M}_t$ and $\hat{\delta}_t = \tau\widehat{M}_t$. We then set $\hat{\mathcal{C}}_r := \widehat{\mathcal{C}}^{\varepsilon}_{\text{gr-cal}}(T_r) \times \widehat{\mathcal{C}}^{\delta}_{\text{DP}}(T_r)$.

**Requirements (CC) and (Proj-dist) hold.** We observe that both $\widehat{\mathcal{C}}^{\varepsilon}_{\text{gr-cal}}(t)$ and $\widehat{\mathcal{C}}^{\delta}_{\text{DP}}(t)$ are convex, closed, and bounded. The boundedness of these sets and the fact that $\boldsymbol{m} = (\widetilde{\boldsymbol{m}}_{\text{gr-cal}}, \widetilde{\boldsymbol{m}}_{\text{DP}})$ is bounded as well ensure the (Proj-dist) property.

**Choice of $\alpha_1(T_r)$ and $\alpha_2(T_r)$, part 1.** We introduce the following sets, indicating that some confidence bounds around the introduced estimators hold, of widths smaller than the introduced parameters $\alpha_1(T_r)$ and $\alpha_2(T_r)$. These sets need only to be considered at times $T_r$, where $r \geqslant 1$:

$$\Omega^{\alpha_1, \alpha_2}_{T_r} := \left\{\left|\widehat{M}_{T_r} - \text{TV}(\mathbf{Q}^0, \mathbf{Q}^1)\right| \leqslant \alpha_1(T_r) \quad \text{and} \quad \forall s \in \{0, 1\}, |\hat{\gamma}_{s,T_r} - \gamma_s| \leqslant \alpha_2(T_r)\right\}.$$

We assume in the sequel that we could pick all $\alpha_1(T_r)$ and $\alpha_2(T_r)$ such that for all $r \geqslant 1$,

$$\mathbb{P}\left(\Omega^{\alpha_1, \alpha_2}_{T_r}\right) \geqslant 1 - \frac{1}{2T_r}, \tag{27}$$

and explain, in the final part of this section, how this can be ensured.

**Requirement (Super-set) holds.** It follows from the assumption above on the probability of $\Omega^{\alpha_1, \alpha_2}_{T_r}$ and from the following lemma.

**Lemma 4.** *On the event $\Omega^{\alpha_1, \alpha_2}_{T_r}$ defined above, it holds that $\widetilde{\mathcal{C}}^{\varepsilon}_{\text{gr-cal}} \subseteq \widehat{\mathcal{C}}^{\varepsilon}_{\text{gr-cal}}(T_r)$ and $\widetilde{\mathcal{C}}^{\delta}_{\text{DP}} \subseteq \widehat{\mathcal{C}}^{\delta}_{\text{DP}}(T_r)$, thus*

$$\mathcal{C} = \left(\widetilde{\mathcal{C}}^{\varepsilon}_{\text{gr-cal}} \times \widetilde{\mathcal{C}}^{\delta}_{\text{DP}}\right) \subseteq \hat{\mathcal{C}}_r = \left(\widehat{\mathcal{C}}^{\varepsilon}_{\text{gr-cal}}(T_r) \times \widehat{\mathcal{C}}^{\delta}_{\text{DP}}(T_r)\right).$$

*Proof.* For brevity, we drop the dependencies in $T_r$ in the notation.

Part I: $\widetilde{\mathcal{C}}^{\varepsilon}_{\text{gr-cal}} \subset \widehat{\mathcal{C}}^{\varepsilon}_{\text{gr-cal}}$. We fix some $(\boldsymbol{v}_0, \boldsymbol{v}_1) \in \widetilde{\mathcal{C}}^{\varepsilon}_{\text{gr-cal}}$. By assumption,

$$\gamma_1\|\boldsymbol{v}_0\|_1 + \gamma_0\|\boldsymbol{v}_1\|_1 \leqslant (1 - \tau)\gamma_0\gamma_1 \cdot \text{TV}(\mathbf{Q}^0, \mathbf{Q}^1). \tag{28}$$

Furthermore, since $\|\boldsymbol{v}_0\|_1 + \|\boldsymbol{v}_1\|_1 \leqslant 1$, it holds on $\Omega^{\alpha_1, \alpha_2}$ that

$$\gamma_1\|\boldsymbol{v}_0\|_1 + \gamma_0\|\boldsymbol{v}_1\|_1 \geqslant \hat{\gamma}_1\|\boldsymbol{v}_0\|_1 + \hat{\gamma}_0\|\boldsymbol{v}_1\|_1 - 2\alpha_2,$$

$$\gamma_0\gamma_1 \cdot \text{TV}(\mathbf{Q}^0, \mathbf{Q}^1) \leqslant \hat{\gamma}_0\gamma_1 \cdot \text{TV}(\mathbf{Q}^0, \mathbf{Q}^1) + \alpha_2 \leqslant \ldots \leqslant \hat{\gamma}_0\hat{\gamma}_1 \cdot \widehat{M} + \alpha_1 + 2\alpha_2. \tag{29}$$

Thus, in view of Eq. (28) and the definition of $\widehat{\mathcal{C}}^{\varepsilon}_{\text{gr-cal}}$, it holds that $(\boldsymbol{v}_0, \boldsymbol{v}_1) \in \widehat{\mathcal{C}}^{\varepsilon}_{\text{gr-cal}}$.

Part II: $\widetilde{\mathcal{C}}_{\mathrm{DP}}^{\delta} \subset \widehat{\mathcal{C}}_{\mathrm{DP}}^{\delta}$. We fix some $(u,v) \in \widetilde{\mathcal{C}}_{\mathrm{DP}}^{\delta}$. By assumption, $u,v \in [0,1]$ and

$$|\gamma_1 u - \gamma_0 v| \leqslant \tau\, \gamma_0 \gamma_1 \cdot \mathrm{TV}(\mathbf{Q}^0, \mathbf{Q}^1)\,. \tag{30}$$

Furthermore, on $\Omega^{\alpha_1,\alpha_2}$,

$$|\gamma_1 u - \gamma_0 v| = |\hat{\gamma}_1 u - \hat{\gamma}_0 v + (\gamma_1 - \hat{\gamma}_1)u - (\gamma_0 - \hat{\gamma}_0)v| \geqslant |\hat{\gamma}_1 u - \hat{\gamma}_0 v| - 2\alpha_2\,.$$

In view of Eq. (30) and the second bound of Eq. (29), we conclude that $(u,v) \in \widehat{\mathcal{C}}_{\mathrm{DP}}^{\delta}$ on $\Omega^{\alpha_1,\alpha_2}$. $\quad\square$

**Requirement (L2-Hausdorff) holds.** We bound separately the two expectations appearing in (L2-Hausdorff). As in the proof above, we omit the dependencies in $T_r$ in the notation.

Part I: bound on $\mathbb{E}\big[d(\mathcal{C}, \hat{\mathcal{C}}_r)^2\big]$. By definition of $d$, given that we are dealing with Euclidean projections onto a product set and are bounding square Euclidean distances, we have the decomposition:

$$\mathbb{E}\Big[d(\mathcal{C}, \hat{\mathcal{C}}_r)^2\Big] = \mathbb{E}\bigg[\sup_{\boldsymbol{x}\in\mathcal{C}} d(\boldsymbol{x}, \hat{\mathcal{C}}_r)^2\bigg] = \mathbb{E}\bigg[\sup_{\boldsymbol{x}\in\widetilde{\mathcal{C}}_{\mathrm{gr\text{-}cal}}^{\varepsilon}\times\widetilde{\mathcal{C}}_{\mathrm{DP}}^{\delta}} d(\boldsymbol{x}, \widehat{\mathcal{C}}_{\mathrm{gr\text{-}cal}}^{\varepsilon}\times\widehat{\mathcal{C}}_{\mathrm{DP}}^{\delta})^2\bigg]$$

$$= \mathbb{E}\bigg[\sup_{\boldsymbol{v}\in\widetilde{\mathcal{C}}_{\mathrm{gr\text{-}cal}}^{\varepsilon}} d(\boldsymbol{v}, \widehat{\mathcal{C}}_{\mathrm{gr\text{-}cal}}^{\varepsilon})^2\bigg] + \mathbb{E}\bigg[\sup_{(u,v)\in\widetilde{\mathcal{C}}_{\mathrm{DP}}^{\delta}} d\big((u,v), \widehat{\mathcal{C}}_{\mathrm{DP}}^{\delta}\big)^2\bigg]\,. \tag{31}$$

We start with the first term in the right-hand side of (31). As $\widehat{\mathcal{C}}_{\mathrm{gr\text{-}cal}}^{\varepsilon}$ always contains the null vector and $\widetilde{\mathcal{C}}_{\mathrm{gr\text{-}cal}}^{\varepsilon} \subseteq B_{\mathbb{R}^{2N}}^{\ell_1}$,

$$\sup_{\boldsymbol{v}\in\widetilde{\mathcal{C}}_{\mathrm{gr\text{-}cal}}^{\varepsilon}} d(\boldsymbol{v}, \widehat{\mathcal{C}}_{\mathrm{gr\text{-}cal}}^{\varepsilon})^2 \leqslant \sup_{\boldsymbol{v}\in\widetilde{\mathcal{C}}_{\mathrm{gr\text{-}cal}}^{\varepsilon}} \|\boldsymbol{v}\|^2 \leqslant \sup_{\boldsymbol{v}\in\widetilde{\mathcal{C}}_{\mathrm{gr\text{-}cal}}^{\varepsilon}} \|\boldsymbol{v}\|_1 \leqslant 1\,. \tag{32}$$

In addition, Lemma 4 ensures that on $\Omega^{\alpha_1,\alpha_2}$ we have $\widetilde{\mathcal{C}}_{\mathrm{gr\text{-}cal}}^{\varepsilon} \subset \widehat{\mathcal{C}}_{\mathrm{gr\text{-}cal}}^{\varepsilon}$, and hence $d(\widetilde{\mathcal{C}}_{\mathrm{gr\text{-}cal}}^{\varepsilon}, \widehat{\mathcal{C}}_{\mathrm{gr\text{-}cal}}^{\varepsilon}) = 0$, on $\Omega^{\alpha_1,\alpha_2}$. Thus, we can write

$$\mathbb{E}\bigg[\sup_{\boldsymbol{v}\in\widetilde{\mathcal{C}}_{\mathrm{gr\text{-}cal}}^{\varepsilon}} d(\boldsymbol{v}, \widehat{\mathcal{C}}_{\mathrm{gr\text{-}cal}}^{\varepsilon})^2\bigg] = \mathbb{E}\bigg[\big(\mathbb{I}\{\Omega^{\alpha_1,\alpha_2}\} + (1 - \mathbb{I}\{\Omega^{\alpha_1,\alpha_2}\})\big) \sup_{\boldsymbol{v}\in\widetilde{\mathcal{C}}_{\mathrm{gr\text{-}cal}}^{\varepsilon}} d(\boldsymbol{v}, \widehat{\mathcal{C}}_{\mathrm{gr\text{-}cal}}^{\varepsilon})^2\bigg]$$

$$\leqslant 0 + 1 - \mathbb{P}(\Omega^{\alpha_1,\alpha_2}) \leqslant \frac{1}{2T_r}\,,$$

where the inequality comes from the assumption made in Eq. (27) combined with Eq. (32).

A bound $1/T_r$ on the second term of Eq. (31) follows similarly, using that

$$\sup_{(u,v)\in\widetilde{\mathcal{C}}_{\mathrm{DP}}^{\delta}} d\big((u,v), \widehat{\mathcal{C}}_{\mathrm{DP}}^{\delta}\big)^2 \leqslant \sup_{(u,v)\in\widetilde{\mathcal{C}}_{\mathrm{DP}}^{\delta}} u^2 + v^2 \leqslant 2\,.$$

Hence, $\mathbb{E}\big[d(\mathcal{C}, \hat{\mathcal{C}}_r)^2\big] \leqslant 3/(2T_r)$.

Part II: bound on $\mathbb{E}\big[d(\hat{\mathcal{C}}_r, \mathcal{C})^2\big]$. We start in a similar manner:

$$\mathbb{E}\Big[d(\hat{\mathcal{C}}_r, \mathcal{C})^2\Big] = \mathbb{E}\bigg[\sup_{\boldsymbol{x}\in\hat{\mathcal{C}}_r} d(\boldsymbol{x}, \mathcal{C})^2\bigg] = \mathbb{E}\bigg[\sup_{\boldsymbol{x}\in\widehat{\mathcal{C}}_{\mathrm{gr\text{-}cal}}^{\varepsilon}\times\widehat{\mathcal{C}}_{\mathrm{DP}}^{\delta}} d(\boldsymbol{x}, \widetilde{\mathcal{C}}_{\mathrm{gr\text{-}cal}}^{\varepsilon}\times\widetilde{\mathcal{C}}_{\mathrm{DP}}^{\delta})^2\bigg]$$

$$= \mathbb{E}\bigg[\underbrace{\sup_{\boldsymbol{v}\in\widehat{\mathcal{C}}_{\mathrm{gr\text{-}cal}}^{\varepsilon}} d(\boldsymbol{v}, \widetilde{\mathcal{C}}_{\mathrm{gr\text{-}cal}}^{\varepsilon})^2}_{\leqslant 1 \text{ a.s.}}\bigg] + \mathbb{E}\bigg[\underbrace{\sup_{(u,v)\in\widehat{\mathcal{C}}_{\mathrm{DP}}^{\delta}} d\big((u,v), \widetilde{\mathcal{C}}_{\mathrm{DP}}^{\delta}\big)^2}_{\leqslant 2 \text{ a.s.}}\bigg]\,. \tag{33}$$

As in Part I, we start with the first term in the right-side of (33) and we split the expectation into two parts

$$\mathbb{E}\left[\sup_{\boldsymbol{v}\in\widehat{\mathcal{C}}^{\varepsilon}_{\text{gr-cal}}} d(\boldsymbol{v},\widetilde{\mathcal{C}}^{\varepsilon}_{\text{gr-cal}})^2\right] \leqslant \mathbb{E}\left[\mathbb{I}\{\Omega^{\alpha_1,\alpha_2}\} \sup_{\boldsymbol{v}\in\widehat{\mathcal{C}}^{\varepsilon}_{\text{gr-cal}}} d(\boldsymbol{v},\widetilde{\mathcal{C}}^{\varepsilon}_{\text{gr-cal}})^2\right] + \frac{1}{2T_r}. \tag{34}$$

Let us upper-bound the right-hand side expectation. We introduce some local short-hand notation. Given two real numbers $a, b$, we denote by $a \vee b$ and $a \wedge b$ the maximum and minimum between $a$ and $b$, respectively. We set $\alpha := \alpha_1 \vee \alpha_2$ and now show that

$$\text{on } \Omega^{\alpha_1,\alpha_2}, \qquad \sup_{\boldsymbol{v}\in\widehat{\mathcal{C}}^{\varepsilon}_{\text{gr-cal}}} d(\boldsymbol{v},\widetilde{\mathcal{C}}^{\varepsilon}_{\text{gr-cal}})^2 \leqslant \alpha^{2/3}\left(\frac{81}{(\gamma_0\gamma_1)^2} \vee \frac{10}{\gamma_0\wedge\gamma_1}\right). \tag{35}$$

We fix some $\boldsymbol{v} = (\boldsymbol{v}_0, \boldsymbol{v}_1) \in \widehat{\mathcal{C}}^{\varepsilon}_{\text{gr-cal}}$ and set $\boldsymbol{v}' = (\boldsymbol{v}'_0, \boldsymbol{v}'_1) := \lambda\boldsymbol{v}$ with

$$\lambda := 1 \wedge \left(\left(\frac{\gamma_0\gamma_1\varepsilon}{\gamma_0\gamma_1\varepsilon + 8\alpha}\right)\left(\frac{\hat{\gamma}_1}{\gamma_1} \wedge \frac{\hat{\gamma}_0}{\gamma_0}\right)\right).$$

The fact that $\boldsymbol{v} \in \widehat{\mathcal{C}}^{\varepsilon}_{\text{gr-cal}}$ entails that on $\Omega^{\alpha_1,\alpha_2}$,

$$\hat{\gamma}_1\|\boldsymbol{v}_0\|_1 + \hat{\gamma}_0\|\boldsymbol{v}_1\|_1 \leqslant \hat{\gamma}_0\hat{\gamma}_1\hat{\varepsilon} + 5\alpha \leqslant \big((\gamma_0+\alpha)\wedge 1\big)\big((\gamma_1+\alpha)\wedge 1\big)\big((\varepsilon+\alpha)\wedge 1\big) + 5\alpha \leqslant \gamma_0\gamma_1\varepsilon + 8\alpha. \tag{36}$$

Here, and in what follows, we repeatedly use that $\gamma_0, \gamma_1, \varepsilon$ and their estimates all lie in $[0,1]$. Furthermore, for the above-defined $\boldsymbol{v}'$, we can write on $\Omega^{\alpha_1,\alpha_2}$, by definition of $\lambda$,

$$\gamma_1\|\boldsymbol{v}'_0\|_1 + \gamma_0\|\boldsymbol{v}'_1\|_1 \leqslant \lambda\left(\frac{\gamma_1}{\hat{\gamma}_1} \vee \frac{\gamma_0}{\hat{\gamma}_0}\right)\underbrace{\big(\hat{\gamma}_1\|\boldsymbol{v}_0\|_1 + \hat{\gamma}_0\|\boldsymbol{v}_1\|_1\big)}_{\leqslant \gamma_0\gamma_1\varepsilon + 8\alpha} \leqslant \gamma_0\gamma_1\varepsilon,$$

implying that $\boldsymbol{v}' \in \widetilde{\mathcal{C}}^{\varepsilon}_{\text{gr-cal}}$. Thus, $d(\boldsymbol{v},\widetilde{\mathcal{C}}^{\varepsilon}_{\text{gr-cal}}) \leqslant \|\boldsymbol{v}-\boldsymbol{v}'\| = (1-\lambda)\|\boldsymbol{v}\|$ on $\Omega^{\alpha_1,\alpha_2}$. Since $\|\boldsymbol{v}\|_1 \leqslant 1$, we have $\|\boldsymbol{v}\| \leqslant \sqrt{\|\boldsymbol{v}\|_1} \leqslant 1$. All in all, we obtained the following upper bound on $\Omega^{\alpha_1,\alpha_2}$:

$$d(\boldsymbol{v},\widetilde{\mathcal{C}}^{\varepsilon}_{\text{gr-cal}})^2 \leqslant \|\boldsymbol{v}-\boldsymbol{v}'\|^2 \leqslant (1-\lambda)^2 \wedge \|\boldsymbol{v}\|_1.$$

We now bound separately each term to obtain the bound (35). First, on $\Omega^{\alpha_1,\alpha_2}$, we have

$$1 \geqslant \lambda \geqslant \left(\frac{\gamma_0\gamma_1\varepsilon}{\gamma_0\gamma_1\varepsilon + 8\alpha}\right)\left(\frac{\gamma_1-\alpha}{\gamma_1} \wedge \frac{\gamma_0-\alpha}{\gamma_0}\right)$$
$$\geqslant \frac{\gamma_0\gamma_1\varepsilon - \alpha\varepsilon(\gamma_0\vee\gamma_1)}{\gamma_0\gamma_1\varepsilon + 8\alpha} \geqslant 1 - \frac{9\alpha}{\gamma_0\gamma_1\varepsilon + 8\alpha} \geqslant 1 - \frac{9\alpha}{\gamma_0\gamma_1\varepsilon},$$

and thus,

$$(1-\lambda)^2 \leqslant \frac{81\alpha^2}{(\gamma_0\gamma_1\varepsilon)^2}.$$

Second, for $\|\boldsymbol{v}\|_1$, we start from (36) and write

$$\big(\gamma_1 \wedge \gamma_0 - \alpha\big)\|\boldsymbol{v}\|_1 \leqslant \big(\hat{\gamma}_1 \wedge \hat{\gamma}_0\big)\|\boldsymbol{v}\|_1 \leqslant \hat{\gamma}_1\|\boldsymbol{v}_0\|_1 + \hat{\gamma}_0\|\boldsymbol{v}_1\|_1 \leqslant \gamma_0\gamma_1\varepsilon + 8\alpha,$$

from which we get $(\gamma_1 \wedge \gamma_0)\|\boldsymbol{v}\|_1 \leqslant \gamma_0\gamma_1\varepsilon + 9\alpha$, which in turn yields

$$\|\boldsymbol{v}\|_1 \leqslant \varepsilon + \frac{9\alpha}{\gamma_0 \wedge \gamma_1}.$$

The bound $(1-\lambda)^2$ is convenient to use when $\varepsilon \geqslant \alpha^{2/3}$, while the bound on $\|\boldsymbol{v}\|_1$ will be used when $\varepsilon \leqslant \alpha^{2/3}$. When combining them by distinguishing these two cases, the $\wedge$ symbol needs to be replaced by a $\vee$ symbol, so, on $\Omega^{\alpha_1,\alpha_2}$

$$d(\boldsymbol{v},\widetilde{\mathcal{C}}^{\varepsilon}_{\text{gr-cal}})^2 \leqslant (1-\lambda)^2 \wedge \|\boldsymbol{v}\|_1 \leqslant \frac{81\alpha^2}{(\gamma_0\gamma_1\varepsilon)^2} \wedge \left(\varepsilon + \frac{9\alpha}{\gamma_0 \wedge \gamma_1}\right)$$
$$\leqslant \frac{81\alpha^{2/3}}{(\gamma_0\gamma_1)^2} \vee \left(\alpha^{2/3} + \frac{9\alpha}{\gamma_0 \wedge \gamma_1}\right),$$

which entails the claimed bound (35), via $\alpha \leqslant \alpha^{2/3} \leqslant 1$. From (34) and (35), we get

$$
\mathbb{E}\left[\sup_{\boldsymbol{v}\in\widehat{\mathcal{C}}^{\varepsilon}_{\text{gr-cal}}} d(\boldsymbol{v}, \widetilde{\mathcal{C}}^{\varepsilon}_{\text{gr-cal}})^2\right] \leqslant \frac{1}{2T_r} + C^{\text{gr-cal}}_{\gamma_0,\gamma_1}\mathbb{E}[\alpha^{2/3}],
$$

for some constant $C^{\text{gr-cal}}_{\gamma_0,\gamma_1}$ only depending on $\gamma_0$ and $\gamma_1$.

The second term of the decomposition (33) of $\mathbb{E}\left[d(\hat{\mathcal{C}}_r, \mathcal{C})^2\right]$ can be handled similarly, leading to the existence of a constant $C_{\gamma_0,\gamma_1}$, only depending on $\gamma_0$ and $\gamma_1$, such that

$$
\mathbb{E}\left[d(\hat{\mathcal{C}}_r, \mathcal{C})^2\right] \leqslant \frac{3}{2T_r} + C_{\gamma_0,\gamma_1}\mathbb{E}[\alpha^{2/3}],
$$

where the expectation in the right-hand side is due to the fact that $\alpha_1, \alpha_2$ might be data-dependent.

Combining Part I and Part II. The bound of Part II contains an additional term compared to the one of Part I. We have thus have proved so far (writing again the dependencies on $T_r$):

$$
\max\left\{\mathbb{E}\left[d(\hat{\mathcal{C}}_r, \mathcal{C})^2\right], \mathbb{E}\left[d(\mathcal{C}, \hat{\mathcal{C}}_r)^2\right]\right\} \leqslant \frac{3}{2T_r} + C_{\gamma_0,\gamma_1}\mathbb{E}[\alpha(T_r)^{2/3}]. \tag{37}
$$

To get the desired property (L2-Hausdorff), we only need to make sure that the right hand side of (37) can be upper bounded by $\beta_r^2$ where $(\beta_r)$ is non-increasing and summable. Recall that our proof also relied on the assumption (27). We now illustrate that indeed, $\alpha_1(T_r) \leqslant 1$ and $\alpha_2(T_r) \leqslant 1$ may be set in a way such that all these facts hold. For the sake of simplicity, we provide the illustration for the case of finite set $\mathcal{X}$.

**Choice of $\alpha_1(T_r)$ and $\alpha_2(T_r)$, part 2: illustration for finite sets $\mathcal{X}$.** Based on the $T$–sample $(x_t, s_t)_{1 \leqslant t \leqslant T}$ with distribution $\mathbf{Q}$, we denote by

$$
N_{s,T} = \sum_{t=1}^{T} \mathbb{I}\{s_t = s\}
$$

the number of occurrences of the value $s \in \{0, 1\}$ of the sensitive context, and consider the empirical frequencies $\hat{\gamma}_{0,T} = N_{0,T}/T$ and $\hat{\gamma}_{1,T} = N_{1,T}/T$ to estimate the frequencies $\gamma_0$ and $\gamma_1$ of the sensitive contexts.

The choice of $\widehat{M}_T$, and hence, the one of $\alpha_1(T)$, depend heavily on the possibly additional assumptions on the marginal distributions $\mathbf{Q}^0$ and $\mathbf{Q}^1$. We illustrate such a choice for the case where $\mathcal{X}$ is a finite set. In that case, we may consider the empirical distributions $\hat{\mathbf{Q}}^0_T$ and $\hat{\mathbf{Q}}^1_T$ for these marginals: for each $s \in \{0, 1\}$, $\hat{\mathbf{Q}}^s_T$ is some arbitrary distribution over $\mathcal{X}$ (say, the uniform distribution) when $N_{s,T} = 0$, and otherwise, for each $x \in \mathcal{X}$,

$$
\hat{\mathbf{Q}}^s_T(x) = \frac{1}{N_{s,T}}\sum_{t=1}^{T} \mathbb{I}\{x_t = x, s_t = s\}.
$$

Then, we consider the plug-in estimate $\widehat{M}_T := \text{TV}(\hat{\mathbf{Q}}^0_T, \hat{\mathbf{Q}}^1_T)$ of $\text{TV}(\mathbf{Q}^0, \mathbf{Q}^1)$.

Proof of (27), part I. We set

$$
\alpha_2(T) = 1 \wedge \sqrt{\frac{\log(8T)}{2T}}
$$

and note that by Hoeffding's inequality (and the fact that we only have two classes and that probabilities sum up to 1), for those $T$ for which $\alpha_2(T) < 1$,

$$
\mathbb{P}\Big(\forall s \in \{0, 1\}, \ |\hat{\gamma}_{s,T} - \gamma_s| > \alpha_2(T)\Big) = \mathbb{P}\big(|\hat{\gamma}_{0,T} - \gamma_0| > \alpha_2(T)\big)
$$

$$
\leqslant 2\exp\big(-2T\,\alpha_2(T)^2\big) = \frac{1}{4T}. \tag{38}
$$

For $T$ such that $\alpha_2(T) = 1$, the probability above is null, as $|\hat{\gamma}_{s,T} - \gamma_s| \leqslant 1$ a.s., and therefore, the final $1/(4T)$ bound holds in particular.

Proof of (27), part II. We set $\theta(0) = 1$ and $\theta(n) := \sqrt{\dfrac{|\mathcal{X}| + \log(8T)}{2n}}$ for $n \geqslant 1$, and define

$$\alpha_1(T) := 1 \wedge \big(\theta(N_{0,T}) + \theta(N_{1,T})\big).$$

We now prove that

$$\mathbb{P}\Big(\big|\widehat{M}_T - \mathrm{TV}(\mathbf{Q}^0, \mathbf{Q}^1)\big| > \alpha_1(T)\Big) \leqslant \frac{1}{4T}. \tag{39}$$

The property (27) then follows from the bounds (38) and (39) at $T = T_r$.

Using that $\big|\widehat{M}_T - \mathrm{TV}(\mathbf{Q}^0, \mathbf{Q}^1)\big| \leqslant 1$ a.s. (for the first inequality in the display below) and the triangle inequality

$$\big|\widehat{M}_T - \mathrm{TV}(\mathbf{Q}^0, \mathbf{Q}^1)\big| \leqslant \mathrm{TV}(\mathbf{Q}^0, \hat{\mathbf{Q}}^0_T) + \mathrm{TV}(\mathbf{Q}^1, \hat{\mathbf{Q}}^1_T)$$

(for the second inequality in the display below), we have

$$\begin{aligned}
\mathbb{P}\Big(\big|\widehat{M}_T - \mathrm{TV}(\mathbf{Q}^0, \mathbf{Q}^1)\big| > \alpha_1(T)\Big) &= \mathbb{P}\Big(\big|\widehat{M}_T - \mathrm{TV}(\mathbf{Q}^0, \mathbf{Q}^1)\big| > \theta(N_{0,T}) + \theta(N_{1,T})\Big) \\
&\leqslant \mathbb{P}\Big(\mathrm{TV}(\mathbf{Q}^0, \hat{\mathbf{Q}}^0_T) + \mathrm{TV}(\mathbf{Q}^1, \hat{\mathbf{Q}}^1_T) > \theta(N_{0,T}) + \theta(N_{1,T})\Big) \\
&\leqslant \sum_{s \in \{0,1\}} \mathbb{P}\Big(\mathrm{TV}(\mathbf{Q}^s, \hat{\mathbf{Q}}^s_T) > \theta(N_{s,T})\Big).
\end{aligned} \tag{40}$$

The conclusion (39) follows from showing that for each $s \in \{0, 1\}$,

$$\mathbb{P}\Big(\mathrm{TV}(\mathbf{Q}^s, \hat{\mathbf{Q}}^s_T) > \theta(N_{s,T})\Big) \leqslant \frac{1}{8T}.$$

A useful auxiliary result to that end is the following. Denote by $\hat{\mathbf{P}}^n$ the empirical frequencies of some probability distribution $\mathbf{P}$ on $\mathcal{X}$ based on a sample of deterministic size $n \geqslant 1$. Hoeffding's inequality and a union bound over the $\leqslant 2^{|\mathcal{X}|}$ subsets of $\mathcal{X}$ ensure that for all $\theta \geqslant 0$,

$$\mathbb{P}\big(\mathrm{TV}(\mathbf{P}, \hat{\mathbf{P}}^n) \geqslant \theta\big) = \mathbb{P}\Big(\max_{A \subset \mathcal{X}} \big(\mathbf{P}(A) - \hat{\mathbf{P}}^n(A)\big) \geqslant \theta\Big) \leqslant 2^{|\mathcal{X}|} \exp(-2n\theta^2). \tag{41}$$

In our case, note however that the estimators $\hat{\mathbf{Q}}^s_T$ at time $T$ are built on a random number $N_{s,T}$ of samples. We therefore decompose the probability of interest according to the values of $N_{s,T}$: for each $s \in \{0, 1\}$,

$$\begin{aligned}
\mathbb{P}\Big(\mathrm{TV}(\mathbf{Q}^s, \hat{\mathbf{Q}}^s_T) > \theta(N_{s,T})\Big) &= \sum_{n=0}^{T} \mathbb{P}\Big(N_{s,T} = n \text{ and } \mathrm{TV}(\mathbf{Q}^s, \hat{\mathbf{Q}}^s_T) > \theta(n)\Big) \\
&= \sum_{n=1}^{T} \mathbb{P}\Big(N_{s,T} = n \text{ and } \mathrm{TV}(\mathbf{Q}^s, \hat{\mathbf{Q}}^s_T) > \theta(n)\Big) \\
&= \sum_{n=1}^{T} \mathbb{P}(N_{s,T} = n) \, \mathbb{P}\Big(\mathrm{TV}(\mathbf{Q}^s, \hat{\mathbf{Q}}^{s,n}) > \theta(n)\Big),
\end{aligned}$$

where the second equality follows from the choice $\theta(0) = 1$ and the fact that a total variation is always smaller than 1, and where the third equality follows by conditional independence with $\hat{\mathbf{Q}}^{s,n}$ denoting the empirical distribution based on a $\mathbf{Q}^s$–sample of size $n$. Substituting the bound (41) and the definition of the $\theta(n)$, we get

$$\begin{aligned}
\mathbb{P}\Big(\mathrm{TV}(\mathbf{Q}^s, \hat{\mathbf{Q}}^s_T) > \theta(N_{s,T})\Big) &\leqslant 2^{|\mathcal{X}|} \sum_{n=1}^{T} \mathbb{P}(N_{s,T} = n) \exp\big(-2n\theta(n)^2\big) \\
&= 2^{|\mathcal{X}|} \sum_{n=1}^{T} \mathbb{P}(N_{s,T} = n) \exp\big(-|\mathcal{X}| - \log(8T)\big) \\
&\leqslant \underbrace{(2/e)^{|\mathcal{X}|}}_{\leqslant 1} \frac{1}{8T} \underbrace{\sum_{n=1}^{T} \mathbb{P}(N_{s,T} = n)}_{\leqslant 1} \leqslant \frac{1}{8T},
\end{aligned}$$

which is exactly what remained to be proven.

Control of the right-hand side of (37). It involves $\mathbb{E}[\alpha(T_r)^{2/3}]$, where

$$\alpha(T_r) = \alpha_1(T_r) \vee \alpha_2(T_r) = \alpha_1(T_r) = 1 \wedge \left( \sqrt{\frac{|\mathcal{X}| + \log(8T)}{2N_{0,T}}} + \sqrt{\frac{|\mathcal{X}| + \log(8T)}{2N_{1,T}}} \right)$$

$$\leqslant \sum_{s \in \{0,1\}} 1 \wedge \sqrt{\frac{|\mathcal{X}| + \log(8T)}{2N_{s,T}}} \, .$$

Now, note that $N_{s,T}$ follows the binomial distribution with parameters $\gamma_s$ and $T$. Thus, for each $s \in \{0,1\}$,

$$\mathbb{E}\left[ \left( \frac{|\mathcal{X}| + \log(8T)}{2N_{s,T}} \right)^{1/3} \wedge 1 \right] \leqslant \mathbb{P}(N_{s,T} \leqslant T\gamma_s/2) + \left( \frac{|\mathcal{X}| + \log(8T)}{\gamma_s T} \right)^{1/3}$$

$$\leqslant \exp\left( -\gamma_s^2 T/2 \right) + \left( \frac{|\mathcal{X}| + \log(8T)}{\gamma_s T} \right)^{1/3} \, ,$$

where we applied Hoeffding's inequality to get the last bound. Therefore, recalling that $T_r = 2^r$, the bound of Eq. (37) may be further bounded as: for all $r \geqslant 2$,

$$\frac{3}{2T_r} + C_{\gamma_0,\gamma_1} \mathbb{E}\left[\alpha(T_r)^{2/3}\right] \leqslant C'_{\gamma_0,\gamma_1} \left( \frac{r}{2^r} \right)^{1/3} =: \beta_r^2 \, ,$$

for some constant $C'_{\gamma_0,\gamma_1}$ depending only on $\gamma_0$ and $\gamma_1$. We observe that $\beta_r = \sqrt{C'_{\gamma_0,\gamma_1}} \, (r\, 2^{-r})^{1/6}$ is non-increasing for $r \geqslant 2$ and summable, as required.

We emphasize that, while exact values of $\alpha_1(T_r)$ and $\alpha_2(T_r)$ are needed for the construction of the set-estimate $\hat{\mathcal{C}}_r$, the knowledge of $\beta_r$ is not required by the algorithm (its choice is required for the sake of the theoretical analysis only).

## D.2 Proof of Theorem 2

We actually prove a more complete and more precise version of Theorem 2.

**Theorem 3** (contains Theorem 2). *Under Assumption 3 and the assumptions of Theorem 1, a convex closed set $\mathcal{C}$, unknown to the Player, is $\boldsymbol{m}$–approachable if and only if Blackwell's condition in Eq. (2) is satisfied. In this case, the strategy of Eq. (7) is an approachability strategy. It achieves the following rates for $L^2$ convergence: for all $r \geqslant 1$ and all $t \in [T_r, T_{r+1} - 1]$,*

$$\sqrt{\mathbb{E}[d_t^2]} \leqslant \frac{\sqrt{6B^2 + 8B\|\boldsymbol{m}\|_{\infty,2}}}{(\sqrt{2}-1)\sqrt{t}} + \frac{4\|\boldsymbol{m}\|_{\infty,2}}{t} \sum_{t'=1}^{t-1} \sqrt{\mathbb{E}[\mathrm{TV}^2(\mathbf{Q}, \hat{\mathbf{Q}}_{t'})]} + \frac{4}{t} \sum_{r'=0}^{r} T_{r'} \beta_{r'} \, .$$

*It also achieves the following rates for almost-sure convergence: for all $r \geqslant 1$,*

$$\mathbb{P}\left( \sup_{t \geqslant T_r} d_t \geqslant 2\varepsilon \right) \leqslant \frac{\Xi_r}{\varepsilon^2} + \frac{1}{\varepsilon^2} \sum_{r' \geqslant r} \beta_{r'}^2 \, ,$$

*where $\Xi_r$ is defined in Eq. (60), page 39, and converges to 0.*

Comments after Assumption 1 explain why the middle term in the $L^2$ bound vanishes. Assumption 3 indicates that the series $(\beta_r)_{r \geqslant 1}$ is summable, hence the following sequence of Cesaro averages built on it also vanishes:

$$\overline{\beta}_r := \frac{1}{T_{r+1}} \sum_{r'=0}^{r} T_{r'} \beta_{r'} \to 0 \, . \tag{42}$$

Finally, the series $(\beta_r^2)_{r \geqslant 1}$ is also summable, hence its associated sequence of remainder sums also vanishes:

$$\sum_{r' \geqslant r} \beta_{r'}^2 \to 0 \, .$$

We now move to the proof. We simply note at this stage that the condition $\|\boldsymbol{v} - \mathrm{Proj}_{\hat{\mathcal{C}}_r}(\boldsymbol{v})\| \leqslant B$ for all $\boldsymbol{v} \in \boldsymbol{m}(\mathcal{A}, \mathcal{B}, \mathcal{X}, \{0,1\})$ of Assumption 3 also holds, by convexity, for all $\boldsymbol{v}$ in the convex hull of $\boldsymbol{m}(\mathcal{A}, \mathcal{B}, \mathcal{X}, \{0,1\})$.

*Proof.* The proof is required only for the sufficiency, since the necessity was proven in Theorem 1.

Recall that, from the perspective of the Player, the game proceeds in phases lasting from $T_r := 2^r$ to $T_{r+1} - 1 := 2^{r+1} - 1$. For each time $t \in [T_r, T_{r+1} - 1]$, the Player uses $\hat{\mathcal{C}}_r$ as an estimate of the true target set $\mathcal{C}$, and updates to $\hat{\mathcal{C}}_{r+1}$ only at $t = T_{r+1}$. The initial stage of the proof is split into two parts: first, we closely follow the proof of Theorem 1 and analyze the game for $t \in [T_r, T_{r+1} - 1]$; then, we handle the case of transition from $\hat{\mathcal{C}}_r$ to $\hat{\mathcal{C}}_{r+1}$.

We introduce the following short-hand notation:

$$\hat{d}_t := \|\overline{\boldsymbol{m}}_t - \hat{\boldsymbol{c}}_t\| \qquad \text{and} \qquad \Omega_r = \left\{ \mathcal{C} \subset \hat{\mathcal{C}}_r \right\}.$$

Note that unlike the quantity of interest $d_t = \|\overline{\boldsymbol{m}}_t - \overline{\boldsymbol{c}}_t\|$, which is equal to the distance from the average payoff $\overline{\boldsymbol{m}}_t$ along the trajectory to the *true* target set $\overline{\boldsymbol{c}}_t$, the distance $\hat{d}_t$ is with respect to the currently used estimate $\hat{\mathcal{C}}_r$. The key insight of the proof is hidden in the fact that, if $\Omega_r$ occurs, then the approachability condition, which is met by $\mathcal{C}$, is also met by the super-set estimate $\hat{\mathcal{C}}_r$.

**Convergence in $L^2$.** Let us start with the following observation, which relates $d_t$ to $\hat{d}_t$, based on Assumption 3. We have for $t \in [T_r, T_{r+1} - 1]$,

$$
\begin{aligned}
d_t = \|\overline{\boldsymbol{m}}_t - \mathrm{Proj}_{\mathcal{C}} \, \overline{\boldsymbol{m}}_t\| &\leqslant \|\overline{\boldsymbol{m}}_t - \mathrm{Proj}_{\mathcal{C}} \, \mathrm{Proj}_{\hat{\mathcal{C}}_r} \, \overline{\boldsymbol{m}}_t\| \\
&\leqslant \|\overline{\boldsymbol{m}}_t - \mathrm{Proj}_{\hat{\mathcal{C}}_r} \, \overline{\boldsymbol{m}}_t\| + \|\mathrm{Proj}_{\hat{\mathcal{C}}_r} \, \overline{\boldsymbol{m}}_t - \mathrm{Proj}_{\mathcal{C}} \, \mathrm{Proj}_{\hat{\mathcal{C}}_r} \, \overline{\boldsymbol{m}}_t\| \\
&\leqslant \hat{d}_t + d(\hat{\mathcal{C}}_r, \mathcal{C}).
\end{aligned}
\tag{43}
$$

Hence, according to the fourth item of Assumption 3 and the $L^2$-triangular inequality, we have

$$\sqrt{\mathbb{E}[d_t^2]} \leqslant \sqrt{\mathbb{E}[\hat{d}_t^2]} + \beta_r.
\tag{44}$$

Since $\beta_r \to 0$ according to Assumption 3, the latter implies that, if $\mathbb{E}[\hat{d}_t^2] \to 0$, then $\mathbb{E}[d_t^2] \to 0$.

As already mentioned, to prove the $L^2$-convergence, we consider two cases. In the first case, we study the evolution of the game withing one phase, that is for $t \in [T_t, T_{r+1} - 2]$ – the case where we project onto $\hat{\mathcal{C}}_r$. The second case is when $t = T_{r+1} - 1$, that is, when in the next round we are going to update the estimate $\hat{\mathcal{C}}_r$.

*Case $T_r \leqslant t \leqslant T_{r+1} - 2$:* Defining

$$Z_{t+1} := \left\langle \overline{\boldsymbol{m}}_t - \hat{\boldsymbol{c}}_t, \boldsymbol{m}_{t+1} - \int_{\mathcal{X} \times \mathcal{S}} \boldsymbol{m}\big(\boldsymbol{p}_{t+1}^x, \boldsymbol{q}_{t+1}^{G(x,s)}, x, s\big) \, \mathrm{d}\mathbf{Q}(x,s) \right\rangle,
\tag{45}$$

$$\text{and} \quad B_t := \left\langle \overline{\boldsymbol{m}}_t - \hat{\boldsymbol{c}}_t, \int_{\mathcal{X} \times \mathcal{S}} \boldsymbol{m}\big(\boldsymbol{p}_{t+1}^x, \boldsymbol{q}_{t+1}^{G(x,s)}, x, s\big) \, \mathrm{d}\mathbf{Q}(x,s) - \hat{\boldsymbol{c}}_t \right\rangle,
\tag{46}$$

we can write

$$
\begin{aligned}
\hat{d}_{t+1}^2 &\leqslant \|\overline{\boldsymbol{m}}_{t+1} - \hat{\boldsymbol{c}}_t\| \\
&\leqslant \frac{t^2}{(t+1)^2} \hat{d}_t^2 + \frac{1}{(t+1)^2} \|\boldsymbol{m}_{t+1} - \hat{\boldsymbol{c}}_t\|^2 + \frac{2t}{(t+1)^2} (Z_{t+1} + B_t).
\end{aligned}
\tag{47}$$

As in the proof of Theorem 1, the main non-standard analysis is connected with the treatment of $B_t$. Observe that thanks to Assumption 3, we always have $|B_t| \leqslant B(B + 2\|\boldsymbol{m}\|_{\infty,2})$, hence

$$B_t \leqslant B_t \mathbb{I}\{\Omega_r\} + B(B + 2\|\boldsymbol{m}\|_{\infty,2})\mathbb{I}\{\Omega_r^c\}.
\tag{48}$$

Furthermore, similarly as for Eq. (14), we have on $\Omega_r$

$$
\begin{aligned}
B_t \leqslant{}& 4\|\boldsymbol{m}\|_{\infty,2} \, \hat{d}_t \cdot \mathrm{TV}(\mathbf{Q}, \hat{\mathbf{Q}}_t) \\
&+ \min_{(\boldsymbol{p}^x)_{x \in \mathcal{X}}} \max_{(\boldsymbol{q}^{G(x,s)})_{(x,s) \in \mathcal{X} \times \mathcal{S}}} \left\langle \overline{\boldsymbol{m}}_t - \hat{\boldsymbol{c}}_t, \int_{\mathcal{X} \times \mathcal{S}} \boldsymbol{m}\big(\boldsymbol{p}^x, \boldsymbol{q}^{G(x,s)}, x, s\big) \, \mathrm{d}\mathbf{Q}(x,s) - \hat{\boldsymbol{c}}_t \right\rangle.
\end{aligned}
\tag{49}$$

Since, by definition of $\Omega_r$, we have the inclusion $\mathcal{C} \subset \hat{\mathcal{C}}_r$ on $\Omega_r$, Blackwell's condition (2) implies that, on $\Omega_r$,

$$\forall (\boldsymbol{q}^{G(x,s)})_{(x,s) \in \mathcal{X} \times \{0,1\}} \ \exists (\boldsymbol{p}^x)_{x \in \mathcal{X}} \quad \text{s.t.} \quad \int_{\mathcal{X} \times \mathcal{S}} \boldsymbol{m}(\boldsymbol{p}^x, \boldsymbol{q}^{G(x,s)}, x, s) \, \mathrm{d}\mathbf{Q}(x,s) \in \mathcal{C} \subset \hat{\mathcal{C}}_r \,.$$

The first item of Assumption 3 requires $\hat{\mathcal{C}}_r$ to be closed convex almost surely. Hence, using the property of Euclidean projection onto a convex closed set, in conjunction with von Neumann's minmax theorem, we conclude that, on $\Omega_r$, it holds that

$$\min_{(\boldsymbol{p}^x)_{x \in \mathcal{X}}} \max_{(\boldsymbol{q}^{G(x,s)})_{(x,s) \in \mathcal{X} \times \mathcal{S}}} \left\langle \overline{\boldsymbol{m}}_t - \hat{\boldsymbol{c}}_t, \int_{\mathcal{X} \times \mathcal{S}} \boldsymbol{m}(\boldsymbol{p}^x, \boldsymbol{q}^{G(x,s)}, x, s) \, \mathrm{d}\mathbf{Q}(x,s) - \hat{\boldsymbol{c}}_t \right\rangle \leqslant 0 \,.$$

The above inequality, combined with Eqs. (47)–(49), yields

$$\hat{d}_{t+1}^2 \leqslant \frac{t^2}{(t+1)^2} \hat{d}_t^2 + \frac{1}{(t+1)^2} \|\boldsymbol{m}_{t+1} - \hat{\boldsymbol{c}}_t\|^2$$
$$+ \frac{2t}{(t+1)^2} \left( Z_{t+1} + 4\|\boldsymbol{m}\|_{\infty,2} \, \hat{d}_t \cdot \mathrm{TV}(\mathbf{Q}, \hat{\mathbf{Q}}_t) \mathbb{I}\{\Omega_r\} + B(B + 2\|\boldsymbol{m}\|_{\infty,2}) \mathbb{I}\{\Omega_r^c\} \right) \,. \tag{50}$$

Since $(Z_t)_{t \geqslant 1}$ is martingale difference (by the same arguments as for the proof of Theorem 1), taking expectations from both sides of the above inequality, in conjunction with the condition on $\mathbb{P}(\Omega_r)$ of Assumption 3 and the Cauchy-Schwartz inequality, yields

$$\mathbb{E}[\hat{d}_{t+1}^2] \leqslant \frac{t^2}{(t+1)^2} \mathbb{E}[\hat{d}_t^2] + \frac{B^2}{(t+1)^2}$$
$$+ \frac{2t}{(t+1)^2} \left( 4\|\boldsymbol{m}\|_{\infty,2} \sqrt{\mathbb{E}[\hat{d}_t^2]} \cdot \sqrt{\mathbb{E}[\mathrm{TV}^2(\mathbf{Q}, \hat{\mathbf{Q}}_t)]} + \frac{B(B + 2\|\boldsymbol{m}\|_{\infty,2})}{2T_r} \right) \,.$$

We deduce from the above that, for all $t \in [T_r, T_{r+1} - 2]$, since $t/(2T_r) \leqslant 1$,

$$\mathbb{E}[\hat{d}_{t+1}^2] \leqslant \frac{t^2}{(t+1)^2} \mathbb{E}[\hat{d}_t^2] + \frac{3B^2 + 4B\|\boldsymbol{m}\|_{\infty,2}}{(t+1)^2}$$
$$+ \frac{2t}{(t+1)^2} \left( 4\|\boldsymbol{m}\|_{\infty,2} \sqrt{\mathbb{E}[\hat{d}_t^2]} \cdot \sqrt{\mathbb{E}[\mathrm{TV}^2(\mathbf{Q}, \hat{\mathbf{Q}}_t)]} \right) \,. \tag{51}$$

Applying Lemma 1 with $t^* = T_r$, $K = 3B^2 + 4B\|\boldsymbol{m}\|_{\infty,2}$, and $\delta_t = 4\|\boldsymbol{m}\|_{\infty,2} \cdot \sqrt{\mathbb{E}[\mathrm{TV}^2(\mathbf{Q}, \hat{\mathbf{Q}}_t)]}$, we obtain that, for all $t \in [T_r, T_{r+1} - 1]$,

$$\sqrt{\mathbb{E}[\hat{d}_t^2]} \leqslant \frac{\sqrt{(3B^2 + 4B\|\boldsymbol{m}\|_{\infty,2})(t - T_r)}}{t} + \frac{4}{t} \sum_{t' = T_r}^{t-1} \|\boldsymbol{m}\|_{\infty,2} \cdot \sqrt{\mathbb{E}[\mathrm{TV}^2(\mathbf{Q}, \hat{\mathbf{Q}}_{t'})]}$$
$$+ \frac{T_r}{t} \sqrt{\mathbb{E}[\hat{d}_{T_r}^2]} \,. \tag{52}$$

*Case $t = T_{r+1} - 1$:* In this case, when passing from $t$ to $t + 1$, the Player updates the estimate of the target set $\mathcal{C}$, which incurs additional price. In particular, the established recursion in Eq (52) does not hold, since by definition $\hat{d}_{T_{r+1}} = \|\overline{\boldsymbol{m}}_{T_{r+1}} - \hat{\boldsymbol{c}}_{T_{r+1}}\|$, where $\hat{\boldsymbol{c}}_{T_{r+1}}$ is the projection onto $\hat{\mathcal{C}}_{r+1}$. However, note that the argument of the first case still holds if we fix the set onto which we project. More formally, the inequality (51) still holds at $t = T_{r+1} - 1$, if we replace $\hat{d}_{T_{r+1}}$ in the left-hand side by

$$\tilde{d}_{T_{r+1}} := \|\overline{\boldsymbol{m}}_{T_{r+1}} - \mathrm{Proj}_{\hat{\mathcal{C}}_r}(\overline{\boldsymbol{m}}_{T_{r+1}})\| \,.$$

Hence $\sqrt{\mathbb{E}[\tilde{d}_{T_{r+1}}^2]}$ is smaller than the right-hand side of Eq. (52) with $t = T_{r+1}$. Applying the same argument as in (43), and applying Minkowski's inequality, we get

$$\sqrt{\mathbb{E}[\hat{d}_{T_{r+1}}^2]} = \sqrt{\mathbb{E}[\|\overline{\boldsymbol{m}}_{T_{r+1}} - \mathrm{Proj}_{\hat{\mathcal{C}}_{r+1}}(\overline{\boldsymbol{m}}_{T_{r+1}})\|^2]}$$
$$\leqslant \sqrt{\mathbb{E}[\|\overline{\boldsymbol{m}}_{T_{r+1}} - \mathrm{Proj}_{\hat{\mathcal{C}}_r}(\overline{\boldsymbol{m}}_{T_{r+1}})\|^2]} + \sqrt{\mathbb{E}[d(\hat{\mathcal{C}}_r, \hat{\mathcal{C}}_{r+1})^2]}$$
$$= \sqrt{\mathbb{E}[\tilde{d}_{T_{r+1}}^2]} + \sqrt{\mathbb{E}[d(\hat{\mathcal{C}}_r, \hat{\mathcal{C}}_{r+1})^2]} \,.$$

Recalling that the bound in Eq. (52) holds for $\sqrt{\mathbb{E}[\tilde{d}_{T_{r+1}}^2]}$, and using the above derived relation, we get for all $r \geqslant 0$

$$\sqrt{\mathbb{E}[\hat{d}_{T_{r+1}}^2]} \leqslant \frac{\sqrt{(3B^2 + 4B\|\boldsymbol{m}\|_{\infty,2})(T_{r+1} - T_r)}}{T_{r+1}} + \frac{4}{T_{r+1}} \sum_{t'=T_r}^{T_{r+1}-1} \|\boldsymbol{m}\|_{\infty,2} \cdot \sqrt{\mathbb{E}[\mathrm{TV}^2(\mathbf{Q}, \hat{\mathbf{Q}}_{t'})]}$$

$$+ \frac{T_r}{T_{r+1}} \sqrt{\mathbb{E}[\hat{d}_{T_r}^2]} + \sqrt{\mathbb{E}\big[d(\hat{\mathcal{C}}_r, \hat{\mathcal{C}}_{r+1})^2\big]} .$$

(53)

Multiplying Eq. (53) by $T_{r+1}$ on both sides and rearranging, we deduce that for all $r \geqslant 0$

$$\left( T_{r+1} \sqrt{\mathbb{E}[\hat{d}_{T_{r+1}}^2]} - T_r \sqrt{\mathbb{E}[\hat{d}_{T_r}^2]} \right) \leqslant \sqrt{(3B^2 + 4B\|\boldsymbol{m}\|_{\infty,2})(T_{r+1} - T_r)}$$

$$+ 4 \sum_{t'=T_r}^{T_{r+1}-1} \|\boldsymbol{m}\|_{\infty,2} \cdot \sqrt{\mathbb{E}[\mathrm{TV}^2(\mathbf{Q}, \hat{\mathbf{Q}}_{t'})]}$$

$$+ T_{r+1} \sqrt{\mathbb{E}\big[d(\hat{\mathcal{C}}_r, \hat{\mathcal{C}}_{r+1})^2\big]} .$$

Summing up the above inequalities over $r \geqslant 0$, and using the fact that, by Assumption 3, $\hat{d}_1 \leqslant B$, we obtain, with the convention $T_{-1} = 0$,

$$\sqrt{\mathbb{E}[\hat{d}_{T_r}^2]} \leqslant \sqrt{3B^2 + 4B\|\boldsymbol{m}\|_{\infty,2}} \frac{1}{T_r} \sum_{r'=0}^{r} \sqrt{T_{r'} - T_{r'-1}}$$

(54)

$$+ 4\|\boldsymbol{m}\|_{\infty,2} \frac{1}{T_r} \sum_{t'=1}^{T_r-1} \sqrt{\mathbb{E}[\mathrm{TV}^2(\mathbf{Q}, \hat{\mathbf{Q}}_{t'})]} + \frac{1}{T_r} \sum_{r'=1}^{r} T_{r'} \sqrt{\mathbb{E}\big[d(\hat{\mathcal{C}}_{r'-1}, \hat{\mathcal{C}}_{r'})^2\big]} .$$

To conclude the convergence in $L^2$, we observe that $d(\hat{\mathcal{C}}_{r'-1}, \hat{\mathcal{C}}_{r'}) \leqslant d(\hat{\mathcal{C}}_{r'-1}, \mathcal{C}) + d(\mathcal{C}, \hat{\mathcal{C}}_{r'})$ and hence, the triangle inequality for $L^2$-norms and Assumption 3 yield

$$\sqrt{\mathbb{E}\big[d(\hat{\mathcal{C}}_{r'-1}, \hat{\mathcal{C}}_{r'})^2\big]} \leqslant \sqrt{\mathbb{E}\big[d(\hat{\mathcal{C}}_{r'-1}, \mathcal{C})^2\big]} + \sqrt{\mathbb{E}\big[d(\mathcal{C}, \hat{\mathcal{C}}_{r'})^2\big]} \leqslant \beta_{r'-1} + \beta_{r'} \leqslant 2\beta_{r'-1}. \quad (55)$$

Substituting the above bound in Eq. (54) (and reindexing, using that $T_{r'} = 2T_{r'-1}$), we get for all $r \geqslant 1$

$$\sqrt{\mathbb{E}[\hat{d}_{T_r}^2]} \leqslant \frac{\sqrt{3B^2 + 4B\|\boldsymbol{m}\|_{\infty,2}}}{T_r} \sum_{r'=0}^{r} \sqrt{T_{r'} - T_{r'-1}} + 4\|\boldsymbol{m}\|_{\infty,2} \frac{1}{T_r} \sum_{t'=1}^{T_r-1} \sqrt{\mathbb{E}[\mathrm{TV}^2(\mathbf{Q}, \hat{\mathbf{Q}}_{t'})]}$$

$$+ \frac{4}{T_r} \sum_{r'=0}^{r-1} T_{r'} \beta_{r'}$$

$$\leqslant \frac{\sqrt{3B^2 + 4B\|\boldsymbol{m}\|_{\infty,2}}}{(\sqrt{2} - 1)\sqrt{T_r}} + 4\|\boldsymbol{m}\|_{\infty,2} \frac{1}{T_r} \sum_{t'=1}^{T_r-1} \sqrt{\mathbb{E}[\mathrm{TV}^2(\mathbf{Q}, \hat{\mathbf{Q}}_{t'})]} + \frac{4}{T_r} \sum_{r'=0}^{r-1} T_{r'} \beta_{r'} .$$

(56)

The $(\sqrt{2} - 1)\sqrt{T_r}$ factor in the denominator of the first term of the final bound was obtained as follows:

$$\sum_{r'=0}^{r} \sqrt{T_{r'} - T_{r'-1}} = 1 + \sum_{r'=1}^{r} \sqrt{2^{r'-1}} = 1 + \frac{\sqrt{2^r} - 1}{\sqrt{2} - 1} \leqslant \frac{\sqrt{2^r}}{\sqrt{2} - 1} = \frac{\sqrt{T_r}}{\sqrt{2} - 1} .$$

Combining the first inequality of the two inequalities of Eqs. (56) with Eq. (52), we get for all $r \geqslant 1$ and $t \in [T_r, T_{r+1} - 1]$,

$$\sqrt{\mathbb{E}[\hat{d}_t^2]} \leqslant \frac{\sqrt{3B^2 + 4B\|\boldsymbol{m}\|_{\infty,2}}}{t} \left( \sqrt{t - T_r} + \sum_{r'=0}^{r} \sqrt{T_{r'} - T_{r'-1}} \right)$$

$$+ \frac{4\|\boldsymbol{m}\|_{\infty,2}}{t} \sum_{t'=1}^{t-1} \sqrt{\mathbb{E}[\mathrm{TV}^2(\mathbf{Q}, \hat{\mathbf{Q}}_{t'})]} + \frac{4}{t} \sum_{r'=0}^{r-1} T_{r'} \beta_{r'}$$

(57)

$$\leqslant \frac{\sqrt{6B^2 + 8B\|\boldsymbol{m}\|_{\infty,2}}}{(\sqrt{2} - 1)\sqrt{t}} + \frac{4\|\boldsymbol{m}\|_{\infty,2}}{t} \sum_{t'=1}^{t-1} \sqrt{\mathbb{E}[\mathrm{TV}^2(\mathbf{Q}, \hat{\mathbf{Q}}_{t'})]} + \frac{4}{t} \sum_{r'=0}^{r-1} T_{r'} \beta_{r'} ,$$

where the last inequality follows from

$$\sqrt{t - T_r} + \sum_{r'=0}^{r} \sqrt{T_{r'} - T_{r'-1}} \leqslant \sum_{r'=0}^{r+1} \sqrt{T_{r'} - T_{r'-1}} \leqslant \frac{\sqrt{T_{r+1}}}{\sqrt{2} - 1} = \frac{\sqrt{2T_r}}{\sqrt{2} - 1}.$$

Combining inequality (57) with (44), i.e., adding $\beta_r$ to the bound above, and using $T_r/t \leqslant 1$, we conclude the stated bound for the $L^2$ convergence.

**Almost-sure convergence.** We observe that, according to (43), by union bounds, Markov's inequality, and the third item of Assumption 3, we have

$$\mathbb{P}\left[\sup_{t \geqslant T_r} d_t \geqslant 2\varepsilon\right] \leqslant \mathbb{P}\left[\sup_{t \geqslant T_r} \hat{d}_t \geqslant \varepsilon\right] + \mathbb{P}\left[\sup_{r' \geqslant r} d(\hat{\mathcal{C}}_{r'}, \mathcal{C}) \geqslant \varepsilon\right]$$

$$\leqslant \mathbb{P}\left[\sup_{t \geqslant T_r} \hat{d}_t \geqslant \varepsilon\right] + \frac{1}{\varepsilon^2} \sum_{r' \geqslant r} \beta_{r'}^2.$$

In what follows, we bound $\mathbb{P}\left[\sup_{t \geqslant T_r} \hat{d}_t \geqslant \varepsilon\right]$ by $\Xi_r/\varepsilon^2$, where $\Xi_r$ is defined in Eq. (60).

As in Theorem 1, we introduce a super-martingale $S_t$ bounding $\hat{d}_t^2$ and whose expectation vanishes; however, the analysis is more involved here due to additional difficulties connected to handling the switches between regimes. More precisely, let us define, for $t \in [T_r, T_r - 1]$,

$$V_t = \frac{B^2}{(t+1)^2} + \frac{2t}{(t+1)^2} \left(4\|\boldsymbol{m}\|_{\infty,2} \, \hat{d}_t \cdot \mathrm{TV}(\mathbf{Q}, \hat{\mathbf{Q}}_t) + B(B + 2\|\boldsymbol{m}\|_{\infty,2}) \, \mathbb{I}\{\Omega_r^c\}\right)$$

$$+ 2B \, d(\hat{C}_r, \hat{C}_{r+1}) \, \mathbb{I}\{t = T_{r+1} - 1\}.$$

Using the above defined $V_t$, we additionally introduce the process

$$S_T = \hat{d}_T^2 + \sum_{t \geqslant T} \mathbb{E}[V_t | H_T]. \tag{58}$$

We observe that, by Assumption 3 and the triangle inequality,

$$\hat{d}_{T_{r+1}}^2 - \tilde{d}_{T_{r+1}}^2 = (\hat{d}_{T_{r+1}} - \tilde{d}_{T_{r+1}})\overbrace{(\hat{d}_{T_{r+1}} + \tilde{d}_{T_{r+1}})}^{\leqslant 2B}$$

$$\leqslant 2B\left(\|\overline{\boldsymbol{m}}_{T_{r+1}} - \mathrm{Proj}_{\hat{\mathcal{C}}_{r+1}}(\overline{\boldsymbol{m}}_{T_{r+1}})\| - \|\overline{\boldsymbol{m}}_{T_{r+1}} - \mathrm{Proj}_{\hat{\mathcal{C}}_r}(\overline{\boldsymbol{m}}_{T_{r+1}})\|\right)$$

$$\leqslant 2B\left(\|\overline{\boldsymbol{m}}_{T_{r+1}} - \mathrm{Proj}_{\hat{\mathcal{C}}_{r+1}}(\mathrm{Proj}_{\hat{\mathcal{C}}_r}(\overline{\boldsymbol{m}}_{T_{r+1}}))\| - \|\overline{\boldsymbol{m}}_{T_{r+1}} - \mathrm{Proj}_{\hat{\mathcal{C}}_r}(\overline{\boldsymbol{m}}_{T_{r+1}})\|\right)$$

$$\leqslant 2B\left\|\mathrm{Proj}_{\hat{\mathcal{C}}_r}(\overline{\boldsymbol{m}}_{T_{r+1}}) - \mathrm{Proj}_{\hat{\mathcal{C}}_{r+1}}(\mathrm{Proj}_{\hat{\mathcal{C}}_r}(\overline{\boldsymbol{m}}_{T_{r+1}}))\right\|$$

$$\leqslant 2B \cdot d(\hat{C}_r, \hat{C}_{r+1}).$$

Thus, in view of Eq. (50), and recalling that the right-hand side of Eq. (50) bounds rather $\tilde{d}_{T_{r+1}}$ at $t = T_{r+1} - 1$, the following recursive relation holds for any $t \in [T_r, T_{r+1} - 1]$:

$$\hat{d}_{t+1}^2 \leqslant \hat{d}_t^2 + V_t + \frac{2t}{(t+1)^2} Z_{t+1}.$$

Recalling that $\mathbb{E}[Z_{t+1} | H_t] = 0$, we deduce

$$\mathbb{E}[S_{T+1} | H_T] = \mathbb{E}[\hat{d}_{T+1}^2 | H_T] + \sum_{t \geqslant T+1} \mathbb{E}[V_t | H_T] \leqslant \hat{d}_T^2 + \sum_{t \geqslant T} \mathbb{E}[V_t | H_T] = S_T,$$

which means that $(S_T)_{T \geqslant 1}$ is a super-martingale.

Since, by definition of $S_T$, it holds that $\hat{d}_T^2 \leqslant S_T$, Doob's maximal inequality for non-negative super-martingales (Lemma 2) gives

$$\mathbb{P}\left(\sup_{t \geqslant T_r} \hat{d}_t \geqslant \varepsilon\right) \leqslant \mathbb{P}\left(\sup_{t \geqslant T_r} S_t \geqslant \varepsilon^2\right) \leqslant \frac{\mathbb{E}[S_{T_r}]}{\varepsilon^2}.$$

It only remains to bound $\mathbb{E}[S_{T_r}]$ by $\Xi_r$.

Note that by the Cauchy-Schwarz inequality and the bound of Eq. (55),

$$\mathbb{E}\big[d(\hat{\mathcal{C}}_{r'},\hat{\mathcal{C}}_{r'+1})\big] \leqslant \sqrt{\mathbb{E}\big[d(\hat{\mathcal{C}}_{r'},\hat{\mathcal{C}}_{r'+1})^2\big]} \leqslant 2\beta_{r'}\,.$$

Thanks to this inequality, to $t\,\mathbb{P}(\Omega_r^c) \leqslant t/(2T_r) \leqslant 1$ for $t \in [T_r, T_{r+1}-1]$, and other manipulations that are standard by now, the expectation of the sum appearing in the definition (58) of the super-martingale $S_T$ can be bounded as

$$\sum_{t\geqslant T_r}\mathbb{E}\left[V_t\right] \leqslant \sum_{t\geqslant T_r}\frac{3B^2+4B\|\boldsymbol{m}\|_{\infty,2}}{(t+1)^2} + \sum_{t\geqslant T_r}\frac{8t\|\boldsymbol{m}\|_{\infty,2}}{(t+1)^2}\sqrt{\mathbb{E}[\hat{d}_t^2]}\sqrt{\mathbb{E}[\mathrm{TV}^2(\mathbf{Q},\hat{\mathbf{Q}}_t)]}$$

$$+ 2B\sum_{r'\geqslant r}\mathbb{E}\big[d(\hat{\mathcal{C}}_{r'},\hat{\mathcal{C}}_{r'+1})\big]$$

$$\leqslant \frac{3B^2+4B\|\boldsymbol{m}\|_{\infty,2}}{T_r} + 4B\sum_{r'\geqslant r}\beta_{r'} + \sum_{t\geqslant T_r}\frac{8\|\boldsymbol{m}\|_{\infty,2}}{t}\sqrt{\mathbb{E}[\hat{d}_t^2]}\sqrt{\mathbb{E}[\mathrm{TV}^2(\mathbf{Q},\hat{\mathbf{Q}}_t)]}.$$

(59)

To bound the right hand side of the above inequality, we observe that for $t \geqslant T_r$, by Eq. (57), we have

$$\sqrt{\mathbb{E}[\hat{d}_t^2]} \leqslant \frac{\sqrt{6B^2+8B\|\boldsymbol{m}\|_{\infty,2}}}{(\sqrt{2}-1)\sqrt{t}} + \overbrace{4\|\boldsymbol{m}\|_{\infty,2}\max_{t\geqslant T_r}\frac{1}{t}\sum_{t'=1}^{t-1}\sqrt{\mathbb{E}[\mathrm{TV}^2(\mathbf{Q},\hat{\mathbf{Q}}_{t'})]}}^{=:\overline{\Delta}_{T_r}^*}$$

$$+ \underbrace{4\max_{r''\geqslant r}\frac{1}{T_{r''}}\sum_{r'=0}^{r''-1}T_{r'}\beta_{r'}}_{=:\beta_r^*}\,.$$

Substituting the above bound into Eq. (59), using $\sum_{t\geqslant T_r}t^{-3/2} \leqslant 2/\sqrt{T_r-1}$ and $\mathrm{TV}(\mathbf{Q},\hat{\mathbf{Q}}_{t'}) \leqslant 1$, we obtain

$$\sum_{t\geqslant T_r}\mathbb{E}\left[V_t\right] \leqslant \frac{3B^2+4B\|\boldsymbol{m}\|_{\infty,2}}{T_r} + 16\|\boldsymbol{m}\|_{\infty,2}\frac{\sqrt{6B^2+8B\|\boldsymbol{m}\|_{\infty,2}}}{(\sqrt{2}-1)\sqrt{T_r-1}} + 4B\sum_{r'\geqslant r}\beta_{r'}$$

$$+ 32\|\boldsymbol{m}\|_{\infty,2}\big(\|\boldsymbol{m}\|_{\infty,2}\overline{\Delta}_{T_r}^* + \beta_r^*\big)\sum_{t\geqslant T_r}\frac{1}{t}\sqrt{\mathbb{E}[\mathrm{TV}^2(\mathbf{Q},\hat{\mathbf{Q}}_{t'})]}\,.$$

Finally, we take into account the definition of the super martingale $S_T$ in Eq. (58) and the upper bound of Eq. (56), which we square, using that $(x+y+z)^2 \leqslant 2x^2+2(y+z)^2$. Doing so, and performing some crude boundings for the sake of readability, we get the final bound $\mathbb{E}[S_{T_r}] \leqslant \Xi_r$, where

$$\Xi_r := (1+2(\sqrt{2}-1)^{-2})\frac{3B^2+4B\|\boldsymbol{m}\|_{\infty,2}}{T_r} + 16\|\boldsymbol{m}\|_{\infty,2}\frac{\sqrt{6B^2+8B\|\boldsymbol{m}\|_{\infty,2}}}{(\sqrt{2}-1)\sqrt{T_r-1}} + 4B\sum_{r'\geqslant r}\beta_{r'}$$

$$+ 32\big(\|\boldsymbol{m}\|_{\infty,2}\overline{\Delta}_{T_r}^* + \beta_r^*\big)\left(\|\boldsymbol{m}\|_{\infty,2}\overline{\Delta}_{T_r}^* + \beta_r^* + \|\boldsymbol{m}\|_{\infty,2}\sum_{t\geqslant T_r}\frac{1}{t}\sqrt{\mathbb{E}[\mathrm{TV}^2(\mathbf{Q},\hat{\mathbf{Q}}_{t'})]}\right).$$

(60)

As indicated in Eq. (42), the Cesaro averages $\overline{\beta}_r$, which are positive, tend to 0; therefore, we also have $\beta_r^* \to 0$. For similar reasons, and as already noted for Theorem 1, the term $\overline{\Delta}_{T_r}^*$ also vanishes under Assumption 1. The latter also implies that the final term in Eq. (60) vanishes. Other terms clearly vanish or were already discussed for the $L^2$-convergence. All in all, $\Xi_r \to 0$, as claimed. $\square$