# OpenReview forum: "A Unified Approach to Fair Online Learning via Blackwell Approachability"
_NeurIPS.cc/2021/Conference — NeurIPS 2021 Spotlight_

### Official Review · Reviewer_yF4A · 2021-07-09

**Rating:** 7
**Confidence:** 4

**Summary:**

The paper looks at the online learning problem with different fairness objectives/constraints from the lens of Blackwell Approachability. Doing so allows one to see what fairness objectives (e.g. group-wise calibration/regret) with possibly additional fairness constraints (e.g. statistical parity) are compatible and immediately give an algorithm, if compatible.

Also, they have given a pretty tight characterization of the pareto frontier for group-wise calibration under demographic parity constraint.

Finally, they give how to try to approach a set where some of the parameters characterizing the set is unknown.

**Limitations And Societal Impact:**

It would be helpful to have even just a few sentences regarding the limitation of this framework. For instance, as mentioned above, in the case of possibly exponentially many groups, will the sample complexity/time complexity of the algorithm still be polynomial?

**Main Review:**

Originality:

-The connection between Blackwell approachability and online learning has been made before. However, this paper also considers various fairness objectives, more specifically group-wise calibration/regret, and fairness constraints such as statistical parity and equalized average payoffs. It is novel to use the Blackwell approachability result to see the compatibility of different objectives and constraints.

Quality/clarity:

-Notations developed here are pretty precise (some comments on notations below though), and the paper is mostly written clearly. However, there are some parts where some of the details are not clear (e.g. comment below on estimating the unknown set in section 5). Also, it would be helpful to have the contribution of this paper more expounded upon especially in relations to the other related work. For instance, it has been known how to leverage blackwell approachability to derive online calibration, and Gupta et al shows using techniques similar to blackwell approachability how to achieve online multicalibration (groupwise calibration). These kinds of discussions seem to be missing.

Significance:

-The framework (looking at online learning problem via Blackwell Approachability) is significant in that even for any new fairness objectives/constraints in the online learning setting, one can immediately see whether it's compatible and and immediately get an algorithm, if compatible.

Question:

-In terms of the impossibility results, the original Blackwell approachability provides that the set is approachable if and only if the condition that resembles best response condition (i.e. no matter what the the other player plays, you can land in the set) is met. Using this idea, the paper, for instance, shows that the set that corresponds to group-wise no-regret is not approachable. But shouldn't the final statement be regarding online learnability — that is, doesn't one need to show that if a set is not Blackwell-approachable, then the corresponding learning problem isn't possible? It seems like one would need to invoke the equivalence result of "Blackwell Approachability and No-Regret Learning are Equivalent" by Abernethy et al?

-In section 6, it's not clear to me how \hat{C}_r is constructed to estimate the unknown set C.

Minor questions/Comments/Suggestions:

-section 2.1, objective 1: N is never defined; I'm assuming it should be N = |A| I think?

-I think the dependence on |S| here will be linear. I wonder whether it's be possible to achieve logarithmic dependence on |S| using the trick in Gupta (using the exponential surrogate loss).

-In Theorem 1, d^2_T hasn't been defined. I'm assuming it's min_{v \in C} ||m_T - v|| the rate of convergence toward the set you are trying to approach to?

____________________________________________________________________________________________
Thanks for answering my questions. I recommend acceptance!

**Time Spent Reviewing:**

3.5

---

> ### Author Response · Authors · 2021-08-06
> **Detailed discussion of 2 references + Clarification on the impossibility of online learning + Estimation of target set**
>
> We fully agree with the evaluation written and thank the reviewer for her/his time and energy.
>
> The reviewer writes "Also, it would be helpful to have the contribution of this paper more expounded upon especially in relations to the other related work. For instance, it has been known how to leverage Blackwell approachability to derive online calibration, and Gupta et al shows using techniques similar to blackwell approachability how to achieve online multicalibration (groupwise calibration). These kinds of discussions seem to be missing."
> --> We acknowledge how online calibration can be solved by Blackwell's approachability on page 3, lines 115–116: ``Mannor and Stoltz [17] and Abernethy et al. [1] rewrote the problem of approximate calibration as an approachability problem as follows […]’'
> --> It is true that because of space constraints we did not discuss in detail the results of Gupta et al (2021)---and neither did we for Hébert-Johnson et al. (2018)---but we cite both papers twice on the first page of the submission, acknowledging that they also considered group-wise calibration. A detailed discussion of our results compared to the ones of Gupta et al. (2021) would be along the following lines: There are key differences between our framework and that of Gupta et al (2021). Indeed, Gupta et al (2021) allow the player to observe the sensitive context (group G in their notation), while we mainly focus on the unawareness setup (s_t is not necessarily revealed, in our notation). The knowledge of the group membership is crucial for their Algorithm 2, which explicitly relies on it; even simpler, when the groups are known, one could simply run several calibration algorithms (one per group) in parallel. Also, the group structure in Gupta et al (2021) is different from ours. In our case it has stochastic nature, while they work with deterministic, possibly overlapping, groups (but known beforehand).
>
> The reviewer also writes "In terms of the impossibility results..."
> --> See Part I of Section A of the supplementary material: if the condition is not met, there exists a mixed action for Nature (to be used at each round) such that the player will always be \alpha-away from the target, i.e., that no online learnability is possible. This follows from Blackwell's original (1956) approach and corresponds to the necessity part of the approachability theorem. The contribution by Abernethy et al (2011) is providing something different---a different proof of the sufficiency part of the approachability theorem, based on online linear optimization.
>
> The discussions above will included in the revised version.
>
> The reviewer asks "In section 6, it's not clear to me how \hat{C}_r is constructed to estimate the unknown set C."
> --> Indeed, the construction is idiosyncratic; we provide an example of such a construction in Section D.1 of the supplementary material for the case of group-wise calibration under a demographic parity constraint (= the case discussed in Section 5). We will better highlight the fact that the target estimates construction have to be problem dependent.
>
> We agree with all minor comments/suggestions and will implement them.
>
> We will write a paragraph on the limitations, discussing the complexity of the algorithm in terms of the numbers of groups.

---

### Official Review · Reviewer_eMwY · 2021-07-16

**Rating:** 7
**Confidence:** 2

**Summary:**

The authors aim to provide a framework, extended from Blackwell approachability, to study online learning with fairness constraints, in the presence of stochastic contexts. The authors assume that the distribution of contexts is unknown and needs to be estimated over time. They use the approachability framework to provide conditions under which certain learning objectives are compatible with certain definitions of fairness.

**Limitations And Societal Impact:**

The paper should have a positive impact as it aims to provide a framework and algorithms for online learning with fairness constraints.

**Main Review:**

Strengths:
- The connection between Blackwell approachability and group notions of fairness (rather than just calibration) seems relatively novel.
- Using this connection, the authors are able to study several group notions of fairness and to figure out whether online learning with said fairness constraints is possible, based on checking whether the approachability condition (2) holds. Of particular interest is the fact that the authors are able to provide new (to the best of my knowledge) results on the trade-off that is achievable between group-wise calibration and demographic parity.
- The authors provide rates of convergence based on the Blackwell approachability framework.
- They extend their results to the case where the set to approach is unknown, and show how to use this extension to provide convergence results even when some of the priors on the probability of being in each group is unknown to the learner.

Weaknesses:
- It is a bit hard from reading the paper to figure out what are the authors’ contributions versus what is new. My understanding is that the connection between Blackwell approachability and no-regret learning was previously known, but that does not seem obvious from reading the paper. Note that I am also not sure about the technical novelty of the paper, since I am not an expert on Blackwell approachability.


**Time Spent Reviewing:**

3-4

---

> ### Author Response · Authors · 2021-08-06
> **Connections between Blackwell's approachability and no-regret learning are indeed known since the end of the 90s;  our main contribution (and other reviewers agree on this) is to attack online learning under fairness constraints in a unified way through Blackwell's approachability**
>
> We fully agree with the evaluation written and thank the reviewer for her/his time and energy.
>
> The reviewier writes "It is a bit hard from reading the paper to figure out what are the authors’ contributions versus what is new. My understanding is that the connection between Blackwell approachability and no-regret learning was previously known, but that does not seem obvious from reading the paper. Note that I am also not sure about the technical novelty of the paper, since I am not an expert on Blackwell approachability."
>
> --> Yes, connections between Blackwell's approachability and no-regret learning are known since the end of the 90s (see, e.g., the works by Fudenberg and Levine) and there was a vast litterature on this topic since then (two main references in the learning community are the article by Abernethy et al., 2011, and the PhD thesis by Vianney Perchet on the links between approachability, no-regret and calibration). We will definitely improve the discussion of literature to make this clear. Our main contribution (and other reviewers agree on this) is to attack online learning under fairness constraints in a unified way through Blackwell's approachability. The logic is the same as for attacking online learning with no constraint by Blackwell's approachability, yet, this angle is new and leads to interesting results, like the one underlined on the trade-off that is achievable between group-wise calibration and demographic parity.

---

> > ### Comment · Reviewer_eMwY · 2021-08-22
> > **Thank you for the clarification!**
> >
> > Thank you for the clarification, and for aiming to improve the discussion of the related work!

---

### Official Review · Reviewer_gm2z · 2021-07-16

**Rating:** 9
**Confidence:** 4

**Summary:**

The paper presents a general Blackwell approachability based framework for online fair learning in the presence of stochastic (sensitive and non-sensitive)
contexts. The framework allows to easily obtain both positive and negative fair learnability results. The authors work out their framework in the standard
cases of no regret or calibrated online learning, paired with demographic parity constraints. They also present novel optimal tradeoffs between the
constraints in the case of group-calibrated learning subject to demographic parity.


**Limitations And Societal Impact:**

Yes.

**Main Review:**

I believe that this paper is an important and timely contribution to the literature of online fair learning. It successfully adapts Blackwell
approachability to handle the case of both sensitive and non-sensitive stochastic contexts coming from a fixed but unknown distribution.
It is important that, subject to a technical assumption, the framework developed by the authors provides "if and only if" learnability results, as well as
that it gives explicit rates in the learnable case. Further, while the main assumption (fast sequential estimation of the contextual distribution) may
appear technical, it is easily satisfied in most important online fairness settings studied in the literature, and thus poses virtually no problem
in terms of the generality of the "if and only if" results.

From my perspective, the identification of Pareto frontiers for group calibration under
the parity constraint is a crucial contribution of this paper. (I would like to point out, in conjunction with this, that while considering the case
S = {0, 1} is a good choice presentation-wise, but it does not immediately appear clear to me how Propositions 2 and 3 regarding the Pareto optimal curve
might look like for more general sets S.)

The proofs are involved, and might be amenable to a more compact exposition. That said, the authors do a good job with helping the reader keep track of
the various assumptions made at various junctures in the proofs.

Some minor corrections:
-The display below line 91, the second regret term should be r(a, b_t, x_t, s_t), not r(a'_s, b_t, x_t, s_t).
-Line 144, I believe it should say "perfect knowledge of the former", not "the latter" (the former being the payoff functions, which are
exposed to the learner as per line 55).
-Line 162, I suggest adding the word "below" after "Assumption 1", as the assumption has not been defined yet.

**Time Spent Reviewing:**

20

---

> ### Author Response · Authors · 2021-08-06
> **We fully agree with the evaluation written**
>
> We fully agree with the evaluation written and thank the reviewer for her/his time and energy.
> We will correct the typos spotted and will think about the generalization to more than 2 groups as far as Propositions 2 and 3 are concerned!

---

### Official Review · Reviewer_oHXk · 2021-07-20

**Rating:** 7
**Confidence:** 3

**Summary:**

The paper studies an online setting formulated as a game between Player and Nature. On each round, a context is drawn stochastically, consisting of non-sensitive and sensitive parts. Player may only observe the non-sensitive part before making a decision, while both cases of Nature observing the entire context or only the non-sensitive part are discussed. After Nature’s decision, a reward based on the context and two decisions is given to Player. Using this formulation, the authors wish to unify previously suggested approaches to fair online learning, adapting Blackwell’s Approachability theory to handle unknown contexts’ distribution. The suggested framework assists them then in studying the compatibility of previously considered objectives [(vanilla/group-wise) no regret, (vanilla/group-wise) calibration] and fairness constraints (statistical parity, equalized average payoffs), where they provide a general sufficient and necessary condition for such compatibility, and study several specific instantiations.

**Limitations And Societal Impact:**

There is no discussion regarding limitations of the suggested framework and results. Please include one.

**Main Review:**

I think that the unified approach provided by the authors makes a nice contribution, as it allows a clean, elegant abstraction and understanding of multiple previously studied objectives and constraints. The suggested framework allows lifting and minimizing implementational details we have seen in previous work (for example, the authors give a condition which directly applies to the convergence rate of the empirical distribution estimate to the true underlying distribution, as opposed to, for example, a VC-bound), which abstracts these parts away, allowing to focus on the objectives and constraints in a cleaner way.

Establishing this unified approach is in my opinion the main contribution of the paper. I do not think the specific compatibility/incompatibility results that were shown were very surprising. However, I do believe the paper gives a very formal analysis of the cases and additional assumptions and their implication on the compatibility, or rather the resulting trade-offs.

The paper is well-written and presentation (translation of the objectives and constraints, adaptation of Blackwell’s theory, specific instantiations) was done in a coherent, easy to read, manner.

Comments/Questions to authors:
1. The authors refer (line 26) to Bechavod et al. 2019, as one example of an online setting with a fairness constraint. However, the problem studied in that paper is of one-sided feedback (also known as "apple tasting"). I am aware of at least one generalization of Blackwell’s approachability theory to handle such partial monitoring (Kwon & Perchet 2017). I think this is relevant to state in the paper, as many settings where fairness is a concern tend to take this form of feedback.
2. (Line 269) How is \tau computed/estimated?
3. There is no discussion regarding the limitations of the suggested framework and results. Can you elaborate on these?

Fixes:
1.	(Line 26) Bechavod et al. study a partial feedback setting, not a delayed feedback one.
2.	(Line 91) In the second term, a_s’ -> a.
3.	(Line 142) “criterion” -> “criteria”.

The submission appears to be technically sound, although I have only glanced at the details in the appendix.

**Time Spent Reviewing:**

12

---

> ### Author Response · Authors · 2021-08-06
> **All the compatibility/incompatibility results exhibited were indeed expected or even known, but we go beyond and describe the best possible trade-offs between the two conflicting criteria (accuracy vs. fairness)**
>
>  We generally agree with the evaluation written and thank the reviewer for her/his time and energy, including the fact that all the compatibility/incompatibility results exhibited were expected or even known. However, as underlined by another reviewer, the trade-off part (Section 5, when group-wise calibration is considered under a demographic parity constraint) is a crucial application of our unified framework. It was expected, of course, that the two criteria of group-wise calibration and demographic parity cannot be satisfied simultaneously in general, but we go beyond this incompatibility and describe the best possible trade-offs between the two conflicting criteria (accuracy vs. fairness). Our approach allows to obtain some optimal balance.
>
> Comments/questions:
> 1. Yes, we will add such a discussion.
> 2. \tau is a parameter of the user---the user sets the trade-off as she/he wishes between the two conflicting criteria.
> 3. We will include one (based on the suggestions of the other reviewers).

---

> > ### Comment · Reviewer_oHXk · 2021-09-02
> > **Thank you for the response**
> >
> > Thank you for the response. Following this discussion and reconsidering, I have decided to update my score.

---

### Decision · Program_Chairs · 2021-09-27

**Decision:**

Accept (Spotlight)

**Comment:**

Thanks for the strong submission! The reviewers unanimously enjoyed the paper.